# Barcoded bulk QTL mapping reveals highly polygenic and epistatic architecture of complex traits in yeast

Alex N Nguyen Ba[1†‡], Katherine R Lawrence[2,3,4†], Artur Rego-Costa[1†], Shreyas Gopalakrishnan[1,5], Daniel Temko[6,7,8], Franziska Michor[6,7,8,9,10,11], Michael M Desai[1,2,3,12]*

[1]Department of Organismic and Evolutionary Biology, Harvard University, Cambridge, United States; [2]NSF-Simons Center for Mathematical and Statistical Analysis of Biology, Harvard University, Cambridge, United States; [3]Quantitative Biology Initiative, Harvard University, Cambridge, United States; [4]Department of Physics, Massachusetts Institute of Technology, Cambridge, United States; [5]Department of Molecular and Cellular Biology, Harvard University, Cambridge, United States; [6]Department of Data Science, Dana-Farber Cancer Institute, Boston, United States; [7]Department of Biostatistics, Harvard T.H. Chan School of Public Health, Boston, United States; [8]Department of Stem Cell and Regenerative Biology, Harvard University, Cambridge, United States; [9]Center for Cancer Evolution, Dana-Farber Cancer Institute, Boston, United States; [10]The Ludwig Center at Harvard, Boston, United States; [11]The Broad Institute of MIT and Harvard, Cambridge, United States; [12]Department of Physics, Harvard University, Cambridge, United States

*For correspondence:
mdesai@oeb.harvard.edu

†These authors contributed equally to this work

Present address: ‡Department of Biology, University of Toronto at Mississauga, Mississauga, Canada

Competing interest: The authors declare that no competing interests exist.

**Abstract** Mapping the genetic basis of complex traits is critical to uncovering the biological mechanisms that underlie disease and other phenotypes. Genome-wide association studies (GWAS) in humans and quantitative trait locus (QTL) mapping in model organisms can now explain much of the observed heritability in many traits, allowing us to predict phenotype from genotype. However, constraints on power due to statistical confounders in large GWAS and smaller sample sizes in QTL studies still limit our ability to resolve numerous small-effect variants, map them to causal genes, identify pleiotropic effects across multiple traits, and infer non-additive interactions between loci (epistasis). Here, we introduce barcoded bulk quantitative trait locus (BB-QTL) mapping, which allows us to construct, genotype, and phenotype 100,000 offspring of a budding yeast cross, two orders of magnitude larger than the previous state of the art. We use this panel to map the genetic basis of eighteen complex traits, finding that the genetic architecture of these traits involves hundreds of small-effect loci densely spaced throughout the genome, many with widespread pleiotropic effects across multiple traits. Epistasis plays a central role, with thousands of interactions that provide insight into genetic networks. By dramatically increasing sample size, BB-QTL mapping demonstrates the potential of natural variants in high-powered QTL studies to reveal the highly polygenic, pleiotropic, and epistatic architecture of complex traits.

## Editor's evaluation

This impressive study not only expands the identification of small-effect QTL, but also reveals epistatic interactions at an unprecedented scale. The approach takes advantage of DNA barcodes to increase the scale of genetic mapping studies in yeast by an order of magnitude over previous

studies, yielding a more complete and precise view of the QTL landscape and confirming widespread epistatic interactions between the different QTL.

## Introduction

In recent years, the sample size and statistical power of genome-wide association studies (GWAS) in humans has expanded dramatically (*Bycroft et al., 2018*; *Eichler et al., 2010*; *Manolio et al., 2009*). Studies investigating the genetic basis of important phenotypes such as height, BMI, and risk for diseases such as schizophrenia now involve sample sizes of hundreds of thousands or even millions of individuals. The corresponding increase in power has shown that these traits are very highly polygenic, with a large fraction of segregating polymorphisms (hundreds of thousands of loci) having a causal effect on phenotype (*Yang et al., 2010*; *International Schizophrenia Consortium et al., 2009*). However, the vast majority of these loci have extremely small effects, and we remain unable to explain most of the heritable variation in many of these traits (the 'missing heritability' problem; *Manolio et al., 2009*).

In contrast to GWAS, quantitative trait locus (QTL) mapping studies in model organisms such as budding yeast tend to have much smaller sample sizes of at most a few thousand individuals (*Steinmetz et al., 2002*; *Bloom et al., 2013*; *Burga et al., 2019*; *Mackay and Huang, 2018*; *Bergelson and Roux, 2010*). Due to their lower power, most of these studies are only able to identify relatively few loci (typically at most dozens, though see below) with a causal effect on phenotype. Despite this, these few loci explain most or all of the observed phenotypic variation in many of the traits studied (*Fay, 2013*).

The reasons for this striking discrepancy between GWAS and QTL mapping studies remain unclear. It may be that segregating variation in human populations has different properties than the between-strain polymorphisms analyzed in QTL mapping studies, or the nature of the traits being studied may be different. However, it is also possible that the discrepancy arises for more technical reasons associated with the limitations of GWAS and/or QTL mapping studies. For example, GWAS studies suffer from statistical confounders due to population structure, and the low median minor allele frequencies in these studies limit power and mapping resolution (*Bloom et al., 2013*; *Sohail et al., 2019*; *King et al., 2012*; *Consortium, 2012*). These factors make it difficult to distinguish between alternative models of genetic architecture, or to detect specific individual small-effect causal loci. Thus, it may be the case that the highly polygenic architectures apparently observed in GWAS studies are at least in part artifacts introduced by these confounding factors. Alternatively, the limited power of existing QTL mapping studies in model organisms such as budding yeast (perhaps combined with the relatively high functional density of these genomes) may cause them to aggregate numerous linked small-effect causal loci into single large-effect 'composite' QTL. This would allow these studies to successfully explain most of the observed phenotypic heritability in terms of an apparently small number of causal loci, even if the true architecture was in fact highly polygenic (*Fay, 2013*).

More recently, numerous studies have worked to advance the power and resolution of QTL mapping studies, and have begun to shed light on the discrepancy with GWAS (*Bloom et al., 2013*; *Bloom et al., 2015*; *She and Jarosz, 2018*; *Bloom et al., 2019*; *Cubillos et al., 2011*). One direction has been to use advanced crosses to introduce more recombination breakpoints into mapping panels (*She and Jarosz, 2018*). This improves fine-mapping resolution and under some circumstances may be able to resolve composite QTL into individual causal loci, but it does not in itself improve power to detect small-effect alleles. Another approach is to use a multiparental cross (*Cubillos et al., 2013*) or multiple individual crosses (e.g. in a round-robin mating; *Bloom et al., 2019*). Several recent studies have constructed somewhat larger mapping panels with this type of design (as many as 14,000 segregants; *Bloom et al., 2019*); these offer the potential to gain more insight into trait architecture by surveying a broader spectrum of natural variation that could potentially contribute to phenotype. However, because multiparental crosses reduce the allele frequency of each variant (and in round-robin schemes each variant is present in only a few matings), these studies also have limited power to detect small-effect alleles. Finally, several recent studies have constructed large panels of diploid genotypes by mating smaller pools of haploid parents (e.g. a 384 × 104 mating leading to 18,126 F6 diploids; *Jakobson and Jarosz, 2019*). These studies are essential to understand potential dominance effects. However, the ability to identify small-effect alleles scales only with the number of unique

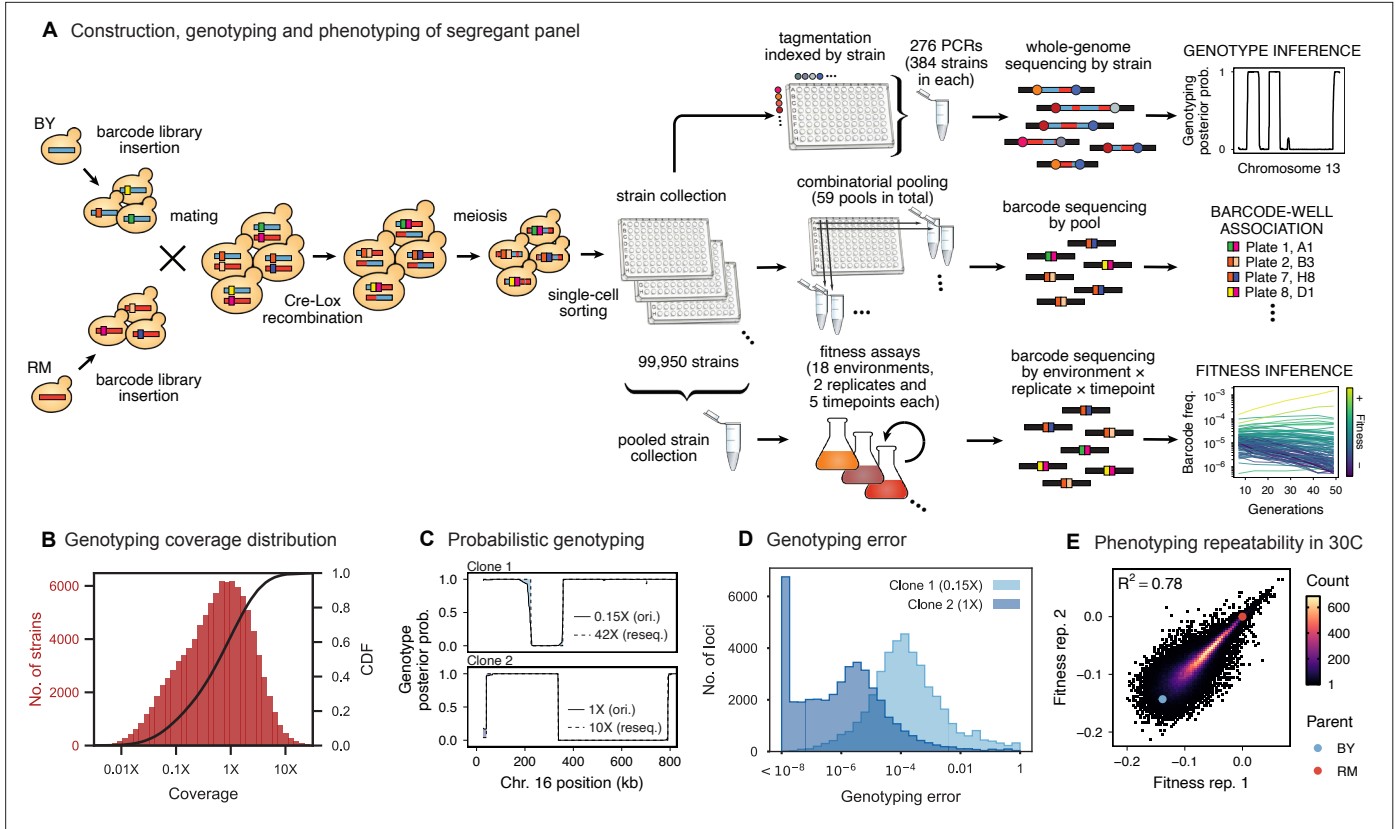

**Figure 1.** Cross design, genotyping, phenotyping, and barcode association. (**A**) Construction, genotyping, and phenotyping of segregant panel. Founding strains BY (blue) and RM (red) are transformed with diverse barcode libraries (colored rectangles) and mated in bulk. Cre recombination combines barcodes onto the same chromosome. After meiosis, sporulation, and selection for barcode retention, we sort single haploid cells into 96-well plates. Top: whole-genome sequencing of segregants via multiplexed tagmentation. Middle: barcode-well association by combinatorial pooling. Bottom: bulk phenotyping by pooled competition assays and barcode frequency tracking. See **Figure 1—figure supplements 1–3**, and Materials and methods for details. (**B**) Histogram and cumulative distribution function (CDF) of genotyping coverage of our panel (**Figure 1—source data 1**). (**C**) Inferred probabilistic genotypes for two representative individuals from low coverage (solid) and high coverage (dashed) sequencing, with the genotyping error (difference between low and high coverage probabilistic genotypes) indicated by shaded blue regions (**Figure 1—source data 2**). (**D**) Distribution of genotyping error by SNP for the two individuals shown in (**C**). (**E**) Reproducibility of phenotype measurements in 30 C environment (see **Figure 1—figure supplement 4** for other environments). Here, fitness values are inferred on data from each individual replicate assay. For all other analyses, we use fitness values jointly inferred across both replicates (see Appendix 2, **Figure 1—source data 3**).

The online version of this article includes the following source data and figure supplement(s) for figure 1:

**Source data 1.** Genotyping coverage of all strains in our panel.

**Source data 2.** Inferred genotype for resequenced clones in Chr XVI.

**Source data 3.** Replicate fitness measurements in 30C.

**Figure supplement 1.** Barcoding plasmids.

**Figure supplement 2.** Detailed schematic of procedure to generate 100,000 F1 segregants.

**Figure supplement 3.** Allele frequencies of the RM parental allele for the genotyped pool of 99,950 segregants.

**Figure supplement 4.** Phenotype measurement reproducibility, as in **Figure 1D**, for all other environments.

haploid parents rather than the number of diploid genotypes, so these studies also lack power for this purpose. Thus, previous studies have been unable to observe the polygenic regime of complex traits or to offer insight into its consequences.

Here, rather than adopting any of these more complex study designs, we sought to increase the power and resolution of QTL mapping in budding yeast simply by dramatically increasing sample size. To do so, we introduce a barcoded bulk QTL (BB-QTL) mapping approach that allows us to construct and measure phenotypes in a panel of 100,000 F1 segregants from a single yeast cross, a sample size almost two orders of magnitude larger than the current state of the art (**Figure 1A**). We combined

several recent technical advances to overcome the challenges of QTL mapping at the scale of 100,000 segregants: (*i*) unique DNA barcoding of every strain, which allows us to conduct sequencing-based bulk phenotype measurements; (*ii*) a highly multiplexed sequencing approach that exploits our knowledge of the parental genotypes to accurately infer the genotype of each segregant from low-coverage (<1x) sequence data; (*iii*) liquid handling robotics and combinatorial pooling to create, array, manipulate, and store this segregant collection in 96/384-well plates; and (*iv*) a highly conservative cross-validated forward search approach to confidently infer large numbers of small-effect QTL.

Using this BB-QTL approach, we mapped the genetic basis of 18 complex phenotypes. Despite the fact that earlier lower-powered QTL mapping studies in yeast have successfully explained most or all of the heritability of similar phenotypes with models involving only a handful of loci, we find that the increased power of our approach reveals that these traits are in fact highly polygenic, with more than a hundred causal loci contributing to almost every phenotype. We also exploit our increased power to investigate widespread patterns of pleiotropy across the eighteen phenotypes, and to analyze the role of epistatic interactions in the genetic architecture of each trait.

## Results
### Construction of the barcoded segregant panel
To generate our segregant collection, we began by mating a laboratory (BY) and vineyard (RM) strain (*Figure 1A*), which differ at 41,594 single-nucleotide polymorphisms (SNPs) and vary in many relevant phenotypes (*Bloom et al., 2013*). We labeled each parent strain with diverse DNA barcodes (a random sequence of 16 nucleotides), to create pools of each parent that are isogenic except for this barcode (12 and 23 pools of ~1000 unique barcodes in the RM and BY parental pools, respectively). Barcodes are integrated at a neutral locus containing *Cre-Lox* machinery for combining barcodes, similar to the 'renewable' barcoding system we introduced in recent work (*Nguyen Ba et al., 2019*). We then created 276 sets by mating all combinations of parental pools to create heterozygous RM/BY diploids, each of which contains one barcode from each parent. After mating, we induce *Cre-Lox* recombination to assemble the two barcodes onto the same chromosome, creating a 32-basepair double barcode. After sporulating the diploids and selecting for doubly-barcoded haploid *MATa* offspring using a mating-type specific promoter and selection markers (*Tong et al., 2001*), we used single-cell sorting to select ~100,000 random segregants and to array them into individual wells in 1,104 96-well plates. Because there are over 1 million possible barcodes per set, and only 384 offspring selected per set, this random sorting is highly unlikely to select duplicates, allowing us to produce a strain collection with one uniquely barcoded genotype in each well that can be manipulated with liquid handling robotics. Finally, we identified the barcode associated with each segregant by constructing orthogonal pools (e.g. all segregants in a given 96-well plate, all segregants in row A of any 96-well plate, all segregants from a set, etc.), and sequencing the barcode locus in each pool. This combinatorial pooling scheme allows us to infer the barcode associated with each segregant in each individual well, based on the unique set of pools in which a given barcode appears (*Erlich et al., 2009*).

### Inferring segregant genotypes
We next conducted whole-genome sequencing of every strain using an automated library preparation pipeline that makes use of custom indexed adapters to conduct tagmentation in 384-well plates, after which samples can be pooled for downstream library preparation (*Figure 1A*). To limit sequencing costs, we infer segregant genotypes from low-coverage sequencing data (median coverage of 0.6 x per segregant; *Figure 1B*). We can obtain high genotyping accuracy despite such low coverage due to our cross design: because we use an F1 rather than an advanced cross, we have a high density of SNPs relative to recombination breakpoints in each individual (>700 SNPs between recombination breakpoints on average). Exploiting this fact in combination with our knowledge of the parental genotypes, we developed a Hidden Markov Model (HMM) to infer the complete segregant genotypes from this data (see Appendix 1). This HMM is similar in spirit to earlier imputation approaches (*Arends et al., 2010*; *Marchini and Howie, 2010*; *Bilton et al., 2018*); it infers genotypes at unobserved loci (and corrects for sequencing errors and index-swapping artifacts) by assuming that each segregant consists of stretches of RM and BY loci, separated by relatively sparse recombination events. We note that this

model produces probabilistic estimates of genotypes (i.e. the posterior probability that segregant genotypes is either RM or BY at each SNP; *Figure 1C*), which we account for in our analysis below.

We assessed two key aspects of the performance of this sequencing approach: the confidence with which it infers genotypes, and the accuracy of the genotypes assigned. We find that at 0.1 x coverage and above, our HMM approach confidently assigns genotypes at almost all loci (posterior probability of >92% of the inferred genotype at >99% of loci; see *Appendix 1—figure 2* and Appendix 1 for a discussion of our validation of these posterior probability estimates). Loci not confidently assigned to either parental genotype largely correspond to SNPs in the immediate vicinity of breakpoints, which cannot be precisely resolved with low-coverage sequencing (we note that these uncertainties do affect mapping resolution, as the precise location of breakpoints is important for this purpose; see Appendix 1-1.4 for an extensive discussion and analysis of this uncertainty). To assess the accuracy of our genotyping, we conducted high-coverage sequencing of a small subset of segregants and compared the results to the inferred genotypes from our low-coverage data. We find that the genotyping is accurate, with detectable error only very near recombination breakpoints (*Figure 1C and D*). In addition, we find that our posterior probabilities are well calibrated (e.g. 80% of the loci with an RM posterior probability of 0.8 are indeed RM; see Appendix 1). We also note that, as expected, most SNPs are present across our segregant panel at an allele frequency of 0.5 (*Figure 1—figure supplement 3*), except for a few marker loci that are selected during engineering of the segregants.

## Barcoded bulk phenotype measurements

Earlier QTL mapping studies in budding yeast have typically assayed phenotypes for each segregant in their mapping panels independently, primarily by measuring colony sizes on solid agar plates (*Steinmetz et al., 2002*; *Bloom et al., 2013*; *Bloom et al., 2015*; *She and Jarosz, 2018*; *Bloom et al., 2019*; *Jakobson and Jarosz, 2019*). These colony size phenotypes can be defined on a variety

**Table 1.** Phenotyping growth conditions.
Summary of the eighteen competitive fitness phenotypes we analyze in this study. All assays were conducted at 30 °C, except when stated otherwise. YP: 1% yeast extract, 2% peptone. YPD: 1% yeast extract, 2% peptone, 2% glucose. SD: synthetic defined medium, 2% glucose. YNB: yeast nitrogen base, 2% glucose. Numbers of inferred additive QTL and epistatic interactions are also shown.

| Name | Description | Additive QTL | Epistatic QTL |
|------|-------------|--------------|---------------|
| 23 C | YPD, 23 °C | 112 | 185 |
| 25 C | YPD, 25 °C | 134 | 189 |
| 27 C | YPD, 27 °C | 149 | 255 |
| 30 C | YPD, 30 °C | 159 | 247 |
| 33 C | YPD, 33 °C | 147 | 216 |
| 35 C | YPD, 35 °C | 117 | 250 |
| 37 C | YPD, 37 °C | 128 | 265 |
| sds | YPD, 0.005% (w/v) SDS | 175 | 263 |
| raff | YP, 2% (w/v) raffinose | 167 | 221 |
| mann | YP, 2% (w/v) mannose | 169 | 341 |
| cu | YPD, 1 mM copper(II) sulfate | 143 | 225 |
| eth | YPD, 5% (v/v) ethanol | 149 | 247 |
| suloc | YPD, 50 µM suloctidil | 173 | 314 |
| 4NQO | SD, 0.05 µg/ml 4-nitroquinoline 1-oxide | 153 | 394 |
| ynb | YNB, w/o AAs, w/ ammonium sulfate | 145 | 303 |
| mol | molasses, diluted to 20% (w/v) sugars | 111 | 235 |
| gu | YPD, 6 mM guanidinium chloride | 185 | 277 |
| li | YPD, 20 mM lithium acetate | 83 | 42 |

of different solid media, but while they are relatively high throughput (often conducted in 384-well format), they are not readily scalable to measurements of 100,000 segregants.

Here, we exploit our barcoding system to instead measure phenotypes for all segregants in parallel, in a single bulk pooled assay for each phenotype. The basic idea is straightforward: we combine all segregants into a pool, sequence the barcode locus to measure the relative frequency of each segregant, apply some selection pressure, and then sequence again to measure how relative frequencies change (*Smith et al., 2009*). These bulk assays are easily scalable and can be applied to any phenotype that can be measured based on changes in relative strain frequencies. Because we only need to sequence the barcode region, we can sequence deeply to obtain high-resolution pheno-type measurements at modest cost. In addition, we can correct sequencing errors because the set of true barcodes is known in advance from combinatorial pooling (see above). Importantly, this system allows us to track the frequency changes of each individual in the pool, assigning a phenotype to each specific segregant genotype. This stands in contrast to 'bulk segregant analysis' approaches that use whole-genome sequencing of pooled segregant panels to track frequency changes of alleles rather than individual genotypes (*Ehrenreich et al., 2010*; *Michelmore et al., 1991*); our approach increases power and allows us to study interaction effects between loci across the genome.

Using this BB-QTL system, we investigate eighteen complex traits, defined as competitive fitness in a variety of liquid growth conditions ('environments'), including minimal, natural, and rich media with several carbon sources and a range of chemical and temperature stressors (*Table 1*). To measure these phenotypes, we pool all strains and track barcode frequencies through 49 generations of competition. We use a maximum likelihood model to jointly infer the relative fitness of each segregant in each assay—a value related to the instantaneous exponential rate of change in frequency of a strain during the course of the assay (*Figure 1A*, lower-right inset; see Appendix 2). These measurements are consistent between replicates (average $R^2$ between replicate assays of 0.77), although we note that the inherent correlation between fitness and barcode read counts means that errors are inversely correlated with fitness (*Figure 1E*; *Figure 1—figure supplement 4*). While genetic changes such as de novo mutations and ploidy changes can occur during bulk selection, we estimate their rates to be sufficiently low such that they impact only a small fraction of barcode lineages and thus do not signifi-cantly bias the inference of QTL effects over the strain collection (see Appendix 2).

## Modified stepwise cross-validated forward search approach to mapping QTL

With genotype and phenotype data for each segregant in hand, we next sought to map the locations and effects of QTL. The typical approach to inferring causal loci would be to use a forward stepwise regression (*Bloom et al., 2013*; *Arends et al., 2010*). This method proceeds by first computing a statistic such as $p$-value or LOD score for each SNP independently, to test for a statistical associa-tion between that SNP and the phenotype. The most-significant SNP is identified as a causal locus, and its estimated effect size is regressed out of the data. This process is then repeated iteratively to identify additional causal loci. These iterations proceed until no loci are identified with a statistic that exceeds a predetermined significance threshold, which is defined based on a desired level of genome-wide significance (e.g. based on a null expectation from permutation tests or assumptions about the numbers of true causal loci). However, although this approach is fast and simple and can identify large numbers of QTL, it is not conservative. Variables added in a stepwise approach do not follow the claimed $F$ or $\chi^2$-distribution, so using p-values or related statistics as a selection criterion is known to produce false positives, especially at large sample sizes or in the presence of strong linkage (*Smith, 2018*). Because our primary goal is to dissect the extent of polygenicity by resolving small-effect loci and decomposing 'composite' QTL, these false positives are particularly problematic and we therefore cannot use this traditional approach.

Fortunately, due to the high statistical power of our study design, we are better positioned to address the question of polygenicity using a more conservative method with lower false discovery rate. To do so, we carried out QTL mapping through a modified stepwise regression approach, with three key differences compared to previous methods (see Appendix 3 for details). First, we use cross-validation rather than statistical significance to terminate the model search procedure, which reduces the false positive rate. Specifically, we divide the data into training and test sets (90% and 10% of segregants respectively, chosen randomly), and add QTL iteratively in a forward stepwise regression

on the training set. We terminate this process when performance on the test set declines, and use this point to define an L0-norm sparsity penalty on the number of QTL. We repeat this process for all possible divisions of the data to identify the average sparsity penalty, and then use this sparsity penalty to infer our final model from all the data (in addition, an outer loop involving a validation set is also used to assess the performance of our final model). The second key difference in our method is that we jointly re-optimize inferred effect sizes (i.e. estimated effect on fitness of having the RM versus the BY version of a QTL) and lead SNP positions (i.e. our best estimate of the actual causal SNP for each QTL) at each step. This further reduces the bias introduced by the greedy forward search procedure. Finally, the third key difference in our approach is to estimate the 95% credible interval around each lead SNP using a Bayesian method rather than LOD-drop methods, which is more suitable in polygenic architectures. We describe and validate this modified stepwise regression approach in detail in Appendix 3. Simulations under various QTL architectures show that this approach has a low false positive rate, accurately identifies lead SNPs and credible intervals even with moderate linkage, and generally calls fewer QTL than in the true model, only missing QTL of extremely small effect sizes. The behavior of this approach is simple and intuitive: the algorithm greedily adds QTL to the model if their expected contribution to the total phenotypic variance exceeds the bias and increasing variance of the forward search procedure, which is greatly reduced at large sample size. Thus, it may fail to identify very small effect size variants and may fail to break up composite QTL in extremely strong linkage.

## Resolving the highly polygenic architecture of complex phenotypes in yeast

We used our modified stepwise cross-validated forward search to infer the genetic basis of the 18 phenotypes described in *Table 1*, assuming an additive model. We find that these phenotypes are highly polygenic: we identify well over 100 QTL spread throughout the genome for almost every trait, an order of magnitude more than that found for similar phenotypes in most earlier studies ($\sim 0.3\%$ of SNPs in our panel; *Figures 2 and 3B*). This increase can be directly attributed to our large sample size: inference on a downsampled dataset of 1000 individuals detects no more than 30 QTL for any trait (see Appendix 3).

The distribution of effect sizes of detected QTL shows a large enrichment of small-effect loci, and has similar shape (though different scale) across all phenotypes (*Figure 3C*), consistent with an exponential-like distribution above the limit of detection. This distribution suggests that further increases in sample size would reveal a further enrichment of even smaller-effect loci. While our SNP density is high relative to the recombination rate, our sample size is large enough that there are many individuals with a recombination breakpoint between any pair of neighboring SNPs (over 100 such individuals with breakpoints between each SNP on average). This allows us to precisely fine-map many of these QTL to causal genes or even nucleotides. We find that most QTL with substantial effect sizes are mapped to one or two genes, with dozens mapped to single SNPs (*Figure 3D*). In many cases these genes and precise causal nucleotides are consistent with previous mapping studies (e.g. MKT1, *Deutschbauer and Davis, 2005*; PHO84, *Perlstein et al., 2007*; HAP1, *Brem et al., 2002*); in some cases we resolve for the first time the causal SNP within a previously identified gene (e.g. IRA2, *Smith and Kruglyak, 2008*; VPS70, *Duitama et al., 2014*). However, we note that because our SNP panel does not capture all genetic variation, such as transposon insertions or small deletions, some QTL lead positions may tag linked variation rather than being causal themselves.

The SNP density in our panel and resolution of our approach highly constrain these regions of linked variation, providing guidance for future studies of specific QTL, but as a whole we find that our collection of lead SNPs displays some characteristic features of causal variants. Across all identified lead SNPs, we observe a significant enrichment of nonsynonymous substitutions, especially when considering lead SNPs with posterior probability above 0.5 (*Figure 3E*; $p < 10^{-10}$, $\chi^2$-test, $\mathrm{df} = 2$), as expected for causal changes in protein function. Lead SNPs are also more likely to be found within disordered regions of proteins (1.22 x fold increase, $p < 10^{-5}$, Fisher's exact test), even when constrained to nonsynonymous variants (1.28 x fold increase beyond the enrichment for nonsynonymous variants in disordered regions, $p < 10^{-4}$, Fisher's exact test), indicating potential causal roles in regulation (*Dyson and Wright, 2005*). Lead SNP alleles, especially those with large effect size, are observed at significantly lower minor allele frequencies (MAF) in the 1011 Yeast Genomes collection

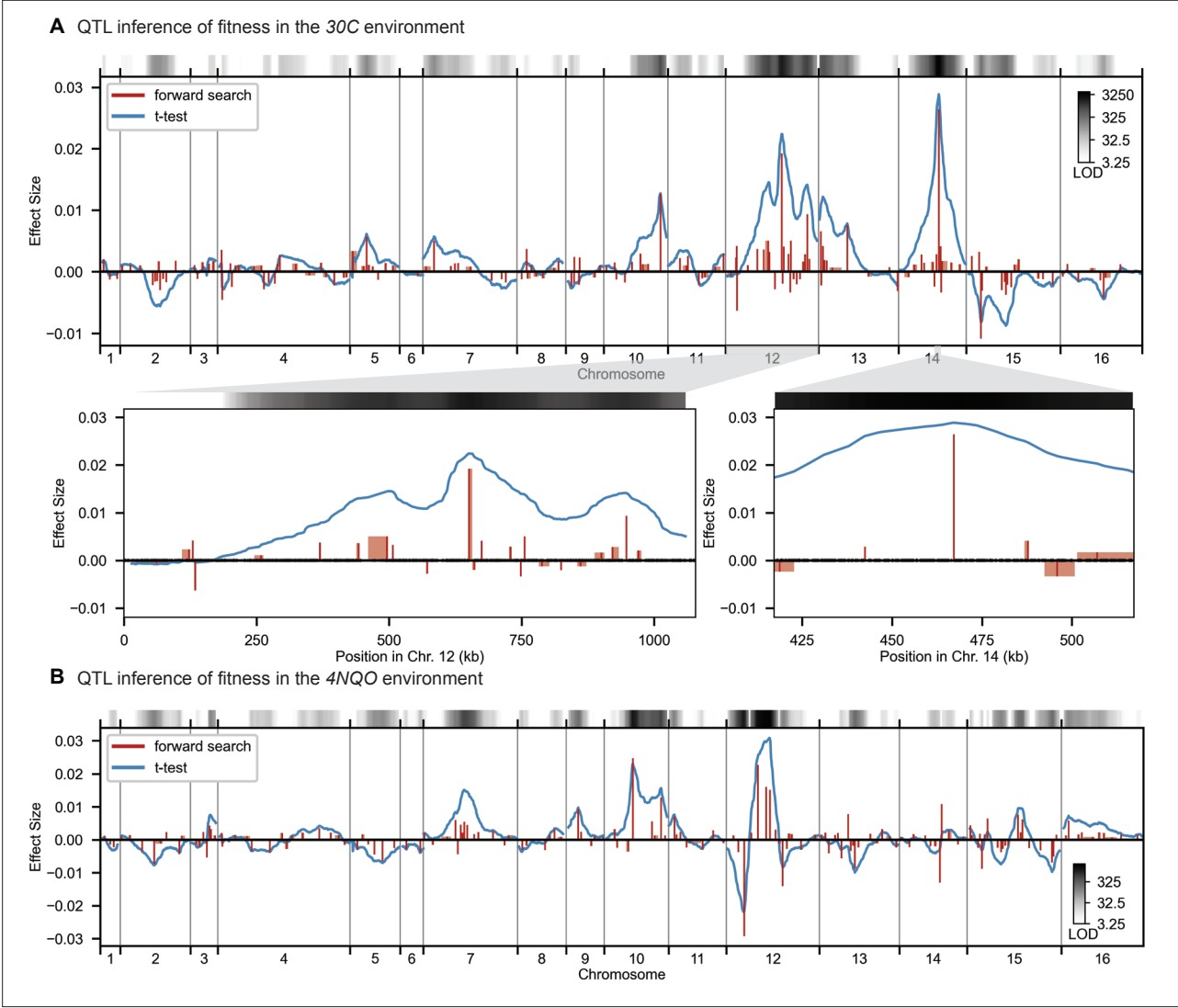

**Figure 2.** High-resolution QTL mapping. QTL mapping for (**A**) YPD at 30 °C and (**B**) SD with 4-nitroquinoline (4NQO). Inferred QTL are shown as red bars; bar height shows effect size and red shaded regions represent credible intervals. For contrast, effect sizes inferred by a Student's *t*-test at each locus are shown in blue. Gray bars at top indicate loci with log-odds (LOD) scores surpassing genome-wide significance in this *t*-test, with shading level corresponding to log-LOD score. See *Figure 2—figure supplements 1–4* for other environments. See *Supplementary file 2* for all inferred additive QTL models.

The online version of this article includes the following figure supplement(s) for figure 2:

**Figure supplement 1.** QTL mapping, as in *Figure 2*, for 23 C, 25 C, 27 C, and 33 C.

**Figure supplement 2.** QTL mapping, as in *Figure 2*, for 35 C, 37 C, Cu, and Eth.

**Figure supplement 3.** QTL mapping, as in *Figure 2*, for Gu, Li, Mann, and Mol.

**Figure supplement 4.** QTL mapping, as in *Figure 2*, for Raff, SDS, Suloc, YNB.

(*Peter et al., 2018*) compared to random SNPs (*Figure 3F*; *p* = 0.0004, Fisher's exact test considering alleles with effect >1% and rare alleles with MAF <5%) and minor alleles are more likely to be deleterious (*p* = 0.006, permutation test) regardless of which parental allele is rarer. These results are consistent with the view that rare, strongly deleterious alleles subject to negative selection can contribute substantially to complex trait architecture (*Bloom et al., 2019*; *Fournier et al., 2019*). However, we were unable to detect strong evidence of directional selection (see Appendix 3-1.7), possibly as a consequence of high polygenicity or weak concordance between our assay environments and selection pressure in the wild.

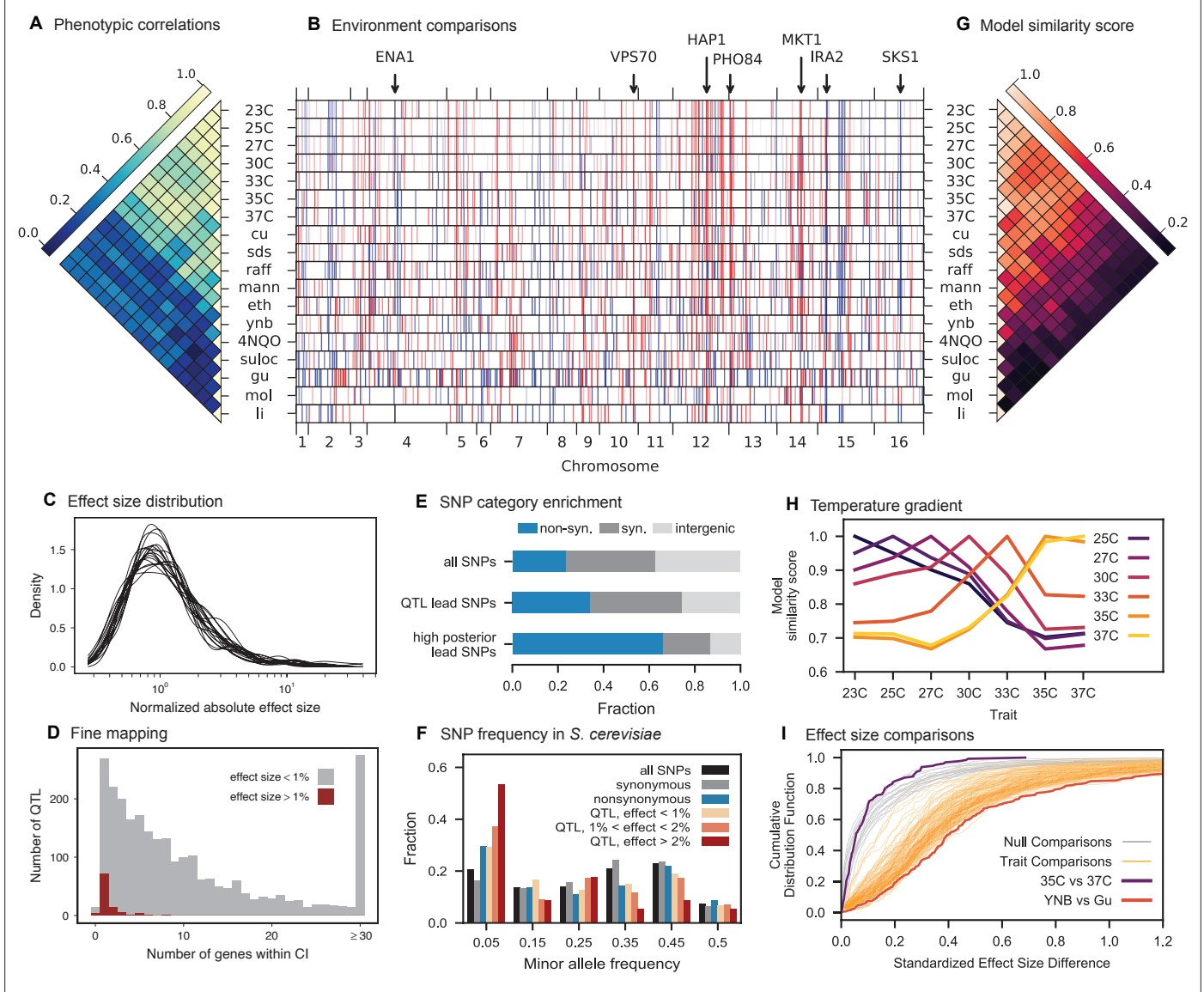

**Figure 3.** Genetic architecture and pleiotropy. (**A**) Pairwise Pearson correlations between phenotype measurements, ordered by hierarchical clustering (*Figure 3—source data 1*). (**B**) Inferred genetic architecture for each trait. Each inferred QTL is denoted by a red or blue line for a positive or negative effect of the RM allele, respectively; color intensity denotes effect size on a log scale. Notable genes are indicated above. See *Figure 3—figure supplement 1* for effect size comparison of the most pleiotropic genes. (**C**) Smoothed distribution of absolute effect sizes for each trait, normalized by the median effect for each trait. See *Figure 3—figure supplement 2* for a breakdown of the distributions by QTL effect sign. (**D**) Distribution of the number of genes within the 95% credible interval for each QTL (*Figure 3—source data 2*). (**E**) Distribution of SNP types. "High posterior" lead SNPs are those with >50% posterior probability. (**F**) Fractions of synonymous SNPs, nonsynonymous SNPs, and QTL lead SNPs as a function of their frequency in the 1011 Yeast Genomes panel (*Figure 3—source data 3*). (**G**) Pairwise model similarity scores (which quantify differences in QTL positions and effect sizes between traits; see Appendix 3) across traits. (**H**) Pairwise model similarity scores for each temperature trait against all other temperature traits (*Figure 3—source data 4*). See *Figure 3—figure supplement 3* for effect size comparisons between related environments. (**I**) Cumulative distribution functions (CDFs) of differences in effect size for each locus between each pair of traits (orange). Grey traces represent null expectations (differences between cross-validation sets for the same trait). The least and most similar trait pairs are highlighted in red and purple, respectively, and indicated in the legend.

The online version of this article includes the following source data and figure supplement(s) for figure 3:

**Source data 1.** Phenotypic correlation across environments.

**Source data 2.** Number of genes within confidence intervals of inferred QTL.

**Source data 3.** Frequency of lead SNPs in 1011 Yeast Genomes panel.

**Source data 4.** Pairwise model similarity scores across environments.

*Figure 3 continued on next page*

*Figure 3 continued*

**Figure supplement 1.** Highly pleiotropic genes.

**Figure supplement 2.** Comparison of positive and negative QTL effect signs.

**Figure supplement 3.** Correlations of pleiotropic effects across example traits.

## Patterns of pleiotropy

Our eighteen assay environments range widely in their similarity to each other: some groups of traits exhibit a high degree of phenotype correlation across strains, such as rich medium at a gradient of temperatures, while other groups of traits are almost completely uncorrelated, such as molasses, rich medium with suloctidil, and rich medium with guanidinium (*Figure 3A*). Because many of these phenotypes are likely to involve overlapping aspects of cellular function, we expect the inferred genetic architectures to exhibit substantial pleiotropy, where individual variants are causal for multiple traits. In addition, in highly polygenic architectures, pleiotropy across highly dissimilar traits is also expected to emerge due to properties of the interconnected cellular network. For example, SNPs in regulatory genes may affect key functional targets (some of them regulatory themselves) that directly influence a given phenotype, as well as other functional targets that may, in turn, influence other phenotypes (*Liu et al., 2019a*).

Consistent with these expectations, we observe diverse patterns of shared QTL across traits (*Figure 3B*). To examine these pleiotropic patterns at the gene level, we group QTL across traits whose lead SNP is within the same gene (or in the case of intergenic lead SNPs, the nearest gene). In total, we identify 449 such pleiotropic genes with lead SNPs affecting multiple traits (see Appendix 3). These genes encompass the majority of QTL across all phenotypes, and are highly enriched for regulatory function (*Table 2*) and for intrinsically disordered regions, which have been implicated in regulation ($p < 0.005$, Fisher's exact test; *Dyson and Wright, 2005*). The most pleiotropic genes (*Figure 3—figure supplement 1*) correspond to known variants frequently associated with quantitative variation in yeast (e.g. MKT1, HAP1, IRA2).

The highly polygenic nature of our phenotypes highlights the difficulty in identifying modules of target genes with interpretable functions related to the measured traits (*Boyle et al., 2017*). However, we can take advantage of our high-powered mapping approach to explore how pleiotropy leads to

**Table 2.** Summary of significant GO enrichment terms (*Table 2—source data 1*).

| Go id | Term | Corrected p-value | # Genes |
|---|---|---|---|
| GO:0000981 | RNA polymerase II transcription factor activity, sequence-specific DNA binding | 0.000026 | 30 |
| GO:0140110 | Transcription regulator activity | 0.000066 | 42 |
| GO:0000976 | Transcription regulatory region sequence-specific DNA binding | 0.000095 | 29 |
| GO:0044212 | Transcription regulatory region DNA binding | 0.000113 | 29 |
| GO:0003700 | DNA binding transcription factor activity | 0.000148 | 31 |
| GO:0001067 | Regulatory region nucleic acid binding | 0.000161 | 29 |
| GO:0043565 | Sequence-specific DNA binding | 0.000269 | 41 |
| GO:0001227 | Transcriptional repressor activity, RNA polymerase II transcription regulatory region sequence-specific DNA binding | 0.000417 | 9 |
| GO:0003677 | DNA binding | 0.000453 | 65 |
| GO:0043167 | Ion binding | 0.000999 | 143 |
| GO:1990837 | Sequence-specific double-stranded DNA binding | 0.001449 | 32 |
| GO:0000977 | RNA polymerase II regulatory region sequence-specific DNA binding | 0.007709 | 21 |
| GO:0001012 | RNA polymerase II regulatory region DNA binding | 0.007709 | 21 |

The online version of this article includes the following source data for table 2:

**Source data 1.** Full results of GO analysis on pleiotropic genes.

diverging phenotypes in different environments. Specifically, to obtain a global view of pleiotropy and characterize the shifting patterns of QTL effects across traits, we adopt a method inspired by sequence alignment strategies to match (or to leave unmatched) QTL from one trait with QTL from another trait, in a way that depends on the similarity in their effect sizes and distance between lead SNPs (see Appendix 3). From this, for each pair of environments we can find the change in effect size for each QTL, as well as an overall metric of model similarity (essentially a Pearson's correlation coefficient between aligned model parameters, with highest score of 1 meaning two identical models, and a score of 0 meaning no similar QTL detected in both position and effect size). We find that pairwise model similarity scores recapitulate the phenotype correlation structure (*Figure 3G*), including smoothly varying similarity across the temperature gradient (*Figure 3H*), indicating that changes in our inferred model coefficients accurately capture patterns of pleiotropy.

For most comparisons between environments, substantial effect size changes are distributed over all QTL, indicating a broad response to the environmental shift (*Figure 3I*). For example, while growth in Li (rich medium+ lithium) is strongly affected by a single locus directly involved in salt tolerance (three tandem repeats of the ENA cluster in S288C, corresponding to 82% of explained variance; *Wieland et al., 1995*), 63 of the remaining 82 QTL are also detected in 30 C (rich medium only), explaining a further 15% of variance. To some extent, these 63 QTL may represent a 'module' of genes with functional relevance for growth in rich medium, but their effect sizes are far less correlated than would be expected from noise or for a similar pair of environments (e.g. 30 C and 27 C, *Figure 3—figure supplement 3*). For the temperature gradient, while we observe high correlations between similar temperatures overall, these are not due to specific subsets of genes with simple, interpretable monotonic changes in effect size. Indeed, effect size differences between temperature pairs are typically uncorrelated; thus, QTL that were more beneficial when moving from 30C to 27C may become less beneficial when moving from 27C to 25C or 25C to 23C (*Figure 3—figure supplement 3*). Together, these patterns of pleiotropy reveal large numbers of regulatory variants with widespread, important, and yet somewhat unpredictable effects on diverse phenotypes, implicating a highly interconnected cellular network while obscuring potential signatures of specific functional genes or modules.

## Epistasis

To characterize the structure of this complex cellular network in more detail, numerous studies have used genetic perturbations to measure epistatic interactions between genes, which in turn shed light on functional interactions (*Tong et al., 2001*; *Horlbeck et al., 2018*; *Butland et al., 2008*; *Dixon et al., 2008*; *Costanzo et al., 2016*; *Costanzo et al., 2010*). However, the role of epistasis in GWAS and QTL mapping studies remains controversial; these studies largely focus on variance partitioning to measure the strength of epistasis, as they are underpowered to infer specific interaction terms (*Huang et al., 2016*). We sought to leverage the large sample size and high allele frequencies of our study to infer epistatic interactions, by extending our inference method to include potential pairwise interactions among the loci previously identified as having an additive effect (see Appendix 3). Our approach builds on the modified stepwise cross-validated search described above: after obtaining the additive model, we perform a similar iterative forward search on pairwise interactions, re-optimizing both additive and pairwise effect sizes at each step and applying a second L0-norm sparsity penalty, similarly chosen by cross-validation, to terminate the model search. We note that restricting our analysis of epistasis to loci identified as having an additive effect does not represent a major limitation. This is because a pair of loci that have a pairwise interaction but no additive effects will tend to be (incorrectly) assigned additive effects in our additive-only model, since the epistatic interaction will typically lead to background-averaged associations between each locus and the phenotype. These spurious additive effects will then tend to be reduced upon addition of the pairwise interaction term.

Using this approach, we detect widespread epistasis: hundreds of pairwise interactions for each phenotype (*Figure 4A and B*, *Table 1*; detected interactions are broadly consistent with results of a pairwise regression approach, as in *Figure 4—figure supplement 4*), which corresponds to an average of 1.7 epistatic interactions per QTL, substantially more than has been detected in previous mapping studies (*Bloom et al., 2015*). Most of these epistatic effects are modest, shifting predicted double mutant fitness values by a small amount in relation to additive predictions, although a small number largely exaggerate or completely suppress additive effects (*Figure 4—figure supplement 5A*). Overall, a slight majority of epistatic interactions are compensatory (*Figure 4—figure supplement*

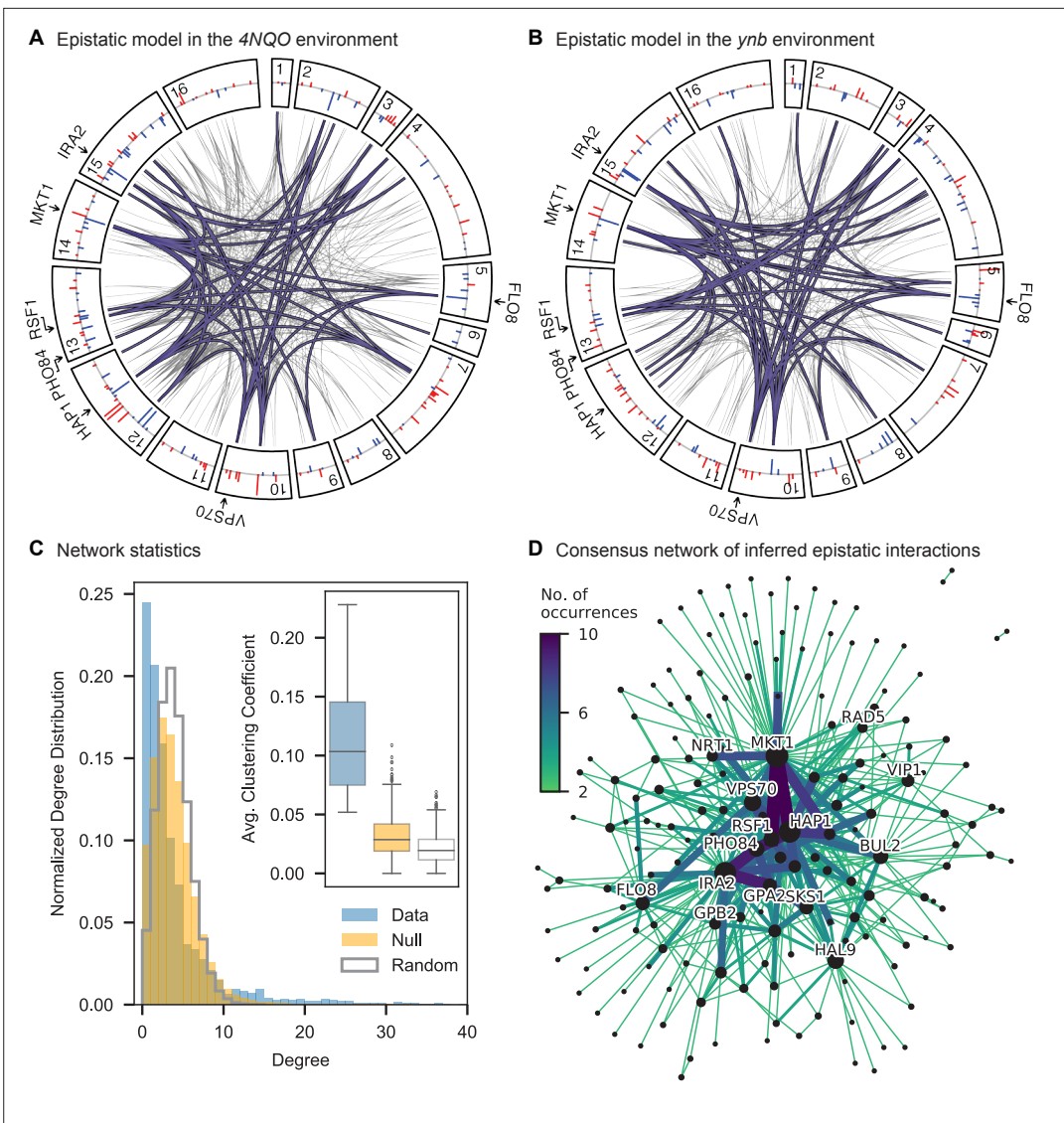

**Figure 4.** Pairwise epistasis. (**A, B**) Inferred pairwise epistatic interactions between QTL (with additive effects as shown in outer ring) for (**A**) the 4NQO environment and (**B**) the ynb environment. Interactions that are also observed for at least one other trait are highlighted in purple. See *Figure 4—figure supplements 1–3* for other environments, *Figure 4—figure supplement 4* for a simpler pairwise regression method, and *Figure 4—figure supplement 5* for a breakdown of epistatic effects and comparison to additive effects. (**C**) Network statistics across environments (*Figure 4—source data 1*). The pooled degree distribution for the eighteen phenotype networks is compared with 50 network realizations generated by an Erdos-Renyi random model (white) or an effect-size-correlation-preserving null model (orange; see Appendix 3). Inset: average clustering coefficient for the eighteen phenotypes, compared to 50 realizations of the null and random models. (**D**) Consensus network of inferred epistatic interactions. Nodes represent genes (with size scaled by degree) and edges represent interactions that were detected in more than one environment (with color and weight scaled by the number of occurrences). Notable genes are labeled. See *Figure 4—figure supplement 6* for the same consensus network restricted to either highly-correlated or uncorrelated traits.

The online version of this article includes the following source data and figure supplement(s) for figure 4:

**Source data 1.** Network statistics of observed and simulated epistatic networks.

**Figure supplement 1.** Epistatic interactions, as in *Figure 4A and B*, for 23 C, 25 C, 27 C, 30 C, 33 C, and 35 C.

**Figure supplement 2.** Epistatic interactions, as in *Figure 4A and B*, for 37 C, Cu, Eth, Gu, Li, and Mann.

**Figure supplement 3.** Epistatic interactions, as in *Figure 4A and B*, for Mol, Raff, SDS, Suloc, and YNB.

**Figure supplement 4.** Parwise regression in *4NQO*.

*Figure 4 continued on next page*

**Figure supplement 5.** Comparison between additive and epistatic effects.

**Figure supplement 6.** Consensus epistatic networks, as in *Figure 4D*.

*5B*), and nearly 55% of epistatic interactions have a positive sign (i.e. RM/RM and BY/BY genotypes are more fit than the additive expectation; *Figure 4—figure supplement 5C*). Finally, our procedure picks more epistatic interactions among intra- than between inter-chromosomal pairs of QTL (2.9% vs 2.3% among all environments; $\chi^2_{df=1} = 25.5$, $p < 10^{-6}$).

To interpret these epistatic interactions in the context of cellular networks, we can represent our model as a graph for each phenotype, where nodes represent genes with QTL lead SNPs and edges represent epistatic interactions between those QTL (this perspective is distinct from and complementary to *Costanzo et al., 2010*, where nodes represent gene deletions and edges represent similar patterns of interaction). Notably, in contrast to a random graph, the epistatic graphs across phenotypes show heavy-tailed degree distributions, high clustering coefficients, and small average shortest paths (~three steps between any pair of genes; *Figure 4C*); these features are characteristic of the small-world networks posited by the 'omnigenic' view of genetic architecture (*Boyle et al., 2017*). These results hold even when accounting for ascertainment bias (i.e. loci with large additive effects have more detected epistatic interactions; see Appendix 3).

We also find that hundreds of epistatic interactions are repeatedly found across environments (*Figure 4D and Figure 4—figure supplement 6*). Overall, epistatic interactions are more likely to be detected in multiple environments than expected by chance, even when considering only uncorrelated environments (simulation-based $p < 10^{-3}$; see Appendix 3), as expected if these interactions accurately represent the underlying regulatory network. Considering interactions found in all environments, we see a small but significant overlap of detected interactions with previous genome-wide deletion screens (*Costanzo et al., 2016*; $p = 0.03$, $\chi^2 = 4.46$, df = 1; see Appendix 3). Taken together, these results suggest that inference of epistatic interactions in a sufficiently high-powered QTL mapping study provides a consistent and complementary method to reveal both global properties and specific features of underlying functional networks.

## Validating QTL inferences with reconstructions

We next sought to experimentally validate the specific inferred QTL and their effect sizes from our additive and additive-plus-pairwise models. To do so, we reconstructed six individual and nine pairs of RM SNPs on the BY background and measured their competitive fitness in 11 of the original 18 conditions in individual competition assays (although note that for technical reasons these measurement conditions are not precisely identical to those used in the original bulk fitness assays; see Materials and methods). These mutations were chosen because they represent a mixture of known and unknown causal SNPs in known QTL, were amenable to the technical details of our reconstruction methods, and had a predicted effect size larger than the detection limit of our low-throughput assays to measure competitive fitness (approximately 0.5%) in at least one environment. We find that the QTL effects inferred with the additive-only models are correlated with the phenotypes of these reconstructed genotypes, although the predicted effects are systematically larger than the measured phenotypes (*Figure 5A*, cyan). To some extent, these errors may arise from differences in measurement conditions, undetected smaller-effect linked loci that bias inferred additive effect sizes, and from the confidence intervals around the lead SNP, which introduce uncertainty about the identity of the precise causal locus, among other factors. However, this limited power is also somewhat unsurprising even if our inferred lead SNPs are correct, because the effect sizes inferred from the additive-only model measure the effect of a given locus averaged across the F1 genetic backgrounds in our segregant panel. Thus, if there is significant epistasis, we expect the effect of these loci in the specific strain background chosen for the reconstruction (the BY parent in this case) to differ from the background-averaged effect inferred by BB-QTL.

In agreement with this interpretation, we find that the predictions from our additive-plus-pairwise inference agree better with the measured values in our reconstructed mutants (*Figure 5A*, magenta). Specifically, we find that the correlation between predicted and measured phenotypes is similar to the additive-only model, but the systematic overestimates of effect sizes are significantly reduced (*Figure 5A*, inset; $p < 10^{-4}$ from permutation test; see Materials and Methods). This suggests a

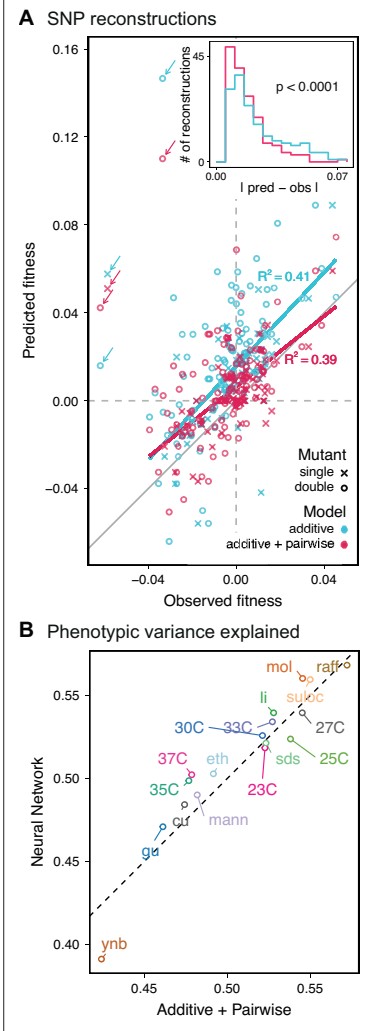

**Figure 5.** Evaluating model performance. (**A**) Comparison between the measured fitness of reconstructions of 6 single (crosses) and nine double mutants (circles) in 11 environments, and their fitness in those environments as predicted by our inferred additive-only (cyan) or additive-plus-pairwise-epistasis models (magenta). The one-to-one line is shown in gray. $R^2$ values correspond to to shown fitted linear regressions for each type of model (colored lines), excluding MKT1 mutants measured in gu environment (outliers indicated by arrows). Inset shows the histogram of the absolute difference between observed and predicted reconstruction fitness under our two models, with the $p$-value from the permutation test of the difference between these distributions indicated. See **Figure 5—figure supplements 1 and 2** for a full breakdown of the data, and **Figure 5—source data 1** for measured and predicted fitness values. (**B**) Comparison between estimated phenotypic variance explained by the additive-plus-pairwise-epistasis model and a trained dense neural network of optimized architecture.

The online version of this article includes the following

*Figure 5 continued on next page*

*Figure 5 continued*

source data and figure supplement(s) for figure 5:

**Figure supplement 1.** Comparison between measured and predicted fitness of reconstructions, as in *Figure 5*, broken down by mutation (single or double) and model type (additive-only or additive-plus-pairwise).

**Figure supplement 2.** Comparison between observed and predicted reconstruction strain fitness, as in the inset of *Figure 5A*.

**Source data 1.** SNP reconstructions' fitness measurements and predictions.

---

substantial effect of nonlinear terms, although the predictive power of our additive-plus-pairwise model remains modest. As above, this limited predictive power could be a consequence of undetected linked loci or errors in the identification of interacting loci. However, it may also indicate the presence of further epistasis of higher order than the pairwise terms we infer. To explore these potential effects of higher-order interactions, we trained a dense neural network to jointly predict 17 out of our 18 phenotypes from genotype data (see Appendix 3). The network architecture involves three densely connected layers, allowing it to capture arbitrary nonlinear mappings. Indeed, we find that this neural network approach does explain slightly more phenotypic variance (on average 1% more variance than the additive-plus-pairwise QTL model, *Figure 5B*; see Appendix 4), although specific interactions and causal SNPs are harder to interpret in this case. Together, these results suggest that although our ability to pinpoint precise causal loci and their effect sizes is likely limited by a variety of factors, the models with epistasis do more closely approach the correct genetic architecture despite explaining only marginally more variance than the additive model (*Appendix 4—figure 1*), as suggested by previous studies (*Forsberg et al., 2017*).

## Discussion

The BB-QTL mapping approach we have introduced in this study increases the scale at which QTL mapping can be performed in budding yeast, primarily by taking advantage of automated liquid handling techniques and barcoded phenotyping. While the initial construction of our segregant panel involved substantial brute force, this has now generated a resource that can be easily shared (particularly in pooled form) and used for similar studies aiming to investigate a variety of related questions in quantitative genetics. In

addition, the approaches we have developed here provide a template for the systematic construction of additional mapping panels in future work, which would offer the opportunity to survey the properties of a broader range of natural variation. While our methods are largely specific to budding yeast, conceptually similar high-throughput automated handling systems and barcoding methods may also offer promise in studying quantitative genetics in other model organisms, though substantial effort would be required to develop appropriate techniques in each specific system.

Here, we have used our large segregant panel to investigate the genetic basis of 18 phenotypes, defined as competitive fitness in a variety of different liquid media. The increased power of our study reveals that these traits are all highly polygenic: using a conservative cross-validation method, we find more than 100 QTL at high precision and low false-positive rate for almost every environment in a single F1 cross. Our detected QTL include many of the key genes identified in earlier studies, along with many novel loci. These QTL overall are consistent with statistical features observed in previous studies. For example, we find an enrichment in nonsynonymous variants among inferred causal loci in regulatory genes, and a tendency for rare variants (as defined by their frequency in the 1,011 Yeast Genomes collection; *Peter et al., 2018*) to have larger effect sizes.

While the QTL we detect do explain most of the narrow-sense heritability across all traits (*Appendix 4—figure 1*), this does not represent a substantial increase relative to the heritability explained by earlier, smaller studies with far fewer QTL detected (*Bloom et al., 2013*; *Bloom et al., 2015*; *Bloom et al., 2019*). Instead, the increased power of our approach allows us to dissect multiple causal loci within broad regions previously detected as a single 'composite' QTL (*Figure 2A*, zoom-ins), and to detect numerous novel small-effect QTL. Thus, our results suggest that, despite their success in explaining observed phenotypic heritability, these earlier QTL mapping studies in budding yeast fail to accurately resolve the highly polygenic nature of these phenotypes. This, in turn, implies that the apparent discrepancy in the extent of polygenicity inferred by GWAS compared to QTL mapping studies in model organisms arises at least in part as an artifact of the limited sample sizes and power of the latter.

Our finding that increasing power reveals increasingly polygenic architectures of complex traits is broadly consistent with several other recent studies that have improved the scale and power of QTL mapping in yeast in different ways. For example, advanced crosses have helped to resolve composite QTL into multiple causal loci (*She and Jarosz, 2018*), and multiparental or round-robin crosses have identified numerous additional causal loci by more broadly surveying natural variation in yeast (*Bloom et al., 2019*; *Cubillos et al., 2013*). In addition, recent work has used very large panels of diploid genotypes to infer highly polygenic trait architectures, though this study involves a much more permissive approach to identifying QTL that may lead to a substantial false positive rate (*Jakobson and Jarosz, 2019*). Here, we have shown that by simply increasing sample size we can both resolve composite QTL into multiple causal loci and also identify numerous additional small-effect loci that previous studies have been underpowered to detect. The distribution of QTL effect sizes we infer is consistent with an exponential distribution up to our limit of detection, suggesting that there may be many more even smaller-effect causal loci that could be revealed by further increases in sample size.

By applying BB-QTL mapping to eighteen different fitness traits, we explored how the effects of individual loci shift across small and large environmental perturbations. Quantifying the structure of these pleiotropic effects is technically challenging, particularly for many QTL of modest effect that are not resolved to a single SNP or gene. In these cases, it is difficult to determine whether a particular region contains a single truly pleiotropic locus, or multiple linked variants that each influence a different trait. While we have used one particular approach to this problem, other choices are also possible, and ideally future work to jointly infer QTL using data from all phenotypes simultaneously could provide a more rigorous method for identifying pleiotropic loci. However, we do find that the structure of the pleiotropy in our inferred models largely recapitulates the observed correlations between phenotypes, suggesting that the causal loci we identify are largely sufficient to explain these patterns. Many of the same genes are implicated across many traits, often with similar strong effect sizes in distinct environments, and as we might expect these highly pleiotropic QTL are enriched for regulatory function. However, dividing QTL into modules that affect subsets of environments, predicting how their effect sizes change across environments (even our temperature gradient), and identifying core or peripheral genes (as in *Boyle et al., 2017*) remains difficult. Future work to assay a larger number and wider range of phenotypes could potentially provide

more detailed insight into the structure of relationships between traits and how they arise from shared genetic architectures.

We also leveraged the statistical power of our approach to explore the role of epistatic interactions between QTL. Previous studies have addressed this question through the lens of variance partitioning, concluding that epistasis contributes less than additive effects to predicting phenotype (*Bloom et al., 2015*). However, it is a well-known statistical phenomenon that variance partitioning alone cannot determine the relative importance of additive, epistatic, or dominance factors in gene action or genetic architectures (*Huang et al., 2016*). Here, we instead explore the role of epistasis by inferring specific pairwise interaction terms and analyzing their statistical and functional patterns. We find that epistasis is widespread, with nearly twice as many interaction terms as additive QTL. The resulting interaction graphs show characteristic features of biological networks, including heavy-tailed degree distributions and small shortest paths, and we see a significant overlap with interaction network maps from previous studies despite the different sources of variation (naturally occurring SNPs versus whole-gene deletions). Notably, the set of genes with the most numerous interactions overlaps with the set of highly pleiotropic genes, which are themselves enriched for regulatory function. Together, these findings indicate that we are capturing features of underlying regulatory and functional networks, although we are far from revealing the complete picture. In particular, we expect that we fail to detect many interactions that have small effect sizes below our detection limit, that the interactions we observe are limited by our choice of phenotypes, and that higher-order interactions may also be widespread.

To validate our QTL inference, we reconstructed a small set of single and double mutations by introducing RM alleles into the BY parental background. We find that our ability to predict the effects of these putatively causal loci remains somewhat limited: the inferred effect sizes in our additive plus pairwise epistasis models have relatively modest power to predict the fitness effects of reconstructed mutations and pairs of mutations. Thus, despite the unprecedented scale and power of our study, we still cannot claim to precisely infer the true genetic architecture of complex traits. This failure presumably arises in part from limitations to our inference, which could lead to inaccuracies in effect sizes or the precise locations of causal loci. In addition, the presence of higher order epistatic interactions (or interactions with the mitochondria) would imply that we cannot expect to accurately predict phenotypes for genotypes far outside of our F1 segregant panel, such as single- and double-SNP reconstructions on the BY genetic background. While both of these sources of error could in principle be reduced by further increases in sample size and power, it is unlikely that substantial improvements are likely to be realized in the near future.

However, despite these limitations, our BB-QTL mapping approach helps bridge the gap between well-controlled laboratory studies and high-powered, large-scale GWAS, revealing that complex trait architecture in model organisms is indeed influenced by large numbers of previously unobserved small-effect variants. We examined in detail how this architecture shifts across a spectrum of related traits, observing that while pleiotropy is common, changes in effects are largely unpredictable, even for similar traits. Further, we characterized specific epistatic interactions across traits, revealing not only their substantial contribution to phenotype but also the underlying network structure, in which a subset of genes occupy central roles. Future work in this and related systems is needed to illuminate the landscape of pleiotropy and epistasis more broadly, which will be crucial not merely for phenotype prediction but for our fundamental understanding of cellular organization and function.

## Materials and methods

**Key resources table**

| Reagent type (species) or resource | Designation | Source or reference | Identifiers | Additional information |
|---|---|---|---|---|
| Strain, strain background (*S. cerevisiae*) | BY4741 | *Baker Brachmann et al., 1998* | – | BY parent background |
| Strain, strain background (*S. cerevisiae*) | RM11-1a | *Brem et al., 2002* | – | RM parent background |

*Continued on next page*

*Continued*

| Reagent type (species) or resource | Designation | Source or reference | Identifiers | Additional information |
|---|---|---|---|---|
| Strain, strain background (*S. cerevisiae*) | YAN696 | This paper | – | BY parent |
| Strain, strain background (*S. cerevisiae*) | YAN695 | This paper | – | RM parent |
| Strain, strain background (*S. cerevisiae*) | BB-QTL F1 strain library | This paper | – | 100,000 strains of barcoded BYxRM F1 produced, genotyped and characterized in this paper |
| Strain, strain background (*E. coli*) | BL21(DE3) | NEB | NEB:C2527I | Zymolyase and barcoded Tn-5 expression system |
| Recombinant DNA reagent | pAN216a pGAL1-HO pSTE2-HIS3 pSTE3 LEU2 (plasmid) | This paper | – | For MAT-type switching |
| Recombinant DNA reagent | pAN3H5a 1/2URA3 KanP1 ccdB LoxPR (plasmid) | This paper | – | Type one barcoding plasmid, without barcode library |
| Recombinant DNA reagent | pAN3H5a LoxPL ccdB HygP1 2/2URA3 (plasmid) | This paper | – | Type two barcoding plasmid, without barcode library |
| Sequence-based reagent | Custom Tn5 adapter oligos | This paper | Tn5-L and -R | See Materials and methods |
| Sequence-based reagent | Custom sequencing adapter primers | This paper | P5mod and P7mod | See Materials and methods |
| Sequence-based reagent | Custom barcode amplification primers | This paper | P1 and P2 | See Materials and methods |
| Sequence-based reagent | Custom sequencing primers for Illumina | This paper | Custom_read_1 through 4 | See Materials and methods |
| Software, algorithm | Custom code for genotype inference | This paper | – | See code repository |
| Software, algorithm | Custom code for phenotype inference | This paper | – | See code repository |
| Software, algorithm | Custom code for compressed sensing | This paper | – | See code repository |
| Software, algorithm | Custom code for qtl inference | This paper | – | See code repository |

## Design and construction of the barcoded cross

A key design choice in any QTL mapping study is what genetic variation will be included (i.e. the choice of parental strains) and how that variation will be arranged (i.e. the cross design). In choosing strains, wild isolates may display more genotypic and phenotypic diversity than lab-adapted strains, but may also have poor efficiency during the mating, sporulation, and transformation steps required for barcoding. In designing the cross, one must balance several competing constraints, prioritizing different factors according to the goals of the particular study. Increasing the number of parental strains involved increases genetic diversity, but can also increase genotyping costs (see Appendix 1); increasing the number of crosses or segregants improves fine-mapping resolution by increasing the number of recombination events, but also increases strain production time and costs. For a study primarily interested in precise fine-mapping of strong effects, a modest number of segregants from a deep cross may be most appropriate *She and Jarosz, 2018*; for a study interested in a broad view of diversity across natural isolates, a modest number of segregants produced from a shallow multi-parent cross may be preferred (*Bloom et al., 2019*). However, we are primarily interested in achieving maximum statistical power to map large numbers of small-effect loci; for this, we choose to map an extremely large number of segregants from a simple, shallow cross. We begin in this section by describing our choice of strains, cross design, and barcoding strategy.

## Strains

The two parental strains used in this study, YAN696 and YAN695, are derived from the BY4741 (*Baker Brachmann et al., 1998*) (S288C: MATa, his3Δ1, ura3Δ0, leu2Δ0, met17Δ0) and RM11-1a (haploid derived from Robert Mortimer's Bb32(3): MATa, ura3Δ0, leu2Δ0, ho::KanMX; *Brem et al., 2002*) backgrounds respectively, with several modifications required for our barcoding strategy. We chose to work in these backgrounds (later denoted as BY and RM) due to their history in QTL mapping studies in yeast (*Ehrenreich et al., 2010*; *Bloom et al., 2013*; *Brem et al., 2002*), which demonstrate a rich phenotypic diversity in many environments.

The ancestral RM11-1a is known to incompletely divide due to the AMN1(A1103) allele (*Yvert et al., 2003*). To avoid this, we introduced the A1103T allele by Delitto Perfetto (*Storici et al., 2001*), yielding YAN497. This strain also contains an HO deletion that is incompatible with HO-targeting plasmids (*Voth et al., 2001*). We replaced this deletion by introducing the S288C version of HO with the NAT marker (resistance to nourseothricin; *Goldstein and McCusker, 1999*) driven by the TEF promoter from *Lachancea waltii*, and terminated with tSynth7 (*Curran et al., 2015*) (ho::LwpTEF-Nat-tSynth7), creating YAN503. In parallel, a MATα strain was created by converting the mating type of the ancestral RM11-1a using a centromeric plasmid carrying a galactose-inducible functional HO endonuclease (pAN216a_pGAL1-HO_pSTE2-HIS3_pSTE3_LEU2), creating YAN494. In this MATα strain, the HIS3 gene was knocked out with the his3d1 allele of BY4741, generating YAN501. This strain was crossed with YAN503 to generate a diploid (YAN501xYAN503) and sporulated to obtain a spore containing the desired set of markers (YAN515: MATα, ura3Δ0, leu2Δ0, his3Δ1, AMN1(A1103T), ho::LwpTEF-NAT-tSynth7). A 'landing pad' for later barcode insertion was introduced into this strain by swapping the NAT marker with pGAL1-Cre-CaURA3MX creating YAN616. This landing pad contains the Cre recombinase under the control of the galactose promoter and the URA3 gene from *Candida albicans* driven by the TEF promoter from *Ashbya gossypii*. This landing pad can be targeted by HO-targeting plasmids containing barcodes (see *Figure 1—figure supplement 1*). To allow future selection for diploids, the G418 resistance marker was introduced between the Cre gene and the CaURA3 gene, yielding the final RM11-1a strain YAN695.

The BY4741 strain was first converted to methionine prototrophy by transforming the *MET17* gene into *Δmet17*, generating YAN599. The CAN1 gene was subsequently replaced with a MATa mating type reporter construct (*Tong and Boone, 2006*) (pSTE2-SpHIS5) which expresses the HIS5 gene from *Schizosaccharomyces pombe* (orthologous to the *S. cerevisiae* HIS3) in MATa cells. A landing pad was introduced into this strain with CaURA3MX-NAT-pGAL1-Cre, yielding YAN696; this landing pad differs from the one in YAN695 by the location of the pGAL1-Cre and the presence of a NAT marker rather than a G418 marker (see *Figure 1—figure supplement 2*).

## Construction of barcoding plasmids

The central consideration in designing our barcoding strategy was ensuring sufficient diversity, such that no two out of 100,000 individual cells sorted from a pool will share the same barcode. To ensure this, the total number of barcodes in the pool must be several orders of magnitude larger than $10^5$, which is infeasible even with maximal transformation efficiency. Our solution employs combinatorics: each spore receives two barcodes, one from each parent, such that mating two parental pools of ~$10^4$ unique barcodes produces a pool of ~$10^8$ uniquely barcoded spores. This requires efficient barcoding of the parental strains, efficient recombination of the two barcodes onto the same chromosome in the mated diploids, and efficient selection of double-barcoded haploids after sporulation.

Our barcoding system makes use of two types of barcoding plasmids, which we refer to as Type 1 and Type 2. Both types are based on the HO-targeting plasmid pAN3H5a (*Figure 1—figure supplement 1*), which contains the LEU2 gene for selection as well as homologous sequences that lead to integration of insert sequences with high efficiency. The Type 1 barcoding plasmid has the configuration pAN3H5a–1/2URA3–KanP1–ccdB–LoxPR, while Type 2 has the configuration pAN3H5a–LoxPL–ccdB–HygP1–2/2URA3. Here the selection marker URA3 is split across the two different plasmids: 1/2URA3 contains the first half of the URA3 coding sequence and an intron 5' splice donor, while 2/2URA3 contains the 3' splice acceptor site and the second half of the URA3 coding sequence. Thus we create an artificial intron within URA3, using the 5' splice donor from RPS17a and the 3' splice acceptor from RPS6a (*Plass, 2020*), that will eventually contain both barcodes. KanP1 and HygP1 represent primer sequences. The ccdB gene provides counter-selection for plasmids lacking barcodes

(see below). Finally, LoxPR and LoxPL are sites in the Cre-Lox recombination system (*Albert et al., 1995*) used for recombining the two barcodes in the mated diploids. These sites contain arm mutations on the right and left side, respectively, that render them partially functional before recombination; after recombination, the site located between the barcodes is fully functional, while the one on the opposite chromosome has slightly reduced recombination efficiency (*Nguyen Ba et al., 2019*).

We next introduced a diverse barcode library into the barcoding plasmids using a Golden Gate reaction (*Engler, 2009*), exploiting the fact that the ccdB gene (with BsaI site removed) is flanked by two BsaI restriction endonuclease sites. Barcodes were ordered as single-stranded oligos from Integrated DNA Technologies (IDT) with degenerate bases (ordered as 'hand-mixed') flanked by priming sequences: P_BC1 = CTAGTT ATTGCT CAGCGG AGGTCT CAtact NNNatN NNNNat NNNNNg cNNNcg ctAGAG ACCGTC ATAGCT GTTTCC TG, P_BC2 = CTAGTT ATTGCT CAGCGG AGGTCT CAtact NNNatN NNNNgc NNNNNa tNNNcg ctAGAG ACCGTC ATAGCT GTTTCC TG. These differ only in nucleotides inserted at the 16 degenerate bases (represented by 'N'), which comprise the barcodes; these barcodes are flanked by two BsaI sites that leave 'tact' and 'cgct' as overhangs. Conversion of the oligos to double-stranded DNA and preparation of the *Escherichia coli* library was performed as in *Nguyen Ba et al., 2019*. Briefly, Golden Gate reactions were carried out to insert P_BC1 and P_BC2 barcodes into Type 1 and Type 2 barcoding plasmids, respectively. Reactions were purified and electroporated into *E. coli* DH10β cells (Invitrogen) and recovered in about 1 L of molten LB (1% tryptone, 0.5% yeast extract) containing 0.3% SeaPrep agarose (Lonza) and 100 μg/mL ampicillin. The mixture was allowed to gel on ice for an hour and then placed at 37 °C overnight for recovery of the transformants. We routinely observed over $10^6$ transformants from this procedure, and the two barcoded plasmid libraries were recovered by standard maxiprep (*Figure 1—figure supplement 2a*).

## Parental barcoding procedure

Distinct libraries of barcoded parental BY and RM strains were generated by transforming parental strains with barcoded plasmid libraries (*Figure 1—figure supplement 2b*): YAN696 (BY) with Type 1, and YAN695 (RM) with Type 2. Barcoded plasmids were digested with PmeI (New England Biolabs) prior to transformation, and transformants were selected on several SD –leu agar plates (6.71 g/L yeast nitrogen base, complete amino acid supplements lacking leucine, 2% dextrose). In total, 23 pools of ~1000 transformants each were obtained for BY, and 12 pools of ~1,000 transformants each were obtained for RM. After scraping the plates, we grew these pools overnight in SD –leu +5 FOA (6.71 g/L yeast nitrogen base, complete amino acid supplements lacking leucine, 2% dextrose, 1 g/L 5-fluoroorotic acid) to select for integration at the correct locus. Each of the BY and RM libraries were kept separate, and for each library we sequenced the barcode locus on an Illumina NextSeq to confirm the diversity and identity of parental barcodes. This allows for 23 × 12 distinct sets of diploid pools with approximately 1 million possible barcode combinations each, for a total of ~$3 × 10^8$ barcodes.

## Generation of 100,000 barcoded F1 segregants

Each of the 23 × 12 combinations of BY and RM barcoded libraries was processed separately, from mating through sorting of the resulting F1 haploids (*Figure 1—figure supplement 2b*). For each distinct mating, the two parental libraries were first grown separately in 5 mL YPD (1% Difco yeast extract, 2% Bacto peptone, 2% dextrose) at 30 °C. The overnight cultures were plated on a single YPD agar plate and allowed to mate overnight at room temperature. The following day, the diploids were scraped and inoculated into 5 mL YPG (1% Difco yeast extract, 2% Bacto peptone, 2% galactose) containing 200 μg/mL G418 (GoldBio) and 10 μg/mL Nourseothricin (GoldBio) at a density of approximately $2 \cdot 10^6$ cells/mL and allowed to grow for 24 hr at 30 °C. The next day, the cells were again diluted 1:$2^5$ into the same media and allowed to grow for 24 hr at 30 °C. This results in ~10 generations of growth in galactose-containing media, which serves to induce Cre recombinase to recombine the barcoded landing pads, generating Ura+ cells (which therefore contain both barcodes on the same chromosome) at very high efficiency.

Recombinants were selected by 10 generations of growth (two cycles of 1:$2^5$ dilution and 24 hr of growth at 30 °C) in SD –ura (6.71 g/L Yeast Nitrogen Base with complete amino acid supplements lacking uracil, 2% dextrose). Cells were then diluted 1:$2^5$ into pre-sporulation media (1% Difco yeast extract, 2% Bacto peptone, 1% potassium acetate, 0.005% zinc acetate) and grown for 24 hr at room

temperature. The next day, the whole culture was pelleted and resuspended into 5 mL sporulation media (1% potassium acetate, 0.005% zinc acetate) and grown for 72 hr at room temperature. Sporulation efficiency typically reached >75%.

Cells were germinated by pelleting approximately 100 µL of spores and digesting their asci with 50 µL of 5 mg/mL Zymolyase 20T (Zymo Research) containing 20 mM DTT for 15 min at 37 °C. The tetrads were subsequently disrupted by mild sonication (3 cycles of 30 s at 70% power), and the dispersed spores were recovered in 100 mL of molten SD –leu –ura –his + canavanine (6.71 g/L Yeast Nitrogen Base, complete amino acid supplements lacking uracil, leucine and histidine, 50 µg/mL canavanine, 2% dextrose) containing 0.3% SeaPrep agarose (Lonza) spread into a thin layer (about 1 cm deep). The mixture was allowed to set on ice for an hour, after which it was kept for 48 hr at 30 °C to allow for dispersed growth of colonies in 3D. This procedure routinely resulted in ~$10^6$ colonies of uniquely double-barcoded MATa F1 haploid segregants for each pair of parental libraries.

F1 segregants obtained using our approach are expected to be prototrophic haploid MATa cells, with approximately half their genome derived from the BY parent and half from the RM parent, except at marker loci (see *Figure 1—figure supplement 3*).

## Sorting F1 segregants into single wells

After growth, germinated cells in soft agar were mixed by thorough shaking. Fifty µL of this suspension was inoculated in 5 mL SD –leu –ura –his + canavanine and allowed to grow for 2–4 hr at 30 °C. The cells were then stained with DiOC6(3) (Sigma) at a final concentration of 0.1 µM and incubated for 1 hr at room temperature. DiOC6(3) is a green-fluorescent lipophilic dye that stains for functional mitochondria (*Petit et al., 1996*). We used flow cytometry to sort single cells into individual wells of polypropylene 96-well round-bottom microtiter plates containing 150 µL YPD per well. During sorting we applied gates for front and side scatter to select for dispersed cells as well as a FITC gate for functional mitochondria. From each mating, we sorted 384 single cells into 4 96-well plates; because the possible diversity in each mating is about $10^6$ unique barcodes, the probability that two F1 segregants share the same barcode is extremely low.

The single cells were then allowed to grow unshaken for 48 hr at 30 °C, forming a single colony at the bottom of each well. The plates were then scanned on a flatbed scanner and the scan images were used to identify blank wells. We observed 4843 blanks among a total of 105,984 wells (sorting efficiency of ≥95%), resulting in a total of 101,141 F1 segregants in our panel. Plates were then used for archiving, genotyping, barcode association, and phenotyping as described below.

We expect F1 segregants obtained using our approach to be prototrophic haploid MATa cells, with approximately half their genome derived from the BY parent and half from the RM parent, except at marker loci in chromosome III, IV, and V (see *Figure 1—figure supplement 3* for allele frequencies).

## Reference parental strains

Although not strictly necessary for future analysis, it is often informative to phenotype the parental strains in the same assay as the F1 segregants. However, the parental strains described above (YAN695 and YAN696) do not contain the full recombined barcode locus and are thus not compatible with bulk-phenotyping strategies described in Appendix 2. We therefore created reference strains whose barcode locus and selection markers are identical to the F1 segregants, while their remaining genotypes are identical to the parental strains.

The parental BY strain YAN696 required little modification: we simply transformed its landing pad with a known double barcode (barcode sequence ATTTGACCCAAAGCTT – GGCATGGCGCCGTACG ). In contrast, the parental RM strain is of the opposite mating type to the F1 segregants and differs at the CAN1 locus. From the YAN501xYAN503 diploid, we generated a MATa spore containing otherwise identical genotype as YAN515, producing YAN516. The CAN1 locus of this strain was replaced with the MATa mating type reporter as described previously (pSTE2-SpHIS5) producing YAN684. Finally, we transformed the landing pad with a known double barcode (barcode sequence AGAAGAAGTCAC CGTA – TACTACGTCTTATTTA). We refer to these strains as RMref and BYref in Appendix 2.

## Whole-genome sequencing of all F1 segregants

The SNP density in the BYxRM cross is such that there is on average one SNP every few hundred basepairs (i.e. on the order of one SNP per sequencing read), making whole-genome sequencing an

attractive genotyping strategy. However, there are obvious time and cost constraints to preparing 100,000 sequencing libraries and sequencing 100,000 whole genomes to high coverage. We overcome these challenges by creating a highly automated and multiplexed library preparation pipeline, and by utilizing low-coverage sequencing in combination with genotype imputation methods. In this section we describe the general procedure for producing sequencing libraries, while sections below describe how we automate the procedure in 96-well format using liquid handling robots.

## Cell lysis and zymolyase purification

Typical tagmentase-based library preparation protocols start with purified genomic DNA of normalized concentration (**Baym et al., 2015**). However, we found that the transposition reaction is efficient in crude yeast lysate, provided that the yeast cells are lysed appropriately. This approach significantly reduces costs and increases throughput of library preparation.

Efficient cell lysis can be accomplished with the use of zymolyase, a mixture of lytic enzymes that can digest the yeast cell wall in the presence of mild detergents. However, we and others (**Miyajima et al., 2009**) have found that commercial preparations of zymolyase from *Arthrobacter luteus* contain contaminating yeast DNA. The amount of contaminating DNA in the preparations is low (estimates of $10^7$ copies of the rRNA gene per mL of extraction buffer can be found **Miyajima et al., 2009**), but this level of contamination is problematic for low-coverage sequencing of yeast strains, as it complicates later genotype imputation (see Appendix 1). Attempts to reduce yeast DNA contamination from either commercial or in-house preparation from *A. luteus* were unsuccessful. Thus, we take a different approach by recombinantly producing large quantities of beta-1,3-glucanase (**Rodríguez-Peña et al., 2013**), the major lytic component of zymolyase, in *E. coli*. The protease that is also required for natural zymolyase activity can be omitted if the reaction is supplemented with DTT (**Scott and Schekman, 1980**).

We produce two isoforms of beta-1,3-glucanase. One is an N-terminal 6xHIS-tag version of the *A. luteus* enzyme, where the nickel-binding tag is inserted after a consensus periplasmic signal sequence (**Heggeset et al., 2013**). The other is the (T149A/G145D/A344V) variant of the enzyme from *Oerskovia xanthineolytica* with the 6xHIS-tag placed at the C-terminus of the enzyme (**Salazar et al., 2006**). Both enzymes are produced by T7 IPTG induction. We grow *E. coli* BL21(DE3) cells in autoinduction medium ZYM-5052 (**Studier, 2005**), induce at 18 °C, and purify the recombinant proteins on nickel-NTA resin. Briefly, cell pellets are lysed by sonication in 50 mM sodium phosphate pH 8, 300 mM NaCl, 10 mM imidazole, 10% glycerol, and 0.1% Triton X-100. After lysate clarification, contaminating DNA from the lysate is removed by ion-exchange by the addition of 1 mL neutralized PEI (10% solution) per 10 g wet cell pellet. The opaque mixture is centrifuged, and the supernatant is incubated on a 10 mL Ni-NTA resin bed. After extensive washing of the resin, the protein is eluted with 250 mM imidazole and dialyzed against 50 mM sodium phosphate pH 7.5, 50 mM NaCl, 10% glycerol. We finally add DTT to 10 mM and Triton X-100% to 0.1%. The enzymes can be stored in these buffers indefinitely at –20 °C. We produce approximately 150 mg of enzyme per liter of culture for the *A. luteus* enzyme and 60 mg of enzyme per liter of culture for the *O. xanthineolytica* enzyme.

We mix the two purified enzymes at a 1:1 ratio, and find that these enzymes act synergistically in the presence of DTT to lyse yeast cells. Cell lysis of the F1 segregants is carried out by mixing saturated cultures with glucanase mix at a 5:3 ratio and incubating for 45 min at 30 °C for cell wall digestion. Cells are then heated to 80 °C for 5 min to lyse the nuclei and denature DNA binding proteins. The resulting lysate can be stored at 4 °C and used directly for tagmentation.

## Barcoded Tn5-mediated sequencing

The tagmentase enzyme Tn5 is often used in library preparation for its ability to fragment double-stranded DNA and add oligos to both ends of each fragment in a single isothermal reaction, a process first described by **Adey et al., 2010** and later commercialized as Nextera by Illumina. The attached oligos serve as priming sites for the addition of indexed sequencing adaptors by PCR, which identify individual samples in a multiplexed pool. Since carrying out 100,000 individual PCR reactions would be challenging, we aimed to incorporate indexed oligos during the tagmentation reaction itself, which would allow us to pool samples for the following PCR step. Similar reaction protocols are commercially available (e.g. plexWell by seqWell).

The tagmentase enzyme Tn5 can be purchased (e.g. from Lucigen) or produced according to published protocols (*Picelli et al., 2014*; *Hennig et al., 2018*). Briefly, *E. coli* cells expressing Tn5 tagged with intein-CBD (chitin binding domain) were grown at 37 °C and induced at 18 °C. Cells were then lysed by sonication in high-salt buffer (20 mM HEPES-KOH pH 7.2, 800 mM NaCl, 10% glycerol, 1 mM EDTA, 0.2% Triton X-100). After clarifying the lysate by centrifugation, neutralized PEI was added dropwise to 0.1% and stirred for 15 min. The opaque mixture was centrifuged and the supernatant was loaded on chitin resin (New England Biolabs). After extensive washing with lysis buffer, DTT was added to 100 mM to initiate intein cleavage, after which the column was incubated for 48 hr. The eluted product was then collected and concentrated to approximately 1.5 mg/mL (as determined by Bradford assay using bovine serum albumin as standard), and dialyzed against dialysis buffer (50 mM HEPES-KOH pH 7.2, 800 mM NaCl, 0.2 mM EDTA, 2 mM DTT, 20% glycerol). The enzyme was stored by mixing 1:1 with a buffer containing 80% glycerol and 800 mM NaCl to produce the final storage conditions (25 mM HEPES-KOH pH 7.2, 800 mM NaCl, 0.1 mM EDTA, 1 mM DTT, 50% glycerol). Aliquots were then flash-frozen in liquid nitrogen and stored at –80 °C.

Before use, purified Tn5 needs to be activated with appropriate oligonucleotide adapters, which in our case carry unique indices. These oligos are double-stranded DNA molecules composed of a constant region for the Illumina sequencing adapters, a variable region for unique indices, and a constant region (mosaic ends) to which the enzyme binds (Tn5ME: AGATGT GTATAA GAGACA G). The Illumina-compatible region comes in two versions, corresponding to the priming sequences to attach P5 and P7 sequencing adapters in a subsequent PCR step; we term these as Left (for P5) and Right (for P7). For unique identification of each well in a 384-well plate, we chose 16 different Left adapter indices and 24 different Right adapter indices, and ordered the resulting constructs as single-stranded oligos. Our oligos have the following configuration: Tn5-L = 5'-GCCTCC CTCGAG CCATGA AGTCGC AGCGTY YYYYYA GATGTG TATAAG AGACAG-3', Tn5-R = 5'-GCCTTG CCAGCC CGAGTG TTGGAC GGTAGY YYYYYA GATGTG TATAAG AGACAG-3', where the variable region for unique indices is represented by YYYYYY. These oligos are converted to double-stranded oligos by annealing to a single reverse sequence complementary to Tn5ME (Tn5ME-Rev: 5'-[phos]CTGTCT CTTATA CACATC T-3'). We mix 157.5 µM Tn5ME-Rev with 157.5 µM indexed oligo (a single L or a single R oligo), 10 mM Tris pH 8, 1 mM EDTA, and 100 mM NaCl in 50 µL final volume. In a thermocycler, the mixture is heated to 95 °C and cooled at a rate of 1 °C/min until 15 °C is reached. To activate the enzyme, we thoroughly mix 50 µL annealed adapters with 600 µL prepared Tn5 (at about 0.75 mg/mL) and incubate for 1 hr at room temperature. Activated Tn5 can then be stored at –20 °C. The enzyme preparation for each index is tested for activity and approximately normalized by dilution if needed (this is important for maintaining consistency in DNA fragment size across samples).

The Tn5 tagmentation reaction of genomic DNA is performed by mixing approximately 40 ng genomic DNA, 25 µL of 2 x transposition buffer (20% propylene glycol, 24% dimethylformamide, 150 mM KCl, 20 mM MgCl$_2$, 4 mM Tris pH 7.6, 200 µg/mL BSA), and an equal mixture of L and R activated Tn5 (at an empirically determined concentration corresponding to optimal activity for each activated preparation of Tn5) in a final reaction volume of 50 µL. The reaction is assembled, mixed, and incubated for 10 min at 55 °C. After the transposition reaction, we immediately add 12.5 µL of stop buffer (100 mM MES pH 5, 4.125 M guanidine thiocyanate, 25% isopropanol, 10 mM EDTA). The resulting reactions from different combinations of L and R indices (i.e. up to 384 reactions) can be pooled at this stage. We proceed with purification by mixing 125 µL of the pooled samples with 375 µL of additional stop buffer and passing through a single silica mini-preparative column. The column is washed once with buffer containing 10% guanidine thiocyanate, 25% isopropanol, and 10 mM EDTA; then washed once with buffer containing 80% ethanol and 10 mM Tris pH 8; then dried and eluted with 10 mM Tris pH 8.5.

The resulting library can be amplified by PCR with indexed primers carrying the P5 and P7 adapter sequences: P5mod = 5'-AATGAT ACGGCG ACCACC GAGATC TACACY YYYYYG CCTCCC TCGAGC CA-3', P7mod = 5'-CAAGCA GAAGAC GGCATA CGAGAT YYYYYY GCCTTG CCAGCC CG-3'. Here YYYYYY represents 6 bp multiplexing indexes, which are chosen to maximize Levenshtein distance between every pair. To 20 µL purified transposase reaction, we add 25 µL 2 x Kapa HIFI master mix (KAPA Biosystems) and 2.5 µL each of P5mod and P7mod primers at 10 µM. We use the following cycling protocol: (1) 72 °C for 3 min, (2) 98 °C for 5 min, (3) 98 °C for 10 s, (4) 63 °C for 10 s, (5) 72 °C for 30 s, (6) Repeat steps (3)–(5) an additional 13 times, (7) 72 °C for 30 s, (8) Hold at 10 °C.

We then perform a double-size selection with AMPure beads (Beckman Coulter) to select for fragments between 300 bp and 600 bp. Finally, we quantify DNA concentration with AccuClear Ultra High Sensitivity (Biotium) and pool libraries at equimolar concentration. For sequencing on Illumina machines with the paired-end workflow of the NextSeq 500, we use the following custom primers: Custom_read_1 = 5'-GCCTCC CTCGAG CCATGA AGTCGC AGCGT-3', Custom_read_2 = 5'3', Custom_read_3 = 5'-ACGCTG CGACTT CATGGC TCGAGG GAGGC-3', Custom_read_4 = 5'-GCCTTG CCAGCC CGAGTG TTGGAC GGTAG-3'. On the NovaSeq 6000, the paired-end workflow does not use Custom_read_3.

To reduce sequencing costs, we sequence each individual only to low coverage and then impute the full genotypes (see Appendix 1). We can quantify the SNP coverage (average number of reads per SNP) and SNP coverage fraction (number of SNPs seen in at least one read) for each individual. From a single NovaSeq S4 sequencing flowcell and 4 NextSeq 500 High flowcells (total ~11.5 billion reads), we obtained a median coverage of 0.57 X (median coverage fraction 0.26) and a mean coverage of 1.23 X (mean coverage fraction 0.34).

## Combinatorial indexing and sequencing of barcodes

One consequence of low-coverage sequencing of our individuals is that the barcode locus is not guaranteed to be observed. Therefore, we need a separate method to associate the genotype of each individual with its barcode. The genotype information is already associated to the physical location of each segregant in the 384-well plates, via the Tn5-L and -R indices used during tagmentation and the P5mod and P7mod indices used during PCR. We can therefore match barcodes with segregants by associating the physical location of each segregant with its barcode. We achieve this with minimal additional sequencing cost using a multidimensional pooling approach (*Erlich et al., 2009*).

### Combinatorial pooling

The key idea of multidimensional pooling is to identify the location of many points using only a few measurements, by projecting the points onto several orthogonal axes and measuring them in bulk along each axis. In our case, we construct various orthogonal pools of all of the segregants (such that each segregant is present in a unique set of pools). We then perform bulk barcode sequencing on each pool, to identify all the barcodes present in that pool. Because we know which segregant is in which specific set of pools, from its physical location, we can then associate barcodes with segregants by looking for the unique barcode which is present in the set of pools corresponding to that segregant.

There are many equivalent choices of pooling strategies; our choice was mainly informed by the physical layout of our segregants and experimental convenience. For example, since all segregants are originally grown in 96-well plates, a natural choice is to pool all segregants in each row across all plates, producing eight row pools, and similarly to pool all segregants in each column across all plates, producing 12 column pools. In addition, we construct three other types of pools that are orthogonal and convenient to assemble within our large-scale experimental workflow: plate pools (4), set pools (12), and batch pools (23). These five pools provide sufficient information to uniquely triangulate the specific well of each barcode within our full collection of archived plates. This pooling strategy also means that we expect each barcode to be present in exactly one of each of the five sets of pools (e.g. one of the eight row pools, one of the twelve column pools, etc.), which is useful in correcting errors that arise during the bulk barcode sequencing.

Amplicon barcode sequencing is performed as described below for bulk phenotyping measurements. Each pool contains between 4608 and 26,496 barcodes, and we sequence the pools with on average 61 x coverage per barcode per pool.

### Barcode assignment to single wells

In theory, with enough orthogonal pools and exact measurements of each pool, the reconstruction of the full data is unambiguous: each barcode appears in exactly one of the row pools, giving its row location, and exactly one of the column pools, giving its column location, and so on. However, because sequencing readout is not exact, there are several types of errors that can occur. First, there can be sequencing errors in the barcode regions themselves, producing novel barcodes. Second, due to index-swapping and PCR chimera artifacts, a barcode may appear in multiple of one type of pool (e.g. some reads in multiple row pools), leaving its location partially undetermined (e.g. missing row

information). Third, due to differences in coverage, a barcode may not appear in any of one type of pool (e.g. no reads in any row pool), again leaving its location partially undetermined. Fourth, due to PCR chimera artifacts, some reads are observed with a barcode pair that does not correspond to a real segregant; these typically appear as singletons in one instance of one type of pool. And finally, our observation for each pool is an integer number of read counts, which can contain more useful information than a simple binary presence/absence.

To address the first source of error, we must 'correct' barcodes with sequencing errors. This is possible because we expect sequencing errors to be rare (affecting at most a few bases in a barcode) and because the barcodes themselves, due to their length and random composition, are well-separated in sequence space. Clusters of similar barcodes separated by only one to three nucleotide substitutions thus typically represent a single true barcode. Here, we leverage the fact that barcodes are recombined from two parental sources (the BY and RM parent). Since there are relatively few single parental barcodes (~23,000 BY barcodes and ~ 12,000 RM barcodes) compared to double-barcode combinations, we can sequence the single parental barcodes to much higher depth. We sequenced the 23 BY and 12 RM parental libraries on an Illumina NextSeq with 383 X reads per barcode on average. We then perform error correction on these parental libraries, by identifying sequence clusters and correcting all reads within each to the consensus sequence, as described in *Nguyen Ba et al., 2019*. This generates two 'dictionaries' containing all verified BY and RM barcodes, respectively. We then use these dictionaries to correct errors in the barcode sequencing reads from the compressed sensing pools, correcting BY (RM) barcodes according to the BY (RM) dictionary. Specifically, a 16 bp barcode within Levenshtein distance three or fewer from a verified dictionary barcode is corrected to the dictionary sequence, while those further than Levenshtein distance three are discarded.

Subsequently, to assign barcodes to unique wells, we use a combination of heuristic and algorithmic assignment procedures that incorporate our prior knowledge of the error processes mentioned above. First, we impose a cutoff threshold of 5 reads for each barcode in each pool, such that any instances of fewer than five reads are discarded. This removes a large portion of index-swapping and PCR chimera errors (and the accuracy of barcode assignment is robust to thresholds of 3–10 reads).

Next, we extract those barcodes with unambiguous assignments (i.e. barcodes that appear in exactly one of each of the ive types of pools). These barcodes can have any number of reads (above the threshold) in the pool where they appear, but must have no reads (i.e. fewer than the threshold) in all other pools of that type. The resulting five numbers (e.g. batch pool 7, set pool 9, plate pool 3, row pool 8, column pool 8) constitute what we term the 'address' of that barcode, and indicate its physical location in our archive plates. From the genotype sequencing indices (Tn5-L and -R, and P5mod and P7mod), we can extract the same five numbers for each genotype. We thus match the genotype with its barcode via its location in the archive collection. Within these 'first pass' assignments, only 190 wells are assigned two or more barcodes, representing 0.2% of all non-blank wells. These wells and barcodes are removed from further analysis. After these are removed, the first pass produces 46,403 unique barcode-address assignments, representing 45.9% of all non-blank wells in our full collection.

From inspection of the remaining unassigned barcodes, we find that most either appear in multiple pools of one type or do not appear in any pool of one type (i.e. the second and third problems mentioned above). However, given that they tend to have clear assignments for three or four of the five pool types, and that we already have many wells successfully matched, we might expect that in some cases the ambiguous pools could be identified by deduction (e.g. if one barcode has a known column but is missing row information, and the other seven of eight wells in that particular column have already been assigned, then this barcode must belong to the only unoccupied well). Once that well is assigned, perhaps other ambiguities in that row will be resolved, and so forth. This is an example of a linear assignment problem (LAP), for which fast algorithms have been developed (*Jonker and Volgenant, 1987*). We can use such an algorithm to identify matches between our remaining unassigned wells and unassigned barcodes.

To implement a LAP solver, we need to define a suitable cost function for assigning a barcode to a well based on read count data. Our intuition is that larger read counts for a particular barcode in any pool should decrease the cost of an assignment to that pool, relative to others. Formally we calculate the likelihood of observing a certain pattern of counts across pools, and our cost function is given by the negative log-likelihood. Our cost function will produce an $n \times w$ matrix where $n$ is the number of

barcodes to be assigned and $w$ is the number of wells to be assigned. Specifically, after the first pass and after removing blank wells, we have n=53,560 barcodes and w=54,483 wells.

For clarity we first consider a single type of pool, here row pools (of which there are 8). We consider a particular barcode belonging to segregant i. This segregant exists at some true frequency $f_{i,r}$ in each row pool $r = 1, \ldots, 8$. One of these pools (that representing the true location) should have a substantial frequency, while the frequency in the other pools would represent index-swapping, cross-contamination, or other sources of error. Instead of these frequencies, what we observe are read counts $m_{i,r}$ for each of the eight pools.

Let us consider a particular pool $r$, where barcode i has true frequency $f_{i,r}$ and is observed with read counts $m_{i,r}$. We will calculate the probability for the observed pattern of read counts under the assumption that this pool $r$ is the true pool, and the other pools $r' \neq r$ are 'error' pools (we will evaluate this probability for each pool in turn, and then normalize to obtain posterior probabilities, see below). We group the other pools together into one error pool where the barcode i exists at a true frequency $\tilde{f}_{i,r}$ and is observed with read counts

$$\tilde{m}_{i,r} = \sum_{r' \neq r} m_{i,r'}. \tag{1}$$

We also define the total read counts observed for the pool under consideration and the remaining pools, respectively, as

$$M_r = \sum_i^n m_{i,r}, \tag{2}$$

$$\tilde{M}_r = \sum_i^n \tilde{m}_{i,r}, \tag{3}$$

where the sums are taken over all segregant barcodes $i$. Since we have observed two counts $m_{i,r}$ and $\tilde{m}_{i,r}$ sampled from two underlying frequencies $f_{i,r}$ and $\tilde{f}_{i,r}$, we write the probability of observing the count data as a product of two binomials:

$$P(m_{i,r}, \tilde{m}_{i,r}|M_r, \tilde{M}_r, f_{i,r}, \tilde{f}_{i,r}) = \left[(f_{i,r})^{m_{i,r}}(1 - f_{i,r})^{M_r - m_{i,r}}\right]\left[(\tilde{f}_{i,r})^{\tilde{m}_{i,r}}(1 - \tilde{f}_{i,r})^{\tilde{M}_r - \tilde{m}_{i,r}}\right]. \tag{4}$$

Since we do not have access to the true frequencies $f_{i,r}$ and $\tilde{f}_{i,r}$, we replace them with their maximum likelihood estimates, that is, the observed frequencies in the read count data:

$$f_{i,r} \rightarrow \frac{m_{i,r}}{M_r}, \tag{5}$$

$$\tilde{f}_{i,r} \rightarrow \frac{\tilde{m}_{i,r}}{\tilde{M}_r}. \tag{6}$$

Under the assumption that pool $r$ is the correct pool, we expect $\tilde{f}_{i,r} < f_{i,r}$. To enforce this prior, we set

$$f_{i,r} \rightarrow \max\left(\frac{m_{i,r}}{M_r}, \frac{\tilde{m}_{i,r}}{\tilde{M}_r}\right), \tag{7}$$

$$\tilde{f}_{i,r} \rightarrow \min\left(\frac{m_{i,r}}{M_r}, \frac{\tilde{m}_{i,r}}{\tilde{M}_r}\right). \tag{8}$$

This also breaks the symmetry inherent in *Equation 4* when there are counts in only two pools (i.e. if there are only pools $r$ and $r'$, then $m_{i,r'} = \tilde{m}_{i,r'}$, and thus $\mathcal{P}(m_{i,r}, \tilde{m}_{i,r}|M_r, \tilde{M}_r, f_{i,r}, \tilde{f}_{i,r}) = \mathcal{P}(m_{i,r'}, \tilde{m}_{i,r'}|M_{r'}, \tilde{M}_{r'}, f_{i,r'}, \tilde{f}_{i,r'})$), such that the probabilities for $r$ and $r'$ to be the true pool would be equal even when the counts are very different.)

We evaluate this probability (*Equation 4* with replacements from *Equation 7* and *Equation 8*) for each pool $r$ in turn. We then normalize to obtain the final posterior probabilities for the observed read counts:

$$\mathcal{P}(i, r) \equiv \mathcal{P}(m_{i,r}, \tilde{m}_{i,r}|M_r, \tilde{M}_r, f_{i,r}, \tilde{f}_{i,r}) = \frac{P(m_{i,r}, \tilde{m}_{i,r}|M_r, \tilde{M}_r, f_{i,r}, \tilde{f}_{i,r})}{\sum_r^8 P(m_{i,r}, \tilde{m}_{i,r}|M_r, \tilde{M}_r, f_{i,r}, \tilde{f}_{i,r})}. \tag{9}$$

Our cost function for assigning barcode $i$ to any well in row $r$ is then given by the negative log posterior probability:

$$\mathcal{C}(i, r) = -\log \mathcal{P}(i, r). \tag{10}$$

For the row in the cost matrix corresponding to segregant $i$, and for one out of every eight columns (i.e. all of the wells across all of the plates that correspond to row $r$), we add this cost. We then repeat for all of the other row pools, completing the row costs for this segregant.

Next we proceed in the same manner with the 12 column pools, 4 plate pools, 12 set pools, and 23 batch pools. For each type of pool, we scan through each pool, evaluating the probability for each pool, normalizing by the sum across pools, taking the negative log, and adding the resulting cost to all wells corresponding to that pool. Once this process is complete, we have finished calculating the costs for segregant i, that is, we have filled in one complete row of the cost matrix. Each entry in this row represents one assignment (i.e. one unique row/column/plate/set/batch combination $r, c, p, s, b$ paired with segregant barcode i), and since we have summed the costs over all of the 5 types of pools, the final posterior probability for each assignment is given by

$$\mathcal{P}(i, r, c, p, s, b) = \mathcal{P}(i, r) \cdot \mathcal{P}(i, c) \cdot \mathcal{P}(i, p) \cdot \mathcal{P}(i, s) \cdot \mathcal{P}(i, b). \tag{11}$$

Finally, we repeat this process for all segregants $i = 1, \ldots, n$. This produces our final cost matrix of dimension $n = 53,560$ barcodes by $w = 54,483$ wells.

With the cost matrix in hand, we use the Python package lapsolver (*Heindl, 2020*) to find the optimal assignment. Since our cost matrix is rectangular (fewer barcodes than wells), we obtain an assignment for every barcode but some wells remain unassigned. These can be clearly interpreted as "quasi-blanks": in our plate scans, we see a small number of wells that contain only tiny colonies after two days of growth, large enough to be distinguished by eye from true blanks but too small to be represented well in pool sequencing, genotyping, or bulk fitness assays.

Combining the first-pass assignments and LAP assignments, we obtain a list of 99,963 unique barcode-well pairs. These assignments allow us to link the genotyping information for each segregant, which is indexed by well location, to the phenotyping information for each segregant, which consists of barcode sequencing from bulk fitness assays (see Appendix 2). By comparing well addresses to our list of inferred genotypes, we find 99,950 matches, producing our final strain collection with confirmed genotype-barcode association.

Next we consider several validations of these assignments. If mis-assignments occur, such that genotype and phenotype data are mis-paired, we expect this to reduce signal and add noise to our QTL inference procedure. We do not expect systematic errors to arise from mis-assignment, as the distribution of genotypes across well locations is sufficiently random (because cells are sorted into plates from large well-mixed liquid cultures). Thus, we expect our QTL inference results to be robust to a small level of mis-assignment (i.e. a few percent of barcodes).

While we cannot have complete confidence in every assignment, especially those with sparse read data, we can validate that the large majority of assignments are indeed correct by making use of our knowledge of parental barcodes. Recall that each segregant contains two barcodes, the first from its BY parent (the 'BY barcode') and the second from its RM parent (the 'RM barcode'), and that the matings were done from 23 × 12 separate pools with unique barcodes. Here the 23 BY parental libraries were used for the 23 batches, and the 12 RM parental libraries were used for 12 'sets' of 4 96-well plates within each batch. We also deeply sequenced each parental library, so that we know which BY barcodes belong to each batch and which RM barcodes belong to each set. Thus, for each assignment, we can check whether the batch and set, obtained from the above procedure, agree with the parental libraries. For the first-pass barcodes (those with complete and unique addresses), we observe that 99.5% of the 46,403 assignments have correct matches for both batch and set. For the second-pass barcodes (those obtained through the LAP algorithm), the fraction with both batch and set correct is 97.7% of the 53,560 assignments. Although this method does not allow us to verify the other three types of pools (plate, row, and column), these high fractions of correct assignments indicate that both passes are extremely effective at extracting robust assignments from our noisy read count data.

## High-throughput liquid handling of the F1 segregants

All of the procedures described in sections above (i.e. the segregant production, selection, and sorting, the combinatorial pooling, and the individual whole genome sequencing library preparation) must

be performed at the scale of 100,000 segregants, corresponding to more than 1000 96-well plates. To accomplish this within time and cost constraints, we relied on extensive use of liquid-handling robots as well as a batch process workflow. Here we describe the operational protocol for large-scale processing.

Many of our processes require simple pipetting in 96-well format and/or pooling across rows and columns of 96-well plates. These operations lend themselves readily to automation by liquid-handling robots. We made extensive use of two Beckman Coulter Biomek FXp instruments, each with a 96-channel pipetting head and one with an additional Span-8 head.

We chose to break the full experiment into manageable 'batches' to reduce equipment costs (by reusing consumables and equipment for each batch) and improve consistency (by ensuring each step is performed near-simultaneously for all segregants in a batch). Due to our time and equipment constraints, we chose a batch size of 48 96-well plates, or 4608 segregants. Each batch requires 14 days to process, but multiple batches can be processed concurrently in a staggered fashion. We regularly maintained 2–6 batches in various stages, allowing us to complete 23 batches in about 6 months.

## Production of barcoded F1 segregants

As described above, we produced multiple separate parental barcode libraries (23 for BY and 12 for RM). For each of the 23 batches, we mated one of the BY barcoded libraries to all 12 RM libraries separately, each mating representing a 'set' within that batch. We proceeded with mating, barcode recombination, barcode selection, sporulation, single-cell sorting and growth as described above, with the 12 sets kept separate throughout. From each of the 12 sets, we sorted 384 individual cells into four 96-well plates, resulting in the 48 96-well plates for that batch. From these plates, we use a portion of cells for the archive construction, combinatorial pool construction, and genotyping as described below. Note that because we sequenced each of the BY and RM parental libraries, we can identify in which batch and set each segregant was produced based on its first (BY) and second (RM) barcode, respectively.

## Archiving of the F1 segregants

Once the cells have grown, we create a frozen archive with segregants individually arrayed in 384-well format, to serve as an ultimate repository of the strain collection. For each batch, we use the Biomek to fill twelve 384-well 'archive' plates (Bio-Rad, Hard-Shell #HSP3801) with 10 µL of 15% glycerol. The 48 96-well plates of saturated cultures were shaken in a tabletop plate shaker (Heidolph Titramax 100) to ensure cell dispersion. The Biomek was then used to transfer and mix 20 µL of culture into each well of the archive plates (final concentration of 5% glycerol) in groups of 8 plates at a time. Archive plates were sealed with aluminum foil and stored at –80 °C.

## Cell lysis

After archiving, another portion of cells is transferred to 384-well 'lysis' plates for the lysis, extraction, and tagmentation reactions. We first use the Biomek to transfer 3 µL of glucanase mix to the twelve 384-well lysis plates. We use the Biomek to add and mix 5 µL of saturated culture into each well, again processing eight plates at a time. We seal the plates with PCR sealing film (VWR 89134–428) and then incubate at 37 °C in an air-incubator for 45 min, followed by heating in a thermocycler at 80 °C for 5 min. The cell lysates are stable at 4 °C or can be immediately processed.

## Tn5 transposition

According to our indexed tagmentation design, each well in a 384-well plate is uniquely indexed by the combination of L-indexed and R-indexed Tn5 preparations that it receives. We implement this reaction by first using the Biomek Span-8 to array the concentrated enzymes in four 96-well master mix plates; for each 384-well lysis plate in the batch, we then add enzyme mixture to the corresponding wells, perform the tagmentation reaction, and then pool the samples into one tube for later purification and PCR.

Starting from 16 L-indexed and 24 R-indexed Tn5 preparations, we aim to generate four 96-well plates where each well is indexed by a unique L/R pair. As the enzyme is very viscous, thorough mixing at this step is critical, and we take advantage of inert dyes to judge mixing. We dye the

Tn5 preparations with xylene cyanol FF at 500 µg/mL, and the 2 x tagmentation buffer with orange G at 140 µg/mL, which when mixed will yield a green enzyme master mix. 82.5 µL of 2 x buffer is first aliquotted into six 96-well polypropylene PCR plates (VWR 89049–178; polystyrene is sensitive to dimethylformamide in the buffer). The L-indexed activated Tn5 preparations are then aliquotted row-wise in four of the plates (7.5 µL per well), while the R-indexes are aliquotted column-wise in the remaining two plates (7.5 µL per well). We then transfer from the R-index plates into the L-index plates in such a way as to form unique L/R combinations in each of the wells of the four 96-well plates. We perform thorough pipette mixing with the Biomek and visually check the master mix color to ensure consistency across all wells before use. These four plates contain sufficient master mix for one batch of reactions (12 384-well plates).

Activated Tn5 is stable on ice. We also found that loss of activity is negligible in the presence of buffer containing magnesium as long as the enzyme is not allowed to warm above 4 °C. To maintain enzyme temperature for the processing duration of a full batch (several hours at ambient temperature on the Biomek), we take advantage of Biomek-compatible prechilled Isofreeze thermal blocks (VWR 490004–218), swapped out as needed, as well as chilled buffer kept at –20 °C.

To form the final tagmentation reaction, we add 12 µL of master mix (10 µL of 2 x buffer, 1 µL of L-index enzyme, 1 µL of R-index enzyme) to the 8 µL of cell lysate in the 384-well plates. After thorough pipette mixing, the reaction is incubated for 10 min at 55 °C in an air incubator. We process three rounds of four 384-well plates to complete the batch.

Once completed, the reaction is quickly stopped by adding 2.5 µL of stop buffer to every well using the Biomek. We then use the Span-8 head to pool each 384-well plate into a single 1.5 mL microfuge tube, resulting in 12 tubes per batch. Column purification and PCR are performed as described above.

## Combinatorial pooling

For our combinatorial pooling barcode association, we chose to create the following pools: 12 column pools, each drawn from one of twelve columns from every 96-well plate in the experiment; eight row pools, each drawn from one of eight rows from every 96-well plate in the experiment; four plate pools, each drawn from one of the four 96-well plates in every set of the experiment; 12 set pools, each drawn from one of the twelve sets in every batch of the experiment; and 23 batch pools, each drawn from a whole batch of the experiment. This resulted in 59 total pools, each containing between 4608 and 26,496 barcodes, depending on the type of pool.

We constructed these pools by processing four 96-well plates at a time on the BioMek using the Span-8 head. Briefly, for each set, we first duplicate the original four plates of segregants (after cells for archiving and tagmentation have been removed) by aliquotting culture into four new 96-well plates. One group was pooled row-wise and the other pooled column-wise into tubes representing row and column pools respectively. For a batch, we process the 12 sets sequentially, adding to the same final tubes. These tubes can later be pooled by hand across batches before DNA extraction, PCR, and barcode sequencing.

For each batch, we also pooled each of the 48 96-well plates into their own single tube. These tubes are later pooled by hand across plates, sets, or batches as appropriate to produce the plate pools, set pools, and batch pools.

## Producing pools for bulk phenotyping assays

It is convenient for our bulk phenotyping assays to produce frozen stocks containing all of our segregants, as well as the reference parental strains BYref and RMref, in a single pool. As we process each batch, we use the Biomek to pool across all 48 96-well plates of segregants into a single tube, to which we manually add glycerol to a final concentration of 5% before storage at –80 °C. After we have processed all 23 batches, these 23 individual pools can be later thawed, diluted 1:2[5] in 5 mL YPD, and grown for 24 hr, after which they are combined at equal frequency into a master pool containing all 99,950 lineages. To this pool, we add our reference (parental) strains BYref and RMref at a frequency approximately 100 times larger than the average barcode frequency. We then add glycerol to 5% final concentration and store this master pool in aliquots at –80 °C. Each aliquot can then be thawed as needed to conduct bulk phenotyping assays.

## Material reuse by washing

In an effort to reduce material cost and waste, we wash and reuse both Biomek liquid handling tips and polypropylene 96-well plates. Polypropylene 96-well plates used to grow and pool yeast cells are thoroughly washed with water and sterilized by autoclave, after which they can be reused with no ill effects.

We developed Biomek-based tip washing protocols that successfully prevent cell and DNA cross-contamination between wells. Tips that contact living cells are sterilized by pipette-mixing several times with sterile water, followed by pipette-mixing several times in 100% ethanol. After air-drying in boxes overnight, tips are ready to be reused. However, this protocol is insufficient for potential DNA cross-contamination during the tagmentation reaction. Tips that contact DNA are first decontaminated by pipette-mixing several times in 20% bleach, followed by water and ethanol wash steps as above. We found this to be sufficient to remove any traces of DNA contamination.

## Bulk phenotyping

A key challenge for QTL mapping at scale is performing large numbers of accurate phenotype measurements. In our case, we require millions of measurements (dozens of phenotypes for each of $\sim 100,000$ segregants). Each measurement must be made with high precision to achieve accurate mapping of polygenic traits. We achieve this using barcode-based assays which we can perform in bulk and read out by sequencing. The key simplification is that, because we know the genotype corresponding to each barcode, sequencing the barcode locus compresses each segregant's full genotype information into a single sequencing read. This high information density, combined with the flexibility of sequencing-based assays and the decreasing cost per base of next-generation sequencing, allow us to collect a wide variety of phenotype measurements for reasonable cost and time expenditures.

One type of trait that lends itself well to bulk sequencing-based assays is growth rate in different media conditions, which we refer to here as 'fitness'. Previous QTL mapping work in yeast often measured growth rate by observing the growth of individual colonies on agar plates (*Bloom et al., 2013*; *Bloom et al., 2019*). Instead, we can combine all of our segregants into one well-mixed liquid culture, resulting in an all-to-all competition throughout multiple rounds of serial dilution passaging. Barcoded lineages will change in frequency in the population according to their relative fitness, which we read out by population barcode sequencing at multiple timepoints. Note that in our serial dilution assays, 'fitness' is determined from the frequency change over multiple passages (see Appendix 2), which is not exactly equal to the growth rate as measured in log phase. We typically maintain such bulk fitness assays for about 50 generations, which provides enough time for lineages of similar fitness to be distinguished, while not allowing for lineages to fix in the population or acquire additional mutations that would significantly impact the frequency trajectories. To avoid the introduction of noise from genetic drift, we maintain effective population sizes of $\sim 10^8$.

All of the phenotypes reported in this paper are 'fitness' phenotypes, obtained by bulk fitness assays as we describe here. However, we also note that bulk barcode-based phenotyping is not limited to fitness phenotypes. Any phenotype that can be translated by some assay into barcode frequency changes is amenable. This includes 'shock' phenotypes, where frequencies are measured before and after a sudden perturbation; plating-based phenotypes, like mating efficiency and transformation efficiency; and fluorescence phenotypes, obtained by flow cytometry sorting and sequencing, among others.

### Growth experiments

The complete frozen pool of F1 segregants (containing reference parental strains) was grown in 5 mL YPD by inoculating approximately $10^7$ total cells. We diluted these populations by 1:$2^7$ daily by passaging 781 μL into 100 mL fresh media in 500 mL baffled flasks. Whole population pellets, obtained from 1.5 mL of saturated culture, were stored daily at –20 °C for later sequencing. As previously described (*Nguyen Ba et al., 2019*), this protocol results in about seven generations per day, with a daily bottleneck size of about $10^8$ in most assay environments. We performed two replicates of each assay and sampled for 49 generations (seven timepoints). Only five timepoints (representing 7, 14, 28, 42, and 49 generations) were sequenced.

## Amplicon barcode sequencing

Genomic DNA from cell pellets was processed as in *Nguyen Ba et al., 2019*. Briefly, DNA was obtained by zymolyase-mediated cell lysis (5 mg/mL Zymolyase 20T (Nacalai Tesque), 1 M sorbitol, 100 mM sodium phosphate pH 7.4, 10 mM EDTA, 0.5% 3-(N,N-Dimethylmyristylammonio)propanesulfonate (Sigma, T7763), 200 µg/mL RNAse A, and 20 mM DTT) and binding on silica mini-preparative columns with guanidine thiocyanate buffer (4 volumes of 100 mM MES pH 5, 4.125 M guanidine thiocyanate, 25% isopropanol, and 10 mM EDTA). After binding, the columns were washed with a first wash buffer (10% guanidine thiocyanate, 25% isopropanol, 10 mM EDTA) and then a second wash buffer (80% ethanol, 10 mM Tris pH 8), followed by elution into elution buffer (10 mM Tris pH 8.5). 1.5 mL of pelleted cells eluted into 100 µL routinely provided about 1–2 µg of total DNA.

PCR of the barcodes was performed using a two-stage procedure previously described to attach unique molecular identifiers (UMIs) to PCR fragments (see *Nguyen Ba et al., 2019* for a detailed protocol). Primers used in the first-stage PCR contained a priming sequence, a multiplexing index, 8 random nucleotides as UMIs, and an overhang that matched the Tn5 transposome present in our indexed tagmentase. These two primers had the configurations P1 = GCCTCC CTCGAG CCATGA AGTCGC AGCGTN NNNNNN NYYYYY YYGCAA TTAACC CTCACT AAAGG, P2 = GCCTTG CCA-GCC CGAGTG TTGGAC GGTAGN NNNNNN NYYYYY YYGCTA GTTATT GCTCAG CGG. Here, N corresponds to degenerate bases used as UMIs, and Y corresponds to multiplexing indexes. These primers anneal within the artificial intron of the URA3 gene in our recombined landing pad, at the KanP1 and HygP1 sites respectively. After attachment of molecular identifiers to template molecules during one PCR cycle, the first stage amplicons were cleaned using Ampure beads according to *Nguyen Ba et al., 2019*. The elution of this clean-up was used directly as template for the second stage PCR with primers that contained multiplexing indexes and adapters that anneal to the Illumina flowcells (P5mod and P7mod primers, as described above). After 31 further cycles, these final PCR products were then purified using Ampure beads, quantified, and pooled to equimolar concentration. The PCR products were sequenced with a NextSeq 500 high-output v.2 (Illumina) or a NovaSeq S2 (Illumina) by paired-end sequencing using custom primers compatible with the indexed tagmentases.

We first processed our raw sequencing reads to identify and extract the indexes and barcode sequences, discarding Lox sites and other extraneous regions. To do so, we developed custom Python scripts using the approximate regular expression library regex (*Barnett, 2010*), which allowed us to handle complications in identifying the barcodes that arise from the irregular lengths of the indices and from sequencing errors. We used the following mismatch tolerances: 2 mismatches in the multiplexing index, 4 mismatches in the priming site, 1 mismatch in the barcode overhangs, 1 mismatch in the barcode spacers, and four mismatches in the Lox sites.

This initial processing results in a set of putative barcodes. However, these putative barcodes do not all correspond to true barcodes from the F1 segregants. Instead, a small fraction of reads contain chimeric barcodes as well as barcodes that differ from true barcodes due to sequencing error. Here we leverage our dictionary of verified barcodes obtained from the barcode association procedure. Because we have knowledge of all individual barcodes that can be present in the assay and we expect errors to be rare, we can make 'corrections' to reads with sequencing errors by direct lookup of the lowest Levenshtein distance to the dictionary of verified barcodes. Chimeric reads (with two barcodes that should never appear together) can be easily discarded.

Finally, we can calculate the counts of each error-corrected true barcode by removing duplicate reads, using the unique molecular identifiers from the first-stage PCRs. Frequencies calculated from these counts are used to infer fitnesses for all segregants, as explained in Appendix 2. After all filtering, and across all assays, our final mean (median) sequencing coverage was 185 (48) reads per barcode per timepoint per replicate.

## Reconstruction of causal variants

To confirm the estimated fitness effects of candidate lead SNPs and their interactions from our model, we reconstructed potentially causal variants either as single or double-mutants onto the BY background. Briefly, we started from YAN564 (MATα, his3Δ1, ura3Δ0, leu2Δ0, lys2Δ0, RME1pr::ins-308A, ybr209w:: Gal10pr-Cre, can1::Ste2pr-SpHIS5-Ste3pr-LEU2, HO::SpCas9-B112-ER), which expresses Cas9 under an inducible promoter with 20 µM estradiol. Cells were induced with estradiol and co-transformed with a plasmid expressing a guide RNA targeting the SNP of interest under the

SNR52 promoter and with a double-stranded linear fragment that simultaneously removes the guide targeting sequence and introduces the SNP of interest (allele from RM11-1a). In some cases, guide RNAs were not effective and reconstructions were performed using Delitto Perfetto. Resulting strains also harbor synonymous mutations on the vicinity of the reconstructed SNP as required for the muta-genesis procedure. Reconstructed SNPs were all confirmed by Sanger sequencing. To maximize the probability to observe an effect, we chose lead SNPs that had predicted effects in multiple environments and had small credible intervals, prioritizing non-synonymous variants, and reconstructed key known QTL. In total, we reconstructed the following QTL: HAP1(Ty1*), IRA2(A345T), VPS70(P199L), BUL2(L883F), PHO84(L259P) and MKT1(D30G). For practical reasons, we reconstructed only 13 out of the 15 possible pairwise combinations of these 6 QTL.

The reconstruction strains, as well as YAN564, were individually competed against the diploid fluorescent reference strain YAN563 (MATα/MATa his3Δ1/his3Δ1, ura3Δ0/ura3Δ0, leu2Δ0/leu2Δ0, lys2Δ0/lys2Δ0, RME1pr::ins-308A/RME1pr::ins-308A, ycr043cΔ0::HphMX4/ycr043cΔ0::NatMX, can1::RPL39pr-ymCherry-Ste2pr-SpHIS5-Ste3pr-LEU2/can1::RPL39pr-ymGFP-Ste2pr-SpHIS5-Ste3pr-LEU2) in each of 11 environments: 23 C, 30 C, 37 C, SDS, Li, Cu, Gu, Mann, Raff, Eth, and Suloc. Competition assays were performed in duplicate in shaken microtiter plates with 128 µL of liquid culture, following a daily $1:2^6$ dilution protocol for a total of 3 days (18 generations). The fraction of reference strain in the cultures was tracked daily by flow cytometry, and used to estimate the fitness of each strain relative to the reference strain. Finally, we subtracted the fitness of YAN564 from the fitness of the reconstruction strains and averaged over duplicates to arrive at the relative fitness of each strain relative to BY, which we use for all analyses below.

We compared the measured fitness of each reconstruction strain to the predictions from our additive-only and additive-plus-pairwise models, for both single and double-mutant strains (*Figure 5—figure supplement 1* top and middle row). We find general concordance between predicted and measured fitnesses, except for discrepancies in magnitude (as discussed in the main text). We also find that four out of the double-mutant strains largely depart from values predicted from the models, as well as from the sum of the measured fitnesses of the respective single mutants (*Figure 5—figure supplement 1* yellow-highlighted datapoints), suggesting either a much larger epistatic effect than estimated by our models, or unaccounted mutations from the transformation process such as ploidy changes or mitochondrial disruptions. In line with the second hypothesis, we found that three out of four of these mutants were non-respiring petites (MKT1/VPS70 being the one that respires), and therefore we removed all four incongruent strains from further analyses. We also excluded the outlier measurements from MKT1 mutants assayed in Gu; a measurement that has been independently confirmed in other experiments from our group (unpublished). Possible explanations for this and other discrepancies between predicted and measured effects are additional epistatic terms not detected by our QTL models (e.g. closely linked, or higher-order effects), systematic differences between the environments in bulk fitness assays and reconstruction competition assays, or differences between the panel and reconstruction strains.

To compare how well the additive-only and the additive-plus-pairwise model predictions match measured reconstruction fitness values, we used a permutation test ($H_0$: additive-only and additive-plus-epistasis terms fit measured effects equally well; $H_1$: additive-plus-epistasis terms fit measured effects better than additive-only). We calculated the ratio between the sum squared errors (SSE) between measured and predicted values of the two models, and computed a one-sided p-value from an empirical distribution obtained through 10,000 random permutations of the data. This test rejected $H_0$ when pooling single and double mutant data (SSE ratio of 0.46, $p < 10^{-4}$; *Figure 5A*, inset), as well as when keeping single and double-mutant data separate (*Figure 5—figure supplement 2*; single: SSE ratio of 0.40, $p < 0.01$; double: SSE ratio of 0.48, $p < 10^{-4}$).

## Acknowledgements

We thank the Bauer Core facility at Harvard and the Broad Institute Genomic Services sequencing core for assistance with sequencing. We thank Alan Moses, Andrew Murray, Hunter Fraser, Dan Rice, and members of the Desai lab for comments on the manuscript. ANNB. acknowledges support from the National Science and Engineering Research Council of Canada (NSERC RGPIN-2021–02716 and DGECR-2021–00117). KRL acknowledges support from the Fannie & John Hertz Foundation Graduate Fellowship Award, the National Science Foundation (NSF) Graduate Research Fellowship Program,

and fellowship award #1764269 from the NSF-Simons Center for Mathematical and Statistical Analysis of Biology at Harvard. FM gratefully acknowledges support of the Dana-Farber Cancer Institute Physical Sciences-Oncology Center (NIH U54CA193461). MMD. acknowledges support from grant PHY-1914916 from the NSF and grant GM104239 from the National Institutes of Health (NIH). The computations in this paper were run on the Faculty of Arts and Sciences Research Computing (FASRC) Cannon cluster supported by the FAS Division of Science Research Computing Group at Harvard University.

---

## Additional information

### Funding

| Funder | Grant reference number | Author |
|---|---|---|
| Natural Sciences and Engineering Research Council of Canada | RGPIN-2021-02716 | Alex N Nguyen Ba |
| National Science Foundation | #1764269 | Katherine R Lawrence |
| National Institutes of Health | U54CA193461 | Michael M Desai |
| National Science Foundation | PHY-1914916 | Franziska Michor |
| National Institutes of Health | GM104239 | Michael M Desai |
| Natural Sciences and Engineering Research Council of Canada | DGECR-2021-00117 | Alex N Nguyen Ba |
| Fannie & John Hertz Foundation | Graduate Fellowship Award | Katherine R Lawrence |

The funders had no role in study design, data collection and interpretation, or the decision to submit the work for publication.

### Author contributions

Alex N Nguyen Ba, Katherine R Lawrence, Artur Rego-Costa, Conceptualization, Formal analysis, Investigation, Methodology, Software, Visualization, Writing – original draft, Writing – review and editing; Shreyas Gopalakrishnan, Investigation, Writing – review and editing; Daniel Temko, Investigation, Methodology, Software, Writing – review and editing; Franziska Michor, Methodology, Writing – review and editing; Michael M Desai, Conceptualization, Funding acquisition, Methodology, Supervision, Writing – original draft, Writing – review and editing

### Author ORCIDs

Alex N Nguyen Ba ⓘ http://orcid.org/0000-0003-1357-6386
Artur Rego-Costa ⓘ http://orcid.org/0000-0001-9604-4208
Michael M Desai ⓘ http://orcid.org/0000-0002-9581-1150

### Decision letter and Author response

Decision letter https://doi.org/10.7554/eLife.73983.sa1
Author response https://doi.org/10.7554/eLife.73983.sa2

---

## Additional files

### Supplementary files

• Supplementary file 1. SNP list .
 Our final list of 41,594 SNPs. We provide the chromosome; the SNP index (note these are indexed from 1, while SNP indices in other files are indexed from 0); the chromosome position in basepairs;

and the BY and RM alleles.

• Supplementary file 2. Inferred additive QTL. For each trait, a list of all inferred additive QTL is ranked in order of decreasing effect size magnitude. For each QTL, we provide the effect size; SNP list index of the lead SNP, credible interval (CI) start, and CI end; chromosome; chromosome position (in basepairs) of the lead SNP, CI start, and CI end; and the list of genes with coding regions at least partially overlapping the CI (gene names are given when possible, otherwise ORF names are given).

• Supplementary file 3. Pleiotropic genes. A list of genes containing a lead SNP in two or more traits ('pleiotropic genes'), ranked by the number of traits. For each gene, we give the number of traits in which a lead SNP was detected; the chromosome; the gene name; the number of consensus genes (genes which overlap the intersection of credible intervals from all traits); the names of consensus genes; the lead SNP index identified in each of the traits (blank entries for traits in which the gene was not detected); the lead SNP chromosome position in basepairs in each of the traits; and the effect size in each of the traits. Following the list of pleiotropic genes, we provide a list of remaining QTL detected in only one environment, with associated data in the same format.

• Supplementary file 4. Inferred epistatic QTL. For each trait, separate lists of the additive QTL and pairwise epistatic QTL inferred in additive-plus-pairwise models, ranked in order of decreasing effect size magnitude. For additive QTL, we provide the effect size; SNP list index of the lead SNP; chromosome; and gene in which the lead SNP is located (uppercase gene/ORF names indicate SNPs in coding regions, and lowercase gene/ORF names indicate that the SNP is intergenic but located closest to the gene given). For epistatic QTL, we provide the effect size; SNP list indices of both lead SNPs; chromosomes of both lead SNPs; chromosome positions in basepairs of both lead SNPs; and the gene name of both lead SNPs (formatted as for additive QTL). For both additive and epistatic QTL, we denote QTL located at selection markers (or immediately neighboring genes) in grey and place them at the bottom of the list; see Appendix 3.

• Supplementary file 5. Multiplicity of epistatic QTL across traits. A list of gene pairs that are observed in epistatic interactions across all traits, ranked by the number of traits in which they occur ('edge multiplicity'). For each gene pair, we provide the gene names, corresponding to those in *Supplementary file 3*; edge multiplicity (number of traits in which this pair of genes had a detected interaction); node pair multiplicity (number of traits in which this pair of genes were both detected as additive QTL, whether or not an interaction was detected); and the list of traits in which an interaction was detected. QTL located at selection markers (or immediately neighboring genes) are denoted in grey and placed at the bottom of the list; see Appendix 3.

• Supplementary file 6. Variance partitioning. Variance partitioning, including error, additive genetic, epistatic genetic components, and neural-network on both raw and resampled phenotype data. See Appendix 4 for full definitions and discussion of all components.

• Transparent reporting form

### Data availability

Code used for this study is available at https://github.com/arturrc/bbqtl_inference, (copy archived at swh:1:rev:2b1c89d6a602a8001b3b18dd00e75a8c97950d9d). FASTQ files from high-throughput sequencing have been deposited in the NCBI BioProject database with accession number PRJNA767876. Inferred genotype and phenotype data is deposited in Dryad (https://doi.org/10.5061/dryad.1rn8pk0vd).

The following datasets were generated:

| Author(s) | Year | Dataset title | Dataset URL | Database and Identifier |
|---|---|---|---|---|
| Nguyen Ba AN, Lawrence KR, Rego-Costa A, Gopalakrishnan S, Temko D, Michor F, Desai MM | 2021 | Barcoded Bulk QTL mapping reveals highly polygenic and epistatic architecture of complex traits in yeast | https://datadryad.org/stash/share/-NXjYqEJrSJcSWn1t7GsZ0YPIdO-l9CK8kvPw9ndxH8 | Dryad Digital Repository, 10.5061/dryad.1rn8pk0vd |
| Nguyen Ba AN, Lawrence KR, Rego-Costa A, Gopalakrishnan S, Temko D, Michor F, Desai MM | 2021 | Barcoded Bulk QTL mapping reveals highly polygenic and epistatic architecture of complex traits in yeast | https://www.ncbi.nlm.nih.gov/bioproject/PRJNA767876 | NCBI BioProject, PRJNA767876 |

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

## Appendix 1

## Genotype inference

### Probabilistic genotype imputation

In humans, plants, and other species with large genomes, genome information for GWAS or QTL mapping has typically been obtained by microarray sequencing of a set of tagging SNPs (although whole-exome sequencing by target, capture methods is becoming more common). In yeast, because the genome is relatively small (12 Mb) and the SNP density in crosses can be high (~1 SNP per short read), whole-genome sequencing is an attractive option. Previous studies aimed for at least ~10X coverage per individual to minimize the impact of sequencing errors (*She and Jarosz, 2018*). However, at the scale of 100,000 individuals, sequencing at 10 X coverage would require over 50 billion reads. Even with state-of-the-art NGS platforms such as the Illumina Novaseq, this scale of sequencing is cost-prohibitive.

Fortunately, the known genetic structure of our cross allows us to infer segregant genotypes very accurately with significantly less than 10 X coverage. This is possible because we know a priori the location and identity of all variants from deep sequencing of the parents, so we can fully determine each segregant genotype if we can infer where the recombination breakpoints occur. In our F1 individuals, there are only on average ~50 breakpoints per individual, meaning that there are on average ~1000 SNPs in each haplotype block. Thus to accurately reconstruct the location of recombination breakpoints, we can sequence some smaller fraction of SNPs and then impute full genotypes (*Broman et al., 2003*). This approach is similar in spirit to methods for genome imputation from SNP genotyping arrays used in humans and other organisms. However, we note that each individual is observed at a *random* subset of SNPs (rather than observing the same tagging subset across all individuals), and due to random variation in sequencing coverage, some individuals are sequenced nearly completely while others are sequenced more sparsely. In this section, we describe our method for genome imputation, validate its accuracy, quantify the extent of genome uncertainty and errors, and discuss the impact of this uncertainty on QTL inference.

### Variant calling

Our first task is to identify the SNPs present in the parental cross. We sequenced both parental strains to about 100 X coverage using conventional Illumina Nextera sequencing. Polymorphisms were identified using the breseq pipeline against the reference BY4741 genome (*Engel et al., 2014*). We chose to exclude indels, as their precise location, especially in homopolymeric regions, does not always map uniquely in read aligners. This produced an initial set of 44,650 variants. We further required that polymorphisms were reliably mapped by our sequencing approach (aligning to a single location in the reference genome) and did not show evidence of read bias in the segregant pool. This produced a list of 42,661 SNPs between the two parental strains.

In addition, pilot experiments showed evidence of chromosomal translocations, primarily near telomeres, when comparing the BY and RM parents. Such translocations or duplications of large (≥1 kb) regions would lead to ambiguities in read mapping, genome inference, and QTL mapping. To identify these problematic regions, we conducted long-read sequencing of both parental strains using the Nanopore MinION platform (FLO-MIN106 flow cell and kit SQK-LSK109). For the RM (BY) strain, we obtained an estimated N50 of 22 kb (17 kb) with 360 X (760 X) coverage of the BY4741 reference genome. We performed de novo assembly using Canu (*Koren et al., 2017*) with default settings. The Canu assemblies each contained 16 contigs that mapped unambiguously to the 16 chromosomes in the reference genome. We then identified regions larger than 1000 bp that mapped to two different positions with greater than 90% sequence identity (either a reference region mapping to two or more contig regions, or a contig region mapping to two or more reference regions). Any SNPs from the original list that were found in these problematic regions were removed, resulting in a final curated list of 41,594 SNPs. We then generated an RM reference genome sequence by introducing these SNPs into the BY reference sequence. The final list of SNPs can be found in *Supplementary file 1*.

Once this list of variants is established, we can call variants for the F1 segregants. Paired-end reads obtained from sequencing with indexed Tn5 first contain the L and R index, which correspond to a specific well within a 384-well plate, and the constant mosaic end sequences. These are first parsed and removed. Reads are then processed with trimmomatic v.035 (*Bolger et al., 2014*).

Trimmed reads are then aligned to both BY and RM reference sequences using bowtie2 (**Langmead and Salzberg, 2012**) with the following commands: `bowtie2 --local -L 20 --ma 1 -mp 3 np 0 --rdg 2,3 --rfg 2,3 --ignore-quals -i S,1,0.25 --score-min L,1,0.75`. We verify that reads align at a single location in both references and indicate the same allele in both references. From this, we produce a genotyping file that contains, for each SNP for each individual, the number of reads calling the BY and RM allele.

## Probabilistic genotyping with an error-correcting Hidden Markov Model

In the absence of sequencing error and with complete coverage, the SNP genotype would be completely known. However, as discussed above, the sequencing cost for high-coverage whole-genome sequencing of 100,000 individuals is prohibitive. Instead, we impute SNP genotypes from low-coverage whole-genome sequencing, using a model that incorporates information about recombination rates and sequencing error rates.

To implement this imputation approach, we use a hidden Markov model (HMM) framework where the true genotypes $g_{i,k}$ are unobserved hidden states. For each chromosome, our model takes the form of a chain across all SNP positions (loci), where there are two possible hidden states (BY and RM) at each locus. Our desired output is the probability that each SNP has the RM (versus BY) parental genotype in each segregant, given the reads at all marker loci and the parameter values (recombination rate and sequencing error rates). In other words, for segregant i at locus $k$ we want to infer:

$$p(g_{i,k} = \text{RM}|\text{reads, parameters}) \equiv \pi_{i,k}. \tag{A1-1}$$

Transition probabilities between hidden states at two neighboring loci are determined by the recombination rate and the inter-SNP distance in basepairs. For two loci $k$ and $k+1$ separated by a distance $d$ basepairs, the transition matrix is as follows:

$$T_{k \rightarrow k+1} = \begin{array}{c} RM \\ BY \end{array} \begin{pmatrix} \overset{RM}{1 - r(d)} & \overset{BY}{r(d)} \\ r(d) & 1 - r(d) \end{pmatrix} \tag{A1-2}$$

where for a recombination rate $\rho$ in centimorgans per basepair,

$$r(d) = \frac{1}{2} \left( 1 - e^{-2d\rho/100} \right). \tag{A1-3}$$

Note that this transition matrix is different for each pair of loci $k, k+1$ but the same across all individuals.

Our observable quantity is not the true state of the genome, but rather read counts for the BY and RM allele at each locus. These are integer numbers rather than binary measurements, they can vary in magnitude between individuals depending on the sequencing coverage obtained, and they can disagree with the underlying state due to sequencing errors. We thus model the emission process as follows. For individual i at locus $k$, given that we observe $N$ total reads, the probability that $N - m$ are correct and are incorrect (representing base substitution errors) is given by a binomial distribution:

$$p_{i,k}(m|N) = \binom{N}{m} (p_{\text{err},1})^m (1 - p_{\text{err},1})^{N-m}. \tag{A1-4}$$

where $p_{\text{err},1} = 0.01$ represents the probability of single-base substitution errors. If we define $m$ to represent the number of sequencing reads with the RM allele, this gives the emission probabilities for the two hidden states:

$$O_{i,k}(m|N, g_{i,k} = RM) = \binom{N}{m} (1 - p_{\text{err},1})^m (p_{\text{err},1})^{N-m},$$
$$\tag{A1-5}$$

$$O_{i,k}(m|N, g_{i,k} = BY) = \binom{N}{m} (p_{\text{err},1})^m (1 - p_{\text{err},1})^{N-m}. \tag{A1-6}$$

This captures the effects of single-base substitution errors: for a single BY read among a long RM haplotype block, the probability for an erroneous emission will outweigh the probability of two close recombination events, so the model will tend to correct the error.

However, there is an additional source of error in Illumina sequencing that is especially prevalent on the Illumina NovaSeq platform: index-swapping, where a full read is assigned to the wrong sample. In this case, a read containing multiple SNPs that are observed as BY may be assigned to another individual that has a RM haplotype block around that locus. This signature –- errors in two or three sequential SNPs within the length of one read –- is not captured by the above model, which will often assign spurious breakpoints in such a situation.

To address this, one approach would be to structure the HMM as a chain of reads rather than a chain of loci, and account for index-swapping with a probability that an entire read is an error. However, this introduces significant computational complexity. Instead, motivated by the fact that in our cross we have at most a few SNPs per read, we adopt an alternative and approximately equivalent approach. Specifically, we expand our HMM to include two additional hidden states, which we term the error chains. An individual in an RM haplotype block will enter the RM error chain with a rate given by $p_{\text{err},2} = 0.01$, the probability for index-swapping. While in the RM error chain states, it will emit as if it were in the BY chain (i.e. more BY than RM reads), although its true genotype state has not changed. The rate to return from the RM error chain to the RM chain, $p_{\text{ret}} = 0.3$, is high enough that an individual will tend to return after a small number of loci (roughly as many as tend to occur in one read). The BY error chain follows a similar pattern, and other transitions are forbidden. The new transition matrices are

$$T_{k \to k+1} = \begin{array}{c} \\ \text{RM} \\ \text{BY} \\ \text{RM-err} \\ \text{BY-err} \end{array} \begin{array}{cccc} \text{RM} & \text{BY} & \text{RM-err} & \text{BY-err} \\ \begin{pmatrix} 1 - r(d) - p_{\text{err},2} & r(d) & p_{\text{err},2} & 0 \\ r(d) & 1 - r(d) - p_{\text{err},2} & 0 & p_{\text{err},2} \\ p_{\text{ret}} & 0 & 1 - p_{\text{ret}} & 0 \\ 0 & p_{\text{ret}} & 0 & 1 - p_{\text{ret}} \end{pmatrix} \end{array} \tag{A1-7}$$

and the emission probabilities for the error chains are given by

$$O_{i,k}(m|N, \text{RM-err}) = O_{i,k}(m|N, \text{BY}), \tag{A1-8}$$

$$O_{i,k}(m|N, \text{BY-err}) = O_{i,k}(m|N, \text{RM}). \tag{A1-9}$$

This defines our HMM. We use the forward-backward algorithm to calculate forward and backward probability vectors for each individual i at each locus $k$ along each chromosome, from $k = 0$ to $k = K$:

$$f_{i,0:k} = \begin{pmatrix} f_{i,0:k}^{\text{RM}} & f_{i,0:k}^{\text{BY}} & f_{i,0:k}^{\text{RM-err}} & f_{i,0:k}^{\text{BY-err}} \end{pmatrix} = f_{i,0:k-1} T_{k-1 \to k} O_{i,k}, \tag{A1-10}$$

$$f_{i,0} = \begin{pmatrix} 0.49 & 0.49 & 0.01 & 0.01 \end{pmatrix}, \tag{A1-11}$$

$$b_{i,k:K} = \begin{pmatrix} b_{i,k:K}^{\text{RM}} \\ b_{i,k:K}^{\text{BY}} \\ b_{i,k:K}^{\text{RM-err}} \\ b_{i,k:K}^{\text{BY-err}} \end{pmatrix} = T_{k \to k+1} O_{i,k+1} b_{i,k+1:K}, \tag{A1-12}$$

$$b_{i,K} = \begin{pmatrix} 1 \\ 1 \\ \vdots \\ 1 \\ 1 \end{pmatrix}.$$ (A1-13)

These are then used to calculate the posterior probabilities for each individual to have the RM genotype:

$$\pi_{i,k} = \frac{f_{i,0:k}^{RM} b_{i,k+1:K}^{RM} + f_{i,0:k}^{RM\text{-}err} b_{i,k+1:K}^{RM\text{-}err}}{f_{i,0:k}^{RM} b_{i,k+1:K}^{RM} + f_{i,0:k}^{BY} b_{i,k+1:K}^{BY} + f_{i,0:k}^{RM\text{-}err} b_{i,k+1:K}^{RM\text{-}err} + f_{i,0:k}^{BY\text{-}err} b_{i,k+1:K}^{BY\text{-}err}}.$$ (A1-14)

We infer this quantity for all 99,950 individuals for all $L = 41,594$ SNPs across the 16 chromosomes. For each individual, we thus produce a vector $\vec{\pi}_i = (\pi_{i,0}, \ldots, \pi_{i,L})$ which we refer to as the 'probabilistic genotype' for that individual, using the convention that one indicates the RM allele and 0 the BY allele. We discuss the statistical implications of using real-valued rather than binary genotypes for QTL mapping analysis in Appendix 2.

The description above assumes known values for the parameters: the error rates $p_{err,1}$ and $p_{err,2}$, the error chain return rate $p_{ret}$, and the recombination rate $\rho$. While we have prior estimates for these values informed by our understanding of sequencing errors and recombination in this cross, we can also learn these parameters from the data. In particular, we expect the recombination rate to vary across genomic locations: for example, rates should increase near recombination hotspots and decrease near centromeres or telomeres. To allow for position-specific recombination rates, we can estimate and update the transition matrix using the Baum-Welch algorithm (*Baum et al., 1970*). Because we observe of order $10^6$ breakpoints in our pool, we can produce a precise recombination map that captures local variation at many scales. In practice, because our genotypes are well-constrained by sequencing already (see below), successive updates of recombination parameters do not significantly change the genotypes inferred for most segregants. Thus for the results below, we use an average recombination rate for each chromosome obtained from a single Baum-Welch iteration.

## Validation and error metrics

To evaluate the accuracy of our genome inference, we used several metrics. First, from the inferred genotypes themselves, we can obtain an estimate of our uncertainty: for loci whose posterior probabilities are near 0 or 1, there is strong evidence for a particular allele from the HMM, while for those whose posterior probabilities are near 0.5, there is very weak evidence for either allele. While these are not interpretable in the sense of p-values or other established statistics, we can define an average uncertainty metric for each individual, inspired by the binomial variance:

$$U_i \equiv \frac{4}{L} \sum_{k=0}^{L} \pi_{i,k} \left(1 - \pi_{i,k}\right).$$ (A1-15)

The factor of 4 ensures that this metric ranges from 0 to 1 (i.e. for completely uncertain genotyping, $\pi_k = 0.5$ for all $k$, we have $U_i = 1$.) For each individual, we also know its coverage (average number of sequencing reads per SNP) and coverage fraction (fraction of SNPs with at least one sequencing read). We expect that our HMM estimates should have very low uncertainty when the coverage/coverage fraction is high (deeply sequenced genomes), and the uncertainty should approach the maximum of 1 when the coverage approaches 0 (no sequencing information). *Appendix 1—figure 1* below shows inferred genome uncertainty as a function of coverage for all segregants, indicating that our inferred genomes maintain very low uncertainty even when the coverage is as low as 0.1 X.

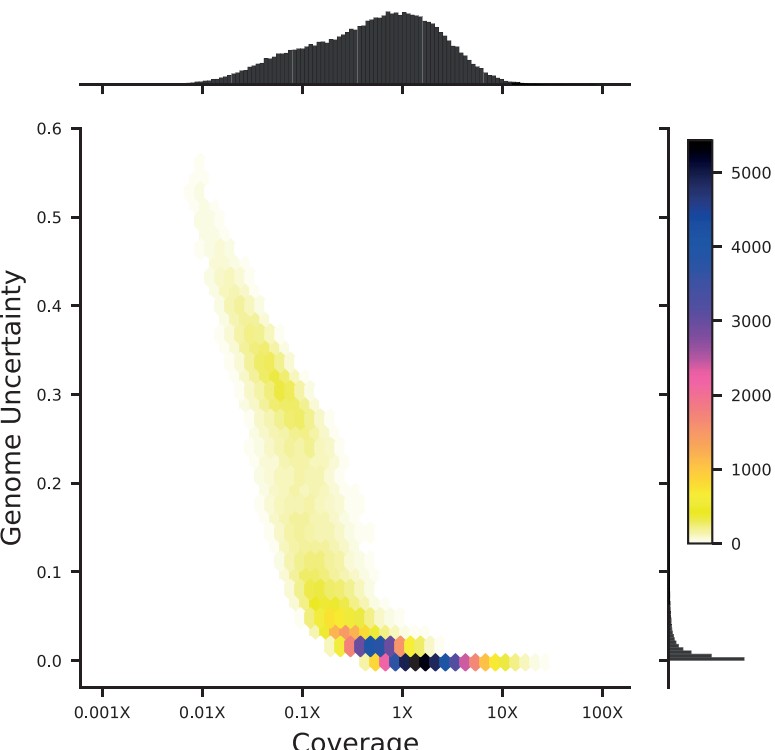

**Appendix 1—figure 1.** Density plot of genome uncertainty as a function of coverage (on a log-10 scale) for all 99,950 segregants. Marginal distributions show the coverage (above) and uncertainty (left) histograms for all segregants.

In addition to uncertainty, we would like a metric for accuracy, or the difference between the genotype inferred by our model from low-coverage data and that inferred from high-coverage data, which we presume to represent true genotypes. To estimate this, we selected several segregants with different coverages from our initial sequencing, ranging from 0.15 X (0.1 coverage fraction) to 2.3 X (0.73 coverage fraction), and then individually re-sequenced them to >10 X coverage ( > 0.99 coverage fraction). We can subtract the original low-coverage genotype from that obtained from re-sequencing to obtain the error at each locus: if the low-coverage inference is entirely wrong, the difference will equal 1, whereas if the low-coverage inference had no information, the difference will be about 0.5. For each individual we can make a histogram of this value over all loci, as shown in *Figure 1C*.

In addition, we can use the resequencing data to check the calibration of our posterior probabilities: for loci with posterior probabilities around, say, 0.4, we expect the high-coverage sequencing to show that ~40% are indeed RM while the remaining ~60% are BY. As we discuss in Appendix 3, we can leverage probabilistic genotype values to improve QTL inference, but this requires accurate calibration. To show this, we choose the segregant with initial sequencing coverage of 0.15 X and bin its genotype posterior probabilities into ten equal bins (0–0.1, 0.1–0.2, etc; as this individual has relatively low coverage, there are >50 loci in all intermediate bins). For each bin, we count the fraction of loci that are revealed to be RM in the high-coverage sequencing data (defined as a posterior probability ≥0.99). These results are plotted in *Appendix 1—figure 1*. We see that the posterior probabilities are well-calibrated overall, even at intermediate values.

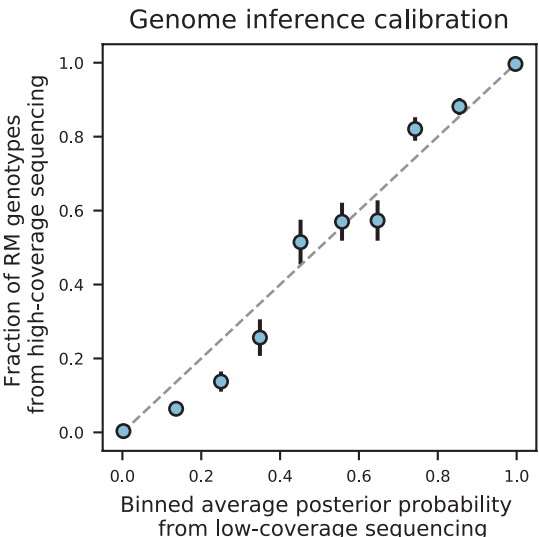

**Appendix 1—figure 2.** Genome inference calibration. For a segregant initially sequenced to 0.15 X coverage, loci were binned into 10 equal bins according to their genotype posterior probability. For each bin, we plot the average posterior probability for each bin against the fraction of those loci that were found to be RM in high-coverage sequencing data (i.e. showed a posterior probability of ≥0.99). The dashed grey line represents the expectation for perfectly calibrated posterior probabilities. Error bars are given by $\sigma = \sqrt{\frac{\hat{p}(1-\hat{p})}{n}}$ where $\hat{p}$ is the fraction of RM loci and $n$ is the total number of loci in the bin.

These results accord with our intuition: all uncertainty in the inferred genotypes is concentrated in the region around each breakpoint, while far away from breakpoints, the genome is inferred very accurately even when observing only about 1 in 10 variants. Since the uncertain loci near breakpoints represent a small minority of all variants, the genome inference overall is quite accurate.

## Impact on QTL mapping

Despite the overall accuracy, of course it remains the case that some fraction of segregants will have substantial noise in their genotypes, and that many segregants will have substantial noise in the regions very close to recombination breakpoints. Thus it is important to quantify what impact this genotyping noise may have on our QTL inference and fine-mapping procedures. Since breakpoints are randomly distributed across individuals, we might expect that poor genotyping at a fraction of loci will add noise without introducing systematic bias. The fact that some segregants are poorly genotyped overall may be due to random effects of tagmentation or sequencing; alternatively, some segregants that are less fit in rich media will have smaller colonies at our time of processing and thus be underrepresented in our fitness assays as well as poorly covered in genotype sequencing. Thus, genotype and phenotype noise may not be purely uncorrelated, although we expect a relatively small fraction of segregants to exhibit this effect. In this section, we present some empirical characterizations of the influence of genotype noise on model inference and performance. Detailed descriptions of our approach to QTL inference and fine-mapping can be found in Appendix 3.

Our approach is to run our QTL inference pipeline on subsets of segregants that have very high and very low sequencing coverage, and to contrast the results. In particular, we divide our segregants into deciles by coverage fraction (ten subsets of 9995 segregants each). Of course, our inference is less powerful using a tenfold smaller sample size, but we expect the relative differences between the lowest and highest deciles to be representative of the differences we would observe if our entire segregant panel was sequenced to the equivalent coverage fractions. The lowest decile has a maximum coverage of 0.05 X (coverage fraction 0.03), while the highest decile has a minimum coverage of 3 X (coverage fraction 0.79). From each of these two sets, one-tenth of segregants (999 individuals) are held out for testing, while the remaining 90% are used to infer QTL models for a selection of 6 traits.

The results are shown in **Appendix 1—figure 3** and **Appendix 1—figure 4**. First, in **Appendix 1—figure 3**, we see that models inferred on low-coverage training sets have significantly fewer inferred

QTL (left panel), but their performance in predicting phenotypes on a high-coverage held-out test set is only marginally reduced (center panel). (We also note that compared to the full panel of 99,950 segregants, the high-coverage panel of 9995 finds fewer QTL but achieves similar performance.) In contrast, the performance of the same models in predicting phenotypes on a low-coverage held-out test set is extremely poor (right panel), regardless of the coverage of the training set. Second, we see in *Appendix 1—figure 4* that models trained on high-coverage segregants show smaller credible intervals overall compared to models trained on low-coverage segregants. (We note that the full panel of 99,950 segregants has even smaller credible intervals than the high-coverage panel of 9995).

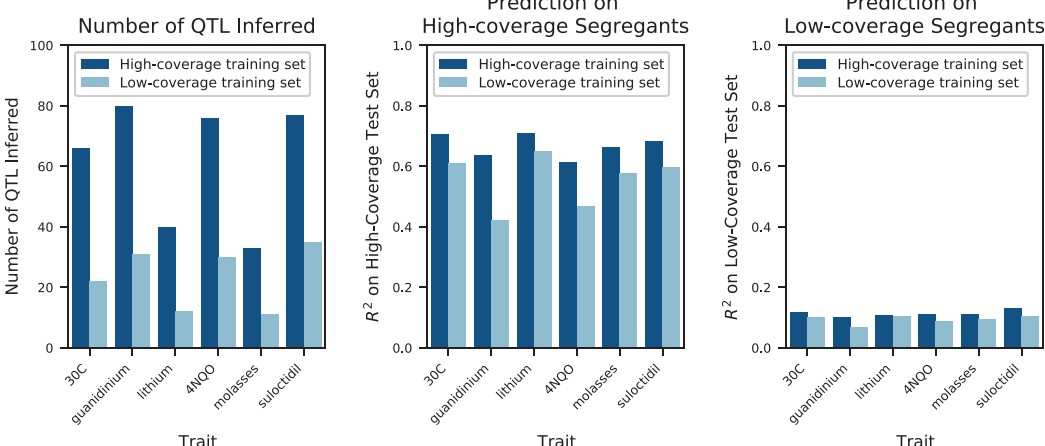

**Appendix 1—figure 3.** Impact of low vs high coverage on both inference and performance of QTL models. Left: numbers of QTL inferred for a selection of 6 traits, using either the highest-coverage decile (dark blue, coverage fraction >0.79, coverage >3X) or lowest-coverage decile (light blue, coverage fraction <0.03, coverage <0.05X) as the training set. Center: performance of the QTL models inferred with high-coverage (dark blue) or low-coverage (light blue) training sets on a high-coverage test set. Right: performance of the QTL models inferred with high-coverage (dark blue) or low-coverage (light blue) training sets on a low-coverage test set.

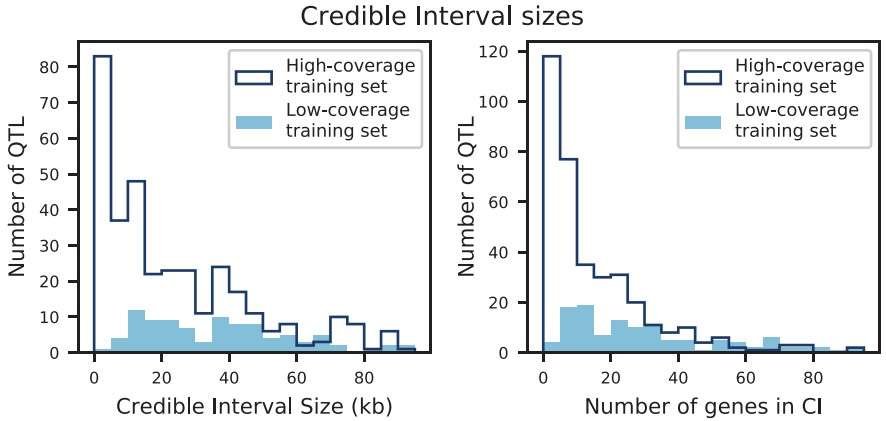

**Appendix 1—figure 4.** Credible interval sizes in basepairs (left) or number of contained genes (right), for models inferred from high-coverage training data (dark blue) or low-coverage training data (light blue). Distributions represent QTL from all six traits.

Thus, we observe that extremely noisy genotype data, such as that in our lowest-coverage decile, can reduce our power both to detect QTL and to fine-map their locations precisely. However, it does not significantly impact predictive performance, and it does not induce systematic biases in QTL positions or effect sizes. In this way, the impact of genotyping noise is very similar to the impact

of reducing sample size, as each poorly-genotyped individual contributes less to power than each well-genotyped individual. From *Appendix 1—figure 1*, we see that the majority of our panel has essentially negligible genotyping noise, with only the bottom two deciles demonstrating significant uncertainty and error. Thus, at our current coverage distribution for our full panel, we expect our 'effective sample size' to be reduced by at most 10–20% compared to high-coverage sequencing of all individuals. Thus, as expected, our sequencing approach achieves a significant cost savings for only a modest decrease in power.

We also find that coverage has a much more significant impact on measuring the performance of inferred QTL models than on inferring those models. This is expected: we leverage the power of large sample sizes to infer accurate QTL models, even from noisy data; but a highly accurate model will still have poor predictive performance on individuals with uncertain or incorrect genotypes. We discuss the implications of this for variance partitioning and measuring model performance in Appendix 4.

Finally, we note from *Figure 1—figure supplement 3* that certain regions in the genome immediately surrounding our selection markers exhibit strongly biased allele frequencies. Specifically, the region near the MAT locus (for mating type selection) on chromosome III, the region near the HO locus (for barcode retention selection) on chromosome IV, and the region near the CAN1 locus (for haploid selection) on chromosome V show allele frequencies significantly different from 50%. The presence of these regions can impact our QTL inference in two ways. First, if true QTL are located close enough to the selective markers that their allele frequencies are very different from 50%, our power to infer them is reduced. Second, we know that small fractions of segregants can "leak" through our selection process. These segregants would be mating type MATα rather than MATa, and/or not have barcodes, and/or be unsporulated diploids rather than haploids. Segregants without barcodes would not appear in barcode amplicon sequencing for barcode association or fitness assays, so we know there can be at most a few thousand such individuals, and they will not affect phenotyping or QTL inference. Diploids or MATα individuals, on the other hand, could show strong fitness benefits in some assay environments. We estimate that diploid and MATα individuals make up ~0.7% of our segregant pool, using a combination of computational methods (examining the marker loci in the genotype data) and biological assays (PCR of the MAT locus, growth in selection media, and examination of morphology). Despite these low numbers, our inference is sensitive enough to detect strong discrepancies in the fitnesses of these individuals, and such effects are assigned to the only locus that is systematically different in these individuals – the marker loci. Thus, in some environments we do observe QTL effects mapped to our marker loci or immediately neighboring genes, which we expect to be an artifact of a small fraction of leaker individuals rather than biologically meaningful findings.

## Appendix 2

### Phenotype inference

#### Likelihood model for time-dependent barcode frequencies

Our raw phenotype data consists of a set of integer barcode read counts over time for each individual in each replicate assay. That is, we observe some set, $n_{i,r}(t)$, of integer numbers of reads that correspond to segregant i in assay replicate $r$ at timepoint $t$. These read counts reflect the competition between segregants to leave offspring in each daily dilution. We define the phenotype of each segregant genotype, i, in a given environment as its fitness, $y_i$. This fitness is in turn defined as the average number of descendants an individual of that genotype will leave in each daily dilution minus the population average (which is by definition 1). If $y_i > 0$, then segregant i is more fit than the population average and its frequency will tend to increase with time; if $y_i < 0$, it is less fit and will tend to decline in frequency. Our goal is then to infer the map between genotype and fitness.

As mentioned above, we could in principle conduct our QTL mapping inference directly from the raw phenotype data (i.e. $n_{i,r}(t)$) by constructing a likelihood function that predicts the probability of a set of read counts given a proposed genetic architecture of the phenotype (which implies a set of predicted fitnesses $y_i$). However, this is impractical in practice. Thus we instead use a two-step approach. First, in this section, we infer the maximum-likelihood fitness of each segregant, $\hat{y}_i$, from the integer barcode read counts, $n_{i,r}(t)$. Next, in later sections, we infer the genetic basis of these fitnesses.

To infer the fitness of each segregant, we begin by defining a model for how the frequency, $f_{i,r}$ of segregant i in replicate $r$ depends on time. Assuming a deterministic growth process, our definition of fitness implies that

$$\frac{\partial f_{i,r}}{\partial t} = f_{i,r} \cdot (y_i - \bar{y}). \tag{A2-1}$$

Here, $\bar{y}(t) = \sum_i f_{i,r}(t)y_i$ is the population mean fitness of replicate $r$ at time $t$. In principle, genetic drift could also affect these frequencies, but because we ensure that the bottleneck sizes of our daily dilutions are sufficiently large (~1000 cells per barcode lineage on average), we expect these effects of drift to be negligible (see **Nguyen Ba et al., 2019** for a discussion of this approximation). Note that we implicitly assume here that the fitness of each genotype does not depend on the frequencies of the other genotypes (i.e. no interaction effects that would lead to frequency-dependent selection). All alleles are present at frequencies close to 50% in our pool (except for those very near selected marker loci), and these frequencies do not change significantly over the course of the fitness assays (except for a few large-effect loci). Thus the effect sizes we infer for QTL are representative of their effect at ~50% frequency. We cannot rule out that some QTL may have frequency-dependent effects far away from 50%, but these are not expected to influence our results, and a detailed investigation of frequency dependence of QTL is beyond the scope of the current study.

If we define $f_{i,r}^0$ to be the initial frequency of lineage i at time $t = 0$ in replicate $r$, then the solution to **Equation A2-1** is given by:

$$f_{i,r}(t) = \frac{f_{i,r}^0 e^{y_i t}}{\sum_j f_{j,r}^0 e^{y_j t}}. \tag{A2-2}$$

Note that the phenotypes of all individuals in one assay are not independent, as the fitness of each is defined relative to the population mean. Thus we must jointly infer the fitnesses of all segregants simultaneously, and these fitnesses are only meaningful in a relative sense (i.e. we can shift all fitnesses by a constant without affecting any predictions). For computational simplicity, a useful variable transformation is setting

$$\tilde{f}_{i,r} = \log\left(f_{i,r}/f_{\text{ref}}\right). \tag{A2-3}$$

This transformed variable represents frequency relative to a reference lineage $f_{\text{ref}}$, where $\tilde{f}_{\text{ref}}^0 = 0$ and we constrain $y_{\text{ref}} = 0$ to solve the system uniquely. In each assay, we assign the lineage with highest total read counts to be the reference lineage. Expected frequencies at time $t$ can now be written as:

$$\tilde{f}_{i,r}(t) = \frac{e^{\tilde{f}^0_{i,r}+y_i t}}{\sum_j e^{\tilde{f}^0_{j,r}+y_j t}}. \tag{A2-4}$$

Note here that we allow each lineage to have different initial frequencies $f^0_{i,r}$ in different replicates, but the fitness $y_i$ is constrained to be the same across replicates.

Using this model, we wish to maximize the overall likelihood of the data with respect to the fitnesses of all lineages. However, we do not observe the frequencies directly, but instead observe them through the read counts $n_{i,r}(t)$ at each timepoint in each replicate. Because these read counts can be thought of as arising from a sampling process, we might expect the read counts to be multinomially distributed. However, in practice the noise in this process (arising from a combination of DNA extraction, PCR amplification, sequencing, and other effects) often tends to be overdispersed, and can vary substantially between assays (*Nguyen Ba et al., 2019*). One simple and popular method to account for overdispersion is found in the quasi-likelihood framework (*McCullagh, 1983*) because simple quasi-likelihood models add a single multiplicative parameter that inflates the variance function of well-known distributions within the exponential family. In this framework, we cannot write explicitly a functional form for the distribution that represents the data. Nevertheless, quasi-likelihood approaches remain useful because least-squares algorithms to optimize generalized linear models, such as iterative reweighted least-squares, only require the mean and variance function. When the variance function of a probability distribution is simply inflated by a constant factor, this factor cancels out during optimization, and therefore approaches that obtain maximum likelihood estimates also obtain the maximum quasi-likelihood estimates. If we were to write a sampling process for the number of reads without any overdispersion, we could consider data being multinomially distributed with mean $\bar{n}_{i,r}(t) = \tilde{f}_{i,r}(t) \sum_i n_{i,r}(t)$ and variance $\sigma^2_i = \bar{n}_{i,r}(t)(1 - \tilde{f}_{i,r}(t))$. Thus, for the quasi-likelihood approach, we can modify the variance to be $\sigma^2_i = \psi \cdot \bar{n}_{i,r}(t)(1 - \tilde{f}_{i,r}(t))$, with $\psi$ the overdispersion parameter. This overdispersion parameter does not alter the parameter estimates but simply inflates their standard error ($\hat{y}_i \rightarrow \mathcal{N}(y_i, \psi\sigma^2_i)$), and can be fit from the data for each assay (one parameter per environment).

The overdispersion parameter can be estimated in several ways. A simple and consistent estimator relies on the Pearson goodness-of-fit (*McCullagh and Nelder, 1989*), which estimates $\psi$ as:

$$\hat{\psi} = \frac{\chi^2}{\text{dof}}, \tag{A2-5}$$

where $\chi^2$ is the usual Pearson goodness-of-fit statistic and dof is the number of degrees of freedom. However, this moment estimator can be biased when the asymptotics of the Pearson goodness-of-fit are not met, specifically when many lineages have low read counts.

We turn instead to a deviance based estimator. If the data were not overdispersed, then it is well-known that the likelihood ratio test statistic (or the deviance) between a parameter-rich statistical model and a nested (more constrained) model is asymptotically distributed as a chi-squared distribution with the usual degrees of freedom:

$$D = LRT = -2 \cdot [\log L_0 - \log L_1] \sim \chi^2_{\text{dof}}. \tag{A2-6}$$

When data is overdispersed with overdispersion factor $\psi$, it is instead the scaled deviance that follows a chi-squared distribution, with:

$$D^* = \frac{D}{\psi} \sim \chi^2_{\text{dof}}. \tag{A2-7}$$

This yields another estimate of $\psi$:

$$\hat{\psi} = \frac{D}{\text{dof}} = \frac{-2[\log L_0 - \log L_1]}{\text{dof}}, \tag{A2-8}$$

under the assumption that deviations between observed and expected are only attributed to the overdispersed variance, and not to misspecifications of the model (due to an omitted variable, for example). Here, our restricted model is our final regression model ($L_0$ is the likelihood of the data under the optimized model described above) while the larger model is one that can fit the observed

read counts perfectly ($L_1$ is the likelihood of the data under the same multinomial sampling noise model, but with an independent parameter for every lineage at every timepoint in every replicate). This estimate is typically very similar to the Pearson goodness-of-fit estimator, but less biased when the number of low-read-count lineages is large. The estimated scaling factor $\hat{\psi}$ obtained in this way will be used to obtain correct estimates of the standard errors in the lineage fitnesses, which are used in Appendix 4, and is reported in **Supplementary file 6**.

Under this multinomial model, the optimized estimates for the lineage fitnesses in the quasi-likelihood framework are the same as optimizing the following likelihood function of all of the read count data given the parameters:

$$\mathcal{L}(\{n\}|\{y,\tilde{f}^0\}) = C \cdot \prod_{t,r,i}^{T,R,N} \left(\tilde{f}_{i,r}(t)\right)^{n_{i,r}(t)}, \tag{A2-9}$$

where the product is taken over all timepoints $t$, replicates $r$, and individuals i, and $C$ represents a constant factor. Again, here $n_{i,r}(t)$ represents the barcode read counts observed for lineage   in replicate assay $r$ at timepoint $t$, and $\tilde{f}_{i,r}(t)$ represents the expected frequency of lineage   in replicate assay $r$ as calculated from **Equation (A2-4)** for the parameters $y_i$ and $\tilde{f}^0_{i,r}$.

Our goal is to optimize this likelihood function to find the maximum-likelihood estimates of the fitnesses, $\hat{y}_i$. To do so, we use a preconditioned nonlinear conjugate gradient approach (**Hager and Zhang, 2006**). We first calculate the partial derivatives of $\log \mathcal{L}$:

$$\frac{\partial \log \mathcal{L}}{\partial y_i} = \sum_{t,r}^{T,R} t \left[n_{i,r}(t) - n_{\text{tot},r}(t) \cdot \tilde{f}_{i,r}(t)\right], \tag{A2-10}$$

$$\frac{\partial \log \mathcal{L}}{\partial \tilde{f}^0_{i,r}} \sum_{t}^{T} \left[n_{i,r}(t) - n_{\text{tot},r}(t) \cdot \tilde{f}_{i,r}(t)\right], \tag{A2-11}$$

where $n_{\text{tot},r}(t) = \sum_i n_{i,r}(t)$ is the total number of reads from timepoint $t$ in replicate assay $r$. As a preconditioner, we use an approximation of the inverse Hessian, which can be assumed to be approximately diagonally block dominant when the number of lineages is large. The dominant second derivatives used to approximate the inverse Hessian are:

$$\frac{\partial^2 \log \mathcal{L}}{\partial y_i^2} = -\sum_{t,r}^{T,R} t^2 \cdot n_{\text{tot},r}(t)\tilde{f}_{i,r}(t)\left(1 - \tilde{f}_{i,r}(t)\right), \tag{A2-12}$$

$$\frac{\partial^2 \log \mathcal{L}}{(\partial \tilde{f}^0_{i,r})^2} = -\sum_{t}^{T} n_{\text{tot},r}(t)\tilde{f}_{i,r}(t)\left(1 - \tilde{f}_{i,r}(t)\right), \tag{A2-13}$$

$$\frac{\partial^2 \log \mathcal{L}}{\partial y_i \partial \tilde{f}^0_{i,r}} = -\sum_{t}^{T} t \cdot n_{\text{tot},r}(t)\tilde{f}_{i,r}(t)\left(1 - \tilde{f}_{i,r}(t)\right). \tag{A2-14}$$

The algorithm is terminated when the gradient norm is sufficiently small, typically on the order of square root of machine-epsilon. Nonlinear conjugate gradient algorithms differ in their choice of the conjugate direction; we use the updates as in CG_DESCENT (**Hager and Zhang, 2013**).

## Imposing a prior to deal with low-frequency lineages

In some cases, the likelihood surface is poorly determined, for example when many lineages are present at low enough frequencies that they do not appear in the sequencing reads at later timepoints. These lineages tend to have relatively low fitness, because they are present at low frequencies and/or decline in frequency over time, but their fitness parameters (and error on these parameters) cannot be accurately determined. Their read count trajectories are consistent with a wide range of negative fitness values, including extremely large ones. We cannot simply remove these segregants from the phenotyping assay using a threshold on read counts, as this would introduce biases (because low-fitness segregants would be preferentially removed). Because the distribution of frequencies is approximately log-normal, it is also impractical to better sample these low-frequency lineages by simply increasing the number of reads.

When the proportion of lineages with sufficiently low fitness is very high, this introduces two problems. First, the fitness inference algorithm may never terminate at all, as the likelihood surface is flat around many possible values of estimated fitnesses while the norm of the first derivatives can remain high. Second, even if the inference terminates, the incorrect fitness estimates may bias the estimation of QTL effect sizes (if low-fitness segregants tend to have a given allele at a specific causal locus and fitness estimates of these segregants have consistently higher noise).

One approach to solving these problems is to impose a bias in the estimation of the fitness of lineages, informed by our expectations about the underlying genomic architecture of the trait. In a Bayesian framework, we would like to impose a prior on the population fitness distribution that will constrain fitness values for low-frequency (weak-evidence) lineages, according to our assumptions, but that will not strongly bias the fitness values for high-frequency (strong-evidence) lineages. In addition to allowing the inference to terminate, this framework would also provide an error estimate for each lineage that, for low-frequency lineages, is dominated by the prior, and for high-frequency lineages, is dominated by the read count evidence.

We now describe our implementation of this approach. We choose to use a Gaussian prior, under the assumption that most traits we consider are at least mildly polygenic, such that the distribution of fitnesses across a large number of segregants will be roughly Gaussian due to the central limit theorem. For each trait, the distribution of true lineage fitnesses will have some unknown mean and some unknown variance due to the genetic architecture of the trait: $y \sim \mathcal{N}(\bar{y}, \sigma_{\text{gen}}^2)$. Our maximum-likelihood estimates of lineage fitnesses from read count data, $\hat{y}$, will have the same mean but an increased variance due to the error in our fitness inference procedure: $\hat{y} \sim \mathcal{N}(\bar{y}, \sigma_{\text{tot}}^2) = \mathcal{N}(\bar{y}, \sigma_{\text{gen}}^2 + \sigma_{\text{err}}^2)$. To incorporate this Gaussian prior into our inference, we multiply our likelihood above (*Equation A2-9*) by the Gaussian term to obtain our posterior probability:

$$\log \mathcal{P}(\{n\}|\{y,\tilde{f}^0\}) = \sum_{t,r,i}^{T,R,N} n_{i,r}(t) \log\left(\tilde{f}_{i,r}(t)\right) - \frac{(y_i - \bar{y})^2}{2\sigma_{\text{tot}}^2} + \text{const.} \qquad \text{(A2-15)}$$

Maximum-likelihood estimates $\{\hat{y}, \hat{f}^0\}$ are obtained using conjugate gradient descent as described above. Since we do not know $\bar{y}$ and $\sigma_{\text{tot}}^2$ a priori, we choose initial estimates and iterate the fitness-inference algorithm until the mean and variance converge.

We now offer some intuition as to why this strategy is appropriate for our case and produces the desirable effects outlined above. We can first write down the standard error obtained for each lineage's fitness estimate, $\sigma_i^2$. This is done by evaluating the negative Hessian at the maximum likelihood estimates (i.e. the observed Fisher information):

$$-\frac{\partial^2 \log \mathcal{P}}{\partial y_i^2} = \sum_{t,r}^{T,R} t^2 \cdot n_{\text{tot},r}(t)\tilde{f}_{i,r}(t)\left(1 - \tilde{f}_{i,r}(t)\right) + \frac{1}{\sigma_{\text{tot}}^2} = \frac{1}{\sigma_i^2}. \qquad \text{(A2-16)}$$

As explained above, the standard errors need to be corrected for the overdispersion factor ($SE = \sqrt{\psi\sigma_i^2}$). Comparing to *Equation A2-12*, we see that the standard error for each lineage now comprises a balance of two terms. For lineages at very low frequencies $\tilde{f}_{i,r}(t)$, the second term (corresponding to the Gaussian prior) dominates. Since there is little information from read counts to constrain the fitness inference, the prior tends to pull the fitness of these lineages toward $\bar{y}$, such that they do not run away to extremely low values. For high-frequency lineages, on the other hand, the quantity $n_{\text{tot},r}(t)\tilde{f}_{i,r}(t)$ is substantial, so the first term will dominate over the second. These lineages are not strongly affected by the presence of the prior, because their large numbers of reads provide strong evidence for their fitness values.

## Simulation-based validation of the prior approach

This procedure inevitably introduces bias into the fitness estimates of low-frequency reads – indeed, the motivation here is to decrease variance in the weak-evidence fitness estimates at the expense of introducing bias. We expect this bias to have negligible impact on our QTL mapping inference only in the regime where the true underlying fitness distribution is sufficiently Gaussian (i.e. our choice of prior is appropriate) and where the proportion of low-frequency lineages is sufficiently small (i.e. the majority of lineages in our population are observed at high read counts). The assumption of a Gaussian fitness distribution, arising from a polygenic trait, also implies that both alleles at a causal locus will be distributed broadly across the full range of segregant fitnesses (i.e. there will be high

and low fitness segregants with both alleles at each causal locus). The inferred causal effect size will be less biased in this case, to the extent that the low-frequency, high-error lineages contain both alleles equally (rather than one allele preferentially). Of course, for traits that are bimodal or otherwise strongly non-Gaussian, or for a population with very poor phenotyping for most lineages, the Gaussian prior can introduce significant bias into the estimation of QTL effect sizes.

To demonstrate this, we conducted simulations of bulk fitness assays with varying true phenotype distributions and varying sequencing coverages. We chose three fitness distributions: first, Gaussian with mean 0 and standard deviation $\sigma = 0.04$, representing a genetic architecture with many small-effect QTL; second, a bimodal distribution with means ±0.1 and standard deviation $\sigma = 0.04$ for each, representing a genetic architecture with one strong QTL of effect 0.2 on a background of many small-effect QTL; and third, a distribution intermediate between these two, with means ±0.05 and standard deviation $\sigma = 0.04$ for each. These standard deviations are consistent with the scale we observe in our fitness assays, with 0.1 and 0.2 representing extremely strong-effect QTL. These distributions are shown in *Appendix 2—figure 1* (inset). From these distributions, 1,000 segregant fitnesses were sampled, with initial frequencies drawn from a log-normal distribution with parameters similar to our assay data (mean of 1/(number of lineages) and standard deviation ~0.5 decades). We simulated deterministic changes in frequency according to the model explained above, and performed multinomial sampling at 5 timepoints over 49 generations (as in our BFAs). We adjusted the depth of sampling to obtain mean coverages (sequencing reads per barcode per timepoint) of 1, 10, 100, and 1000. We then inferred fitnesses using same procedure as for the real data (described above) and examined the performance.

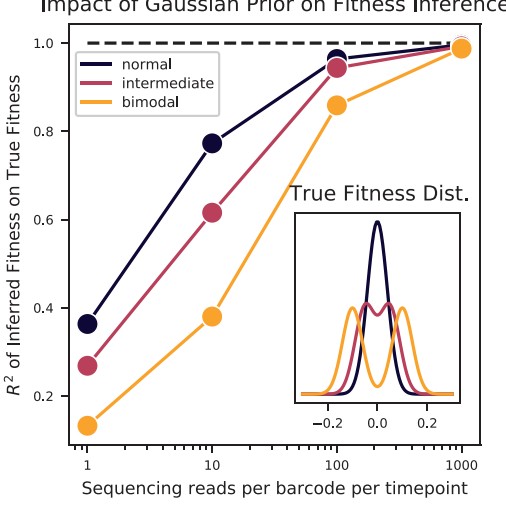
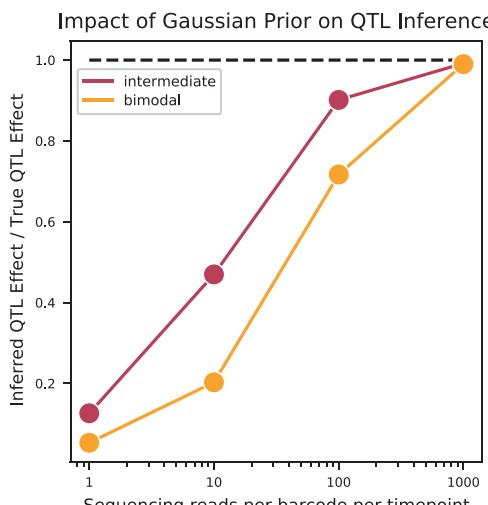

**Appendix 2—figure 1.** Bulk Fitness Assay simulations. Simulated phenotypes are drawn from the distributions shown in the left plot inset, for normal, bimodal, and intermediate distributions. Left: accuracy of fitness inference ($R^2$ of inferred fitnesses on true fitnesses) as a function of sequencing coverage, for the three distributions. Right: Inference of the strong QTL effect as a fraction of the true effect for the bimodal and intermediate distributions, as a function of sequencing coverage.

As we see in the left panel of *Appendix 2—figure 1*, the accuracy of our fitness inference (given by the $R^2$ between the inferred and true fitnesses) is strongly dependent on sequencing coverage for all distributions. The bimodal and intermediate phenotypes require more coverage to achieve the same accuracy as the normally distributed phenotype, but all distributions are inferred with high accuracy for sequencing coverage above 100 reads per barcode per timepoint. In addition, for the bimodal and intermediate distributions, we can attempt to infer the strong QTL effect from the peak separation in the inferred phenotype distribution. This inferred effect is plotted as a fraction of the true effect (0.1 for intermediate, 0.2 for bimodal) in the right panel of *Appendix 2—figure 1*. We can observe that the magnitude of the QTL effect is underestimated when sequencing coverage is low, as the Gaussian prior tends to merge the two peaks towards the mean, and this underestimation is more severe for the more strongly bimodal distribution. However, for sufficiently high coverage, the

true fitness distribution is almost perfectly inferred and so the effect estimation is similarly accurate. From this, and given our average coverage of 185 reads per barcode per timepoint, we conclude that fitness inference with the Gaussian prior will be largely accurate, with the caveat that extremely strong effects may be biased towards zero.

## Appendix 3

## QTL inference and analysis

### QTL mapping statistical analysis

The basic idea of QTL mapping is to statistically associate genotype to phenotype. This requires genotype information for a large number of individuals along with their corresponding phenotypes. As described in Appendix 1 and Appendix 2, both genotype and phenotype in our pool are estimated values rather than known values. Thus, in principle, it might be possible to perform QTL mapping from completely raw data, which for genotyping includes the read information for a random subset of the SNPs and for phenotyping includes the number of sequencing reads at each timepoint of the bulk fitness assays. However, for computational simplicity, we choose instead to perform QTL mapping on summary and sufficient statistics of these raw data. Briefly, we transform the raw genotyping reads into inferred probabilistic genotype values (Appendix 1), and we transform the raw read counts per timepoint into inferred fitness values and standard errors (Appendix 2). As described in this section and as demonstrated in simulations (Section A3-1.5), these summary statistics can capture much of the complexities of working with raw information, while being far more tractable.

### QTL mapping: single-locus and multiple-locus approaches

Traditional QTL mapping analysis proceeds as a genome-wide scan of a single-QTL model (**Broman et al., 2003**). At each individual locus, one conducts a t-test to determine whether there is a significant difference between the distribution of phenotypes of segregants with one parental allele and the distribution of phenotypes of segregants with the other parental allele. This comparison leads to a log odds score (LOD score) which represents the log of the ratio of the likelihood that the segregants are drawn from different distributions to the likelihood they are drawn from the same distribution. A large LOD score indicates that the locus is associated with the phenotype, either because that locus itself is causal or because it is linked to a causal locus. The LOD score is often plotted as a function of genomic position, typically in so-called Manhattan plots. Because loci linked to a causal locus are also expected to show an association with the phenotype, one expects broad peaks of elevated LOD scores around a causal locus. To identify individual causal loci, one typically looks for peaks above a genome-wide statistical significance threshold (often obtained through permutation tests). Confidence intervals for the precise location of the causal locus can then be obtained in several ways (e.g. the −1.5 LOD drop approach **Lander and Botstein, 1989**).

These classical techniques are adequate for small crosses which aim to map a relatively small number of QTL with large effects. However, they have assumptions that are clearly violated when considering highly polygenic traits. For example, when multiple causal loci are sufficiently close together, the effect size inferred at each true QTL position will reflect the sum over all linked effects (weighted by their linkage) rather than the isolated single effect. Small-effect loci that are substantially (but not completely) linked to larger-effect loci will be particularly difficult to detect from simple examination of LOD curves. These are critical problems in our regime of extremely large numbers of segregants, where our statistical power is such that entire regions of chromosomes lie above the threshold of statistical significance, due to the presence of many linked QTL (see **Figure 2**).

Therefore, to accurately infer the genetic architecture of polygenic traits, we must move to a multiple-locus framework that jointly estimates positions and effect sizes of large numbers of causal loci. This alternative framing of the problem introduces novel approaches as well as novel complications. We consider the problem in two pieces. First, given a list of putative QTL locations, we develop a method to accurately infer the maximum likelihood effect sizes given our genotype and phenotype data (the optimization problem). Second, we analyze how to select which loci to consider as putative QTL locations, and which to confirm as statistically significant (the feature selection or model search problem). The following two sections address these two problems in turn.

### Maximum-likelihood model for QTL effect sizes at multiple loci

Here, we aim to develop an inference procedure that, when given a list of loci, will jointly infer maximum-likelihood QTL effect sizes for all. There are two specific considerations regarding our data that differ from traditional QTL analyses, and that must be addressed in choosing this inference framework: (1) the heteroskedasticity in our phenotype estimates (introduced in Appendix 2), and (2) the probabilistic nature of our genotype data (introduced in Appendix 1). While our procedure makes use of classical approaches for QTL mapping, we employ them in novel ways to account for these considerations, as described in this section.

We begin by defining a general model of genetic architecture for a complex trait. Here we are given a list of loci, indexed by $k$; each has a true but unknown causal effect $s_k$. An individual segregant, indexed by $i$, will have a particular true but unknown genotype $\gamma_i = \{g_{i,k}\}$ consisting of values $g_{i,k} = 0$ or 1 at each locus. Under this model, the true phenotype of individual $i$ will be determined by its causal alleles and their effect sizes according to a function $\Phi$:

$$y_i = \Phi\left(g_{i,k}s_k\right). \tag{A3-1}$$

As mentioned above, one significant aspect of our genotype data is that we cannot assume the true binary genotype values $g_{i,k}$ to be known, as we instead obtain probabilistic estimates $\pi_{i,k} \in [0,1]$ (see Appendix 1-1.2). Although for the majority of segregants and loci this probabilistic value is in fact very close to 0 or 1 (Appendix 1-1.2), we would like to appropriately account for the genotype uncertainty as measured by these posterior probabilities.

For this, we draw inspiration from the classical method of interval mapping (**Lander and Botstein, 1989**). Instead of performing interval mapping between two markers of known genotypes, we instead perform interval mapping on one marker of uncertain genotype. For clarity, we will first examine the case of a single locus; we then extend to the multi-locus case and discuss several approximations that must be made.

In the single-locus case, where we assume one causal locus with effect size $s$, we only have two genotypes: $\gamma_0 = \{g_i = 0\}$ and $\gamma_1 = \{g_i = 1\}$. In the simplest model, these genotypes give rise to only two true phenotypes, $y_{\gamma_0} = 0$ and $y_{\gamma_1} = s$. However, in a more realistic scenario, there will be variation in the phenotypes observed from a single genotype, whether due to experimental noise or non-genetic factors. We model this as a normal distribution of phenotypes observed from one true underlying genotype: $\hat{y}_{\gamma_0} \sim \mathcal{N}(0, \sigma^2_{\gamma_0})$ and $\hat{y}_{\gamma_1} \sim \mathcal{N}(s, \sigma^2_{\gamma_1})$. (Note we use $\hat{y}$ here to refer to *measured* phenotypes, i.e. the maximum likelihood fitness estimates obtained from the barcode trajectory model in Appendix 2).

For each segregant, we observe some genotyping reads that map either to $\gamma_0$ or $\gamma_1$ at the locus of interest. From these, we write the probability for individual i to have genotype $\gamma$ as $p_{i,\gamma} = \Pr(g_i = \gamma|\text{reads})$, with $p_{i,\gamma_0} + p_{i,\gamma_1} = 1$. We then expect that the observed phenotype for that individual will be sampled from a mixture of the two normal distributions: $\hat{y}_i \sim p_{i,\gamma_0}\mathcal{N}(0, \sigma^2_{\gamma_0}) + p_{i,\gamma_1}\mathcal{N}(s, \sigma^2_{\gamma_1})$. Our goal is then to estimate the parameter of interest $s$ as well as $\sigma^2_{\gamma_0}, \sigma^2_{\gamma_1}$ by maximum likelihood. We can write the incomplete-data likelihood function for the phenotype data for all individuals as

$$\mathcal{L}(\{\hat{y}\}|s, \sigma^2_{\gamma_0}, \sigma^2_{\gamma_1}) = \prod_i^N \left[ p_{i,\gamma_0}\varphi(\hat{y}|0, \sigma^2_{\gamma_0}\%) + p_{i,\gamma_1}\varphi(\hat{y}|s, \sigma^2_{\gamma_1}) \right], \tag{A3-2}$$

where $\varphi(\hat{y}|0, \sigma^2)$ is the PDF of the normal distribution $\mathcal{N}(0, \sigma^2)$ evaluated at $\hat{y}$. The maximum likelihood estimates cannot be obtained in closed form, but can be estimated by expectation-maximization. Assuming that the genotype information is a latent variable, maximizing the expectation of the complete-data log-likelihood over the two possible genotypes (i.e. maximizing the Q function) will yield the maximum-likelihood estimates (**Dempster et al., 1977**). Here this function is:

$$E_{\{\gamma\}|\{\hat{y}\}, s, \sigma^2_{\gamma_0}, \sigma^2_{\gamma_0}} \left[ \log \mathcal{L}(\{\hat{y}\}|s, \sigma^2_{\gamma_0}, \sigma^2_{\gamma_1}) \right] = \sum_i^N \left[ w_{i,\gamma_0} \log \varphi(\hat{y}|0, \sigma^2_{\gamma_0}) + w_{i,\gamma_1} \log \varphi(\hat{y}|s, \sigma^2_{\gamma_1}) \right], \tag{A3-3}$$

where $w_{i,\gamma}$ is the conditional probability that an individual has genotype $\gamma$ given its genotype data as well as its phenotype and current estimates of the parameters:

$$w_{i\gamma_0} = \Pr(g_i = \gamma_0|p_{i,\gamma_0}, \hat{y}_i, s, \sigma^2_{\gamma_0}, \sigma^2_{\gamma_1}) = \frac{p_{i,\gamma_0}\varphi(\hat{y}|0, \sigma^2_{\gamma_0})}{p_{i,\gamma_0}\varphi(\hat{y}|0, \sigma^2_{\gamma_0}) + p_{i,\gamma_1}\varphi(\hat{y}|s, \sigma^2_{\gamma_1})}, \tag{A3-4}$$

$$w_{i,\gamma_1} = 1 - w_{i,\gamma_0}. \tag{A3-5}$$

After sufficient expectation-maximization iterations over the parameters and conditional probabilities, we obtain final maximum-likelihood estimates for $\sigma^2_{\gamma_0}, \sigma^2_{\gamma_1}$, and $s$.

We now extend this framework to multiple loci, referred to in the literature as composite-interval mapping (**Zeng, 1994**). If we consider $K$ causal loci with effects $s_k$, then we will have $2^K$ distinct genotypes in $\mathcal{G}$. Phenotypes observed from each genotype $\gamma$ will be distributed as $\hat{y}_\gamma \sim \mathcal{N}(\mu_\gamma, \sigma^2_\gamma)$, where $\mu_\gamma$ is determined by the effects $s_k$ for the alleles present in $\gamma$ according to **Equation A3-1**. Each individual's observed phenotype is now drawn from a normal mixture of all possible genotypes:

$\hat{y}_i \sim \sum_\gamma p_{i,\gamma} \mathcal{N}(\mu_\gamma, \sigma_\gamma^2)$. Here we wish to estimate all means $\mu_\gamma$, from which we can extract effect sizes $s_k$, as well as variances $\sigma_\gamma^2$. The incomplete-data likelihood function can now be written as

$$\mathcal{L}(\{\hat{y}\}|\{\mu, \sigma^2\}) = \prod_i^N \sum_\gamma^{\mathcal{G}} p_{i,\gamma} \varphi(\hat{y}_i|\mu_\gamma, \sigma_\gamma^2). \tag{A3-6}$$

As before, we will instead maximize the Q-function, or the expectation of the complete-data log likelihood over all possible genotypes:

$$E_{\{\gamma\}|\{\hat{y}\},\{\mu,\sigma^2\}} \left[ \log \mathcal{L}(\{\hat{y}\}|\{\mu, \sigma^2\}), \{g\} \right] = \sum_i^N \sum_\gamma^{\mathcal{G}} w_{i\gamma} \log \varphi(\hat{y}_i|\mu_\gamma, \sigma_\gamma^2), \tag{A3-7}$$

where the conditional probabilities are now given by

$$w_{i\gamma} = \Pr\left(g_i = \gamma|p_{i,\gamma}, \hat{y}_i, \mu_\gamma, \sigma_\gamma^2\right) = \frac{p_{i,\gamma} \varphi(\hat{y}_i|\mu_\gamma, \sigma_\gamma^2)}{\sum_\gamma^{\mathcal{G}} p_{i,\gamma} \varphi(\hat{y}_i|\mu_\gamma, \sigma_\gamma^2)}. \tag{A3-8}$$

Unfortunately, the number of possible genotypes $\gamma$, over which we must perform all of these sums, grows exponentially as $2^K$. Performing the expectation-maximization procedure with these expressions is computationally intractable if we wish to consider hundreds or thousands of potential causal loci, as we might for a highly polygenic trait.

To proceed, we make use of two approximations to the interval mapping paradigm. First, we will use a well-known approximation to interval mapping called Haley-Knott regression (**Haley and Knott, 1992**; **Kearsey and Hyne, 1994**). Here we approximate $w_{i\gamma}$ by $p_{i,\gamma}$, such that mixing proportions no longer depend on phenotype data. We also approximate the Gaussian mixture model for each individual's phenotype by a single Gaussian with the appropriate mean, and a variance that is constant over different genotypes:

$$\hat{y}_i \sim \sum_\gamma p_{i,\gamma} \mathcal{N}(\mu_\gamma, \sigma_\gamma^2) \approx \mathcal{N}\left(\sum_\gamma p_{i,\gamma}\mu_\gamma, \sigma^2\right). \tag{A3-9}$$

This is only a good approximation when the phenotype distributions for different genotypes overlap significantly (such that the mixture model is well captured by a single normal distribution). This is expected to be satisfied for polygenic traits, where many different combinations of causal alleles can give similar phenotypes. With this simplification, we can rewrite the likelihood function as

$$\mathcal{L}(\{\hat{y}\}|\{\mu\}, \sigma^2) = \prod_i^N \varphi\left(\hat{y}_i \middle| \sum_\gamma p_{i,\gamma}\mu_\gamma, \sigma^2\right). \tag{A3-10}$$

We now recognize this as an ordinary-least-squares likelihood function. We could now solve for the mean for each genotype $\mu_\gamma$ by standard linear regression methods instead of expectation-maximization iterations, and from there extract the locus effect sizes $s_k$. However, this still scales with the total genotype number $2^K$, and the genotype probabilities for each segregant $p_{i,\gamma}$ are still unspecified.

To improve this scaling, here we make our second approximation: we restrict our general model of genetic architecture in **Equation A3-1** to an additive model, meaning that causal effects at different loci are independent:

$$y_i = \sum_k^K g_{i,k} s_k. \tag{A3-11}$$

While this approximation substantially restricts the class of genetic architecture models we consider (although see below for a discussion of simple epistasis), it allows us to convert the sum over genotypes in **Equation A3-10**, which scales as $2^K$, to a sum over loci, which scales as $K$:

$$\mathcal{L}(\{\hat{y}\}|\{s\}, \sigma^2) = \prod_i^N \varphi\left(\hat{y}_i \middle| \sum_k^K \pi_{i,k} s_k, \sigma^2\right). \tag{A3-12}$$

Here we have also assumed that $p_{i,\gamma}$ is well captured by $\{\pi_{i,k}\}$, the posterior probabilities obtained from our HMM at the total set of loci $k$ under consideration. This approximation is valid when most values $\pi_{i,k}$ are close to 0 or 1, as is the case for our genotype data (see Appendix 1-1.2).

We now rewrite our log-likelihood explicitly as an ordinary-least-squares likelihood:

$$\log \mathcal{L}(\{\hat{y}\}|\{s\}, \sigma^2) = -\frac{1}{2\sigma^2} \sum_i^N \left(\hat{y}_i - \sum_k^K \pi_{i,k}s_k\right)^2 + \text{const.} \tag{A3-13}$$

In vector notation, we wish to compute

$$\arg\min_{\beta} \left\{\|Y - X\beta\|_2^2\right\}, \tag{A3-14}$$

where $Y$ is a vector of phenotype values $\hat{y}_i$ for individuals $i$, $X$ is a matrix of genotype values $\pi_{i,k}$ for all individuals $i$ at locus $k$, and $\beta$ is a vector of QTL effect sizes $s_k$ at locus $k$. We can obtain effect sizes directly by the normal equations:

$$\hat{\beta} = (X^TX)^{-1}X^TY. \tag{A3-15}$$

Unfortunately, a major assumption of this approach is that phenotype errors are assumed to be equal for all individuals. We already observed above in Appendix 2 that standard errors on phenotype estimates are correlated with frequency: lineages with low frequencies have larger errors on their fitness estimates. Since low-frequency lineages tend to be low-fitness lineages, our phenotype errors are significantly nonrandom (heteroskedastic). As discussed above, these low-frequency lineages can bias the inferred QTL effect sizes towards zero, if they are not equally represented in both parental alleles at the loci under consideration. One might consider using the standard error estimates for each lineage to perform a weighted linear regression to account for this heteroskedasticity. However, there are additional sources of noise that affect phenotyping errors on small numbers of lineages: uncertain genotyping, as described in Appendix 1; novel mutations that arise during strain construction, as described in Appendix 4-1.5; and leakage of MATα and diploid individuals, as described in Appendix 1-1.4. These errors affect only a very small fraction of strains, but they can become problematic when these individuals reach high frequency in the fitness assays (which could occur if, for example, diploidy or a novel mutation is strongly beneficial in an assay environment). These high-frequency, high-error lineages would tend to bias QTL effects away from zero, especially if they were strongly weighted due to their high frequencies. Thus, in our attempt to be conservative in QTL identification in light of multiple sources of heteroskedasticity, we prefer to use unweighted least-squares estimates with their implicit regularization towards zero.

## QTL mapping as feature selection

So far we have addressed how we can compute QTL effects given a list of known QTL. However, typically, the QTL are unknown and must be identified through a model search procedure. Although some recent models of QTL architecture suggest that nearly all the SNPs in the genome may have small but nonzero effects on some traits (**Boyle et al., 2017**), we assume that we cannot detect QTL that contribute a sufficiently small fraction of the population variance. To see this, consider a model with $k$ QTL that all contribute with equal effect sizes to the population genetic variance $\sigma_{\text{gen}}^2$, and whose locations are all known precisely. Each QTL then contributes $\sigma_{QTL}^2 = \sigma_{\text{gen}}^2/k$ to the genetic variance. Assuming each QTL is found in exactly half of the $N$ progeny in the measured panel, the effect size of each QTL is then $s_{QTL} = 2\sigma_{QTL}$. With this, we can rewrite our two-sample squared Student's t-statistic as

$$t^2 = \frac{s_{QTL}^2}{\sigma_{\text{tot}}^2(\frac{4}{N})} = \frac{N}{4}\frac{4\sigma_{\text{gen}}^2/k}{\sigma_{\text{tot}}^2} = \frac{N}{k}\left(1 - \frac{\sigma_{\text{err}}^2}{\sigma_{\text{tot}}^2}\right). \tag{A3-16}$$

We can input our panel size ($N = 100,000$), a conservative phenotyping error variance ($\frac{\sigma_{\text{err}}^2}{\sigma_{\text{tot}}^2} = 0.15$), and our desired t-statistic for a specific p-value to solve for the number of QTL $k$ that could be detected under these ideal conditions. For a genome-wide p-value of 0.05, this corresponds to $k \sim 3600$ QTL, while for a p-value of 0.01, we obtain only $k \sim 3200$ QTL. (For a simple environment such as rich media at 30°, given the phenotypic variance we observe in our assays, this would correspond to an effect size of $s_{QTL} \sim 6 \cdot 10^{-4}$.) These would already represent sparse subsets of the total number of SNPs in our cross. In practice, of course, there are several factors that will further reduce our power to detect QTL, most notably the uneven distribution of effect sizes (with more QTL having small effects than large effects), linkage between QTL, and uncertainties in QTL locations.

Thus, identifying QTL in an experimental cross can be viewed as a model selection problem: we seek to identify the sparse subset of SNPs for which the estimated effect sizes $\hat{s}$ can be distinguished from zero given our statistical power. Unfortunately, the set of possible models is very large: for $L$

polymorphisms, there are $2^L$ possible models. Further, collinearity between neighboring SNPs due to strong linkage complicates the problem, as parameters are no longer independent. Therefore, a model search strategy must choose a model that approximates the best model if we could explore the complete space. Here we focus mainly on the class of additive linear models (later including epistasis, see Section A3-3) as it extends from the marker regression framework, but the approaches we describe here are not limited to these classes of models.

One regularization approach to enforce model sparsity that is widely used in linear regression is the LASSO $l^1$ penalty. In its basic form, the objective of the LASSO is to solve:

$$\min_\beta \left\{ \|Y - X\beta\|_2^2 + \lambda\|\beta\|_1 \right\}, \tag{A3-17}$$

where, here, $Y$ is a vector of phenotypes $y_i$, $X$ is a matrix of genotype values $\pi_{i,k}$, and $\beta$ is a vector of QTL effects $s_k$. Here, sparsity is enforced by penalizing the residuals by the magnitude of the parameters, scaled by a coefficient $\lambda$. The LASSO penalty has several attractive properties, particularly that it can be easily computed while allowing model consistency under some conditions. Unfortunately, a harsh condition for model consistency is the lack of strong collinearity between true and spurious predictors (**Zhao and Yu, 2006**). This is always violated in QTL mapping studies if recombination frequencies between nearby SNPs are low. In these cases, the LASSO will almost always choose multiple correlated predictors and distribute the true QTL effect amongst them. An intuition for why this occurs is presented in the LARS algorithm, which is a general case of the LASSO (**Efron et al., 2004**). We also demonstrated this tendency to over-estimate the number of QTL using simulations (see Section A3-1.5).

Thus, to enforce sparsity in collinear cases, we turn to norm $l^\alpha$ penalties where $\alpha < 1$ (since norm penalties where $\alpha > 1$ do not enforce sparse models **Tibshirani, 1996**). The $l^0$ norm (best-subset problem) has the following objective:

$$\min_\beta \left\{ \|Y - X\beta\|_2^2 + \lambda\|\beta\|_0 \right\}, \tag{A3-18}$$

which penalizes the residuals by the number of non-zero parameters, again scaled by a coefficient $\lambda$. The least-squares equation can be rewritten as a likelihood function:

$$\min_\beta \left\{ 2\log \mathcal{L}(\text{data}|\beta) + \lambda\|\beta\|_0 \right\}. \tag{A3-19}$$

Certain choices of $\lambda$ amount to well-known asymptotic properties: $\lambda = 2$ is the Akaike information criterion (AIC), which is equivalent to model selection under leave-one-out cross validation and is known to not yield consistent model selection (**Shao, 1993**); $\lambda = \log(N)$ is the Bayesian information criterion (BIC). Unfortunately, the best-subset problem cannot be easily solved; solving $l^\alpha$ regularization where $\alpha < 1$ is NP-hard (**Natarajan, 1995**). Several approximations to this optimization function exist (for example one can develop an algorithm based on modifications of the $l^2$ penalty, e.g. **Liu et al., 2017**). However, these algorithms do not perform well in the presence of collinearity.

Therefore, we choose to employ stepwise regression with forward selection and positional refinement, which we found to have good performance (see Section A3-1.5 for assessment of its accuracy on simulated data). Similar types of forward-search approaches are used in the popular QTL mapping package R/qtl (**Arends et al., 2010**), although other implementational details differ from our methods described here.

The forward-search approach in variable selection is conceptually simple. We start with an empty model and try adding each possible remaining variable one by one, greedily choosing at each step the best available variable to include in the model (the one that minimizes the residual sum-of-squares (RSS) after optimizing the complete set of coefficients $\hat{\beta}$). This process is repeated until a desired stopping criterion (described below) is reached or until all variables have been included in the model. Practically, the optimized coefficients $\hat{\beta}$ could be obtained by directly solving the normal equations (**Equation A3-15**). However, this approach is computationally slow. Instead, this forward-search can be implemented by QR decomposition of X:

$$X = QR, \tag{A3-20}$$

$$\hat{\beta} = R^{-1}Q^T Y. \tag{A3-21}$$

This transformation is useful because adding a new variable to X does not require a complete QR re-factorization (*Daniel et al., 1976*).

One disadvantage of the forward-search approach is that once variables are included in the model, they are never removed. We can improve on this by allowing variables to explore their local neighborhood. Specifically, after each addition of a new QTL, we allow each putative QTL to "search" all locations between its flanking QTL (or the chromosome ends), selecting the location that minimizes the residuals. In this manner, we identify a set of models of increasing size where we search for local maxima at each iteration. We note that several other variations of stepwise search algorithms exist, such as variations on MCMC (*Yi, 2004*), however our motivation here is to implement a relatively simple algorithm which we found to have good performance.

A major area of contention in stepwise-search approaches is the choice of stopping criteria and/ or calculation of p-values for individual variables (*Harrell, 2001*; *Tibshirani, 1996*). We refrain from performing either of these. First, variables are added until well beyond the penalization using $\lambda = \log(N)$, which is the BIC penalty. Although this penalty asymptotically yields the correct model if all models could be explored (in practice we only have a list of $K$ models, where $K$ is the number of variables that were added), we prefer to choose a sparser model if it increases predictive performance (the correct model may not be the best predictive model), and thus enforce $\lambda \geq \log(N)$. To identify the desired $\lambda$, we perform an $a * b$ nested cross-validation where $a = 10$ and $b = 9$. Here, the total data set is split in $a = 10$ sets. One by one, a set is selected as (outer) test set for unbiased assessment of model predictive performance, and the other $a - 1 = 9$ sets are combined into a (outer) training set. We then divided each (outer) training set into $b = 9$ inner sets. One by one, an inner set is selected as (inner) test set for choosing the optimal $\lambda$, while the remaining $b - 1 = 8$ inner sets are combined into a (inner) training set. In this cross-validation procedure, the inner loops are used to choose the hyperparameter $\lambda$, and the outer loop to obtain an unbiased estimate of the performance of the model selection procedure (*Cawley and Talbot, 2010*). Finally, the outer loop can be used to choose a new hyperparameter $\lambda$, which can be used on the whole dataset to provide a final model with expected performance from this cross-validation procedure. Of course, selecting $\lambda$ based on predictive performance cannot guarantee the correct model. Note that coefficients identified through a forward stepwise regression procedure are likely to be biased (upwards) if true QTL have been excluded by the search procedure, and have enlarged variances if false positives have been included (*Smith, 2018*). However, simulations under various QTL architectures (see Section A3-1.5) have found that this approach has a tendency to underestimate the total number of QTL, stopping only when simulated QTL effects are lower than the typical error from the experiment, and to perform relatively well at identifying the QTL locations. The bias is thus likely to be small unless the majority of QTL are below the detection limit. These results align with our objective of extracting the QTL information most likely to be biologically meaningful.

Nevertheless, in a few cases, our SNPs demonstrate extreme collinearity due to tight linkage (e.g. SNPs separated by only a few basepairs). Strong collinearity in the genotype matrix leads to an ill-conditioned problem, with unstable estimates of coefficients due to numerical instabilities in QR decomposition or simply because the likelihood surface is poorly defined. In some cases, our inference procedure detects QTL at SNPs in tight linkage with opposite effects of extreme magnitude, which we believe is not a realistic feature of the genetic architecture of our traits of interest. Typically, collinear features are pre-processed before variable selection. However, in the case of dense genotypic markers, most markers exhibit some degree of collinearity with neighboring markers, and so there is no clear strategy to identify or remove problematic features. Instead, we choose to process the collinear features after feature selection. Once we have obtained the set of cross-validated QTL, we calculate the pairwise genotype correlation between neighboring QTL, and choose a degree of correlation where QTL are 'merged'. The cutoff value is chosen by cross-validation. When several QTL are merged, we treat it as one effective QTL and re-infer all effect sizes of the model. Finally, we re-optimize all the QTL positions after merging. In rare cases where the re-optimized positions again place two QTL in close proximity (and thus become highly collinear), the procedure is repeated until no more merging occurs. Thus, this procedure is similar to a backwards variable selection process, but the QTL removed by merging are only the ones that are within a high degree of collinearity with other QTL. In our analysis, we report the results of merging for a threshold that is higher than the threshold that minimizes the cross-validation error, but within one standard error of this minimum, as suggested by *Friedman et al., 2010*.

## Fine-mapping of QTL locations

Once we have obtained a list of putative QTL ("lead SNPs"), we may seek an interval estimate of the location of each QTL; the size of these intervals will be significant for interpreting causal effects of single genes or single SNPs. We expect that our resolution for localizing QTL will depend on the effect size and local recombination rate.

If we assume a single QTL, then a Bayes credible interval can be obtained by computing the maximum likelihood of the data at every SNP $l$ in the genome (*Manichaikul et al., 2006*). The 95% credible interval can then be defined as the interval I where $\sum_{l \in I} \mathcal{L}(\text{location} = l | \text{data}) \geq 0.95$. For multiple QTL, this is more complicated, as this requires joint optimization of QTL positions to obtain proper maximum likelihood estimates. Our approach to simplify this problem is to consider one QTL at a time and fix all other QTL positions. The QTL of interest is then constrained to SNP positions between its neighboring QTL (or chromosome ends), and the 95% credible interval is computed as above by obtaining the maximum likelihood of the data given that a single QTL is found at each possible SNP position between its neighboring QTL and given all detected other QTL (thus obtaining a likelihood profile for the considered positions of the QTL). We then used uniform prior on the location of the QTL (between the neighboring QTL) to derive a posterior distribution, from which one can derive an interval that exceeds 0.95.

The distribution of CI sizes in basepairs across all traits is shown in *Appendix 3—figure 1*, as well as the distribution of the number of genes at least partially overlapping each. The mean (median) CI size is ~22 kb (~12 kb) and the mean (median) gene count is 13 (7) genes. In addition, 282 QTL (11%) are mapped to a single gene (or single intergenic region), and 52 QTL (2%) are mapped to a single lead SNP.

### Credible Interval Sizes

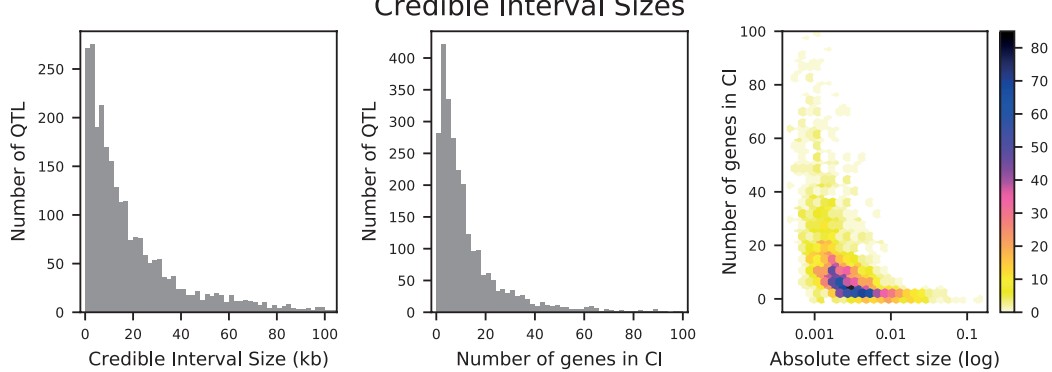

**Appendix 3—figure 1.** Distributions of credible interval (CI) sizes for inferred QTL across all traits. Left: physical size in kilobases, with 2 kb bins. Center: number of genes that overlap each CI, with 2-gene bins. Right: density plot of number of genes that overlap each CI versus absolute effect size (on a log-10 scale). The count in each bin is indicated by the colorbar.

## QTL simulations

To validate the performance of our analysis methods and its robustness to experimental parameters, we turn to simulated data where the true genomic architecture (positions and effect sizes of QTL) is completely known.

We first simulated various QTL architectures, by placing 15, 50, 150, or 1000 QTL at random locations in the genome and sampling effect sizes according to an exponential distribution with a mean effect of 1% and a random sign. Using our real genome data, we chose samples of 1000, 10,000, and 99,950 individuals and calculated their phenotypes according to an additive model. We then added varying levels of phenotypic noise to test the effect of heritability on model search performance: we added random Gaussian error with variance $\sigma_E^2$ to the true phenotypes with variance $\sigma_G^2$ to recover the desired $H^2 = \sigma_G^2/(\sigma_G^2 + \sigma_E^2)$.

With these simulated phenotypes, before turning to our forward search approach, we explored inference with the widely-used LASSO approach (model selection with $l^1$ regularization, see Section A3-1.3 above). We use the R package biglasso (*Zeng and Breheny, 2020*) which is designed for

memory-efficient computation on large datasets. For sample sizes of 10,000 segregants, we perform 5-fold cross-validation to choose the optimal regularization penalty $\lambda$ and obtain the optimized model coefficients.

We see in **Appendix 3—figure 2** that the models inferred by LASSO consistently overestimate the number of QTL by factors of 2–10, for all simulated architectures and all heritabilities, despite achieving high performance ($R^2/H^2 > 0.9$). By examining the location of inferred QTL (see **Appendix 3—figure 3**), we see that for each true QTL, the model identifies a cluster of several QTL in the vicinity. Thus the LASSO approach does identify correct genomic regions, and can give highly accurate predictive performance, but attempting any detailed analysis of the number of QTL or their positions would be misleading. This accords with our expectation: because our SNPs are dense along the genome and often in strong linkage, LASSO regularization is not effective at enforcing sparsity (see Section A3-1.3 for more detail on the drawbacks of LASSO in the presence of strong collinearities). In addition, when heritability is low, some false positives are identified that are far from any true QTL.

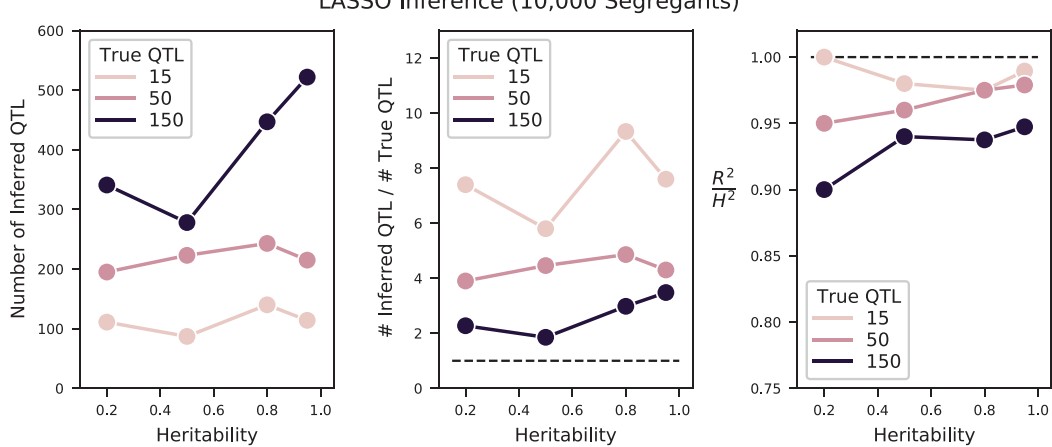

**Appendix 3—figure 2.** Performance of LASSO inference on simulated QTL architectures with 15, 50, or 150 true QTL at varying heritabilities. Left: number of inferred QTL. Center: number of inferred QTL divided by number of true QTL. The dotted line indicates a value of 1 (the correct number of QTL). Right: Proportion of variance explained ($R^2$) of the model, estimated from cross-validation, as a fraction of the simulated broad-sense heritability. $H^2$ The dotted line indicates a value of 1 (all broad-sense heritability is explained by the model).

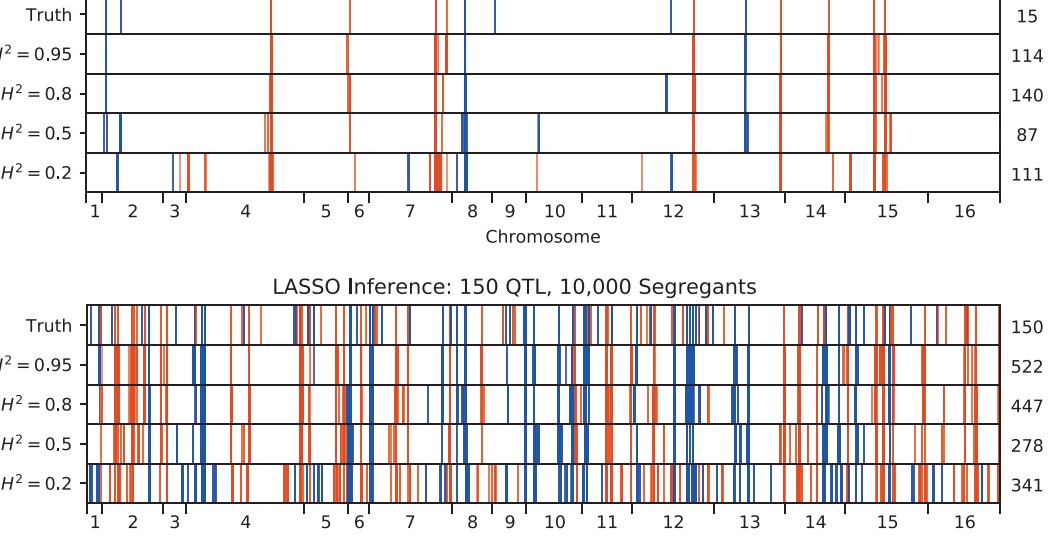

**Appendix 3—figure 3.** QTL inferred by LASSO plotted along the genome, for different heritabilities. Models were inferred with a sample of 10,000 segregants, for simulated architectures with (top) 15 or (bottom) 150 true QTL. QTL are colored red (blue) if their effect is positive (negative) and opacity is given by effect size (on a log scale). The number of QTL in the true or inferred models is given to the right.

We then turn to our cross-validated forward search algorithm as described in Section A3-1.3 to infer QTL locations and effect sizes, and assess the performance of our approach via several metrics. As expected, increasing the number of segregants increases the number of QTL identified, and this is especially evident at low heritabilities (*Appendix 3—figure 4*, left panel). When the number of segregants used is low, the variance explained can be high despite recovering only a small fraction of the correct genetic architecture (*Appendix 3—figure 4*, center panel). When the number of QTL is very high, models inferred on small sample sizes often combine several QTL into a single leading one, and the positions are more often incorrect. The QTL that are not identified tend to be of lower effect sizes or are linked in such a way that their identification is more difficult (*Appendix 3—figure 7* and *Appendix 3—figure 8*). This is more evident at very high polygenicity (1000 QTL, *Appendix 3—figure 9*). Importantly, increasing the number of segregants does not lead to overfitting: the number of QTL identified is always less than the true model, for all models and heritabilities.

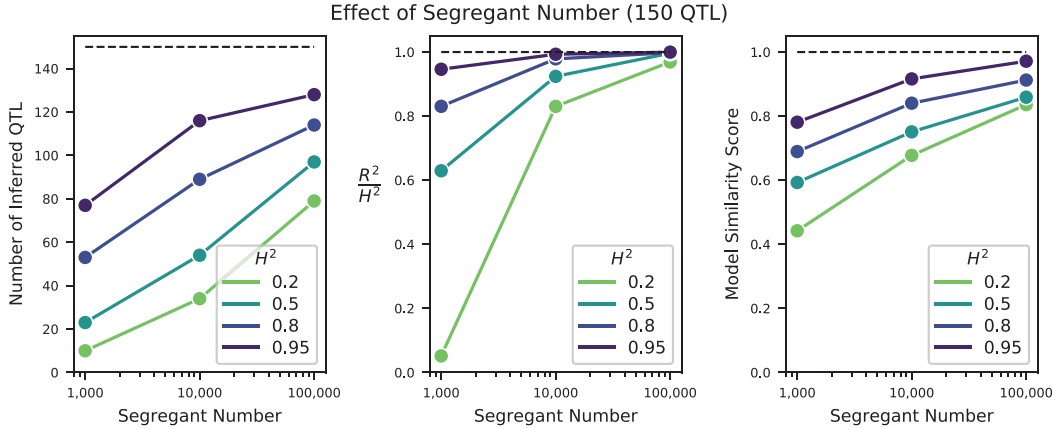

**Appendix 3—figure 4.** Effect of sample size on QTL inference. Left: Number of inferred QTL. Dashed line represents the number of QTL in the true model (150). Center: Model performance ($R^2$ on a test set of individuals)
*Appendix 3—figure 4 continued on next page*

*Appendix 3—figure 4 continued*

as a fraction of heritability $H^2$. Dashed line represents $R^2/H^2 = 1$, meaning all of the genetic variance is explained by the model. Right: Model similarity score (see Section A3-2.3) between the true and inferred models. Dashed line represents perfect recovery of the true model.

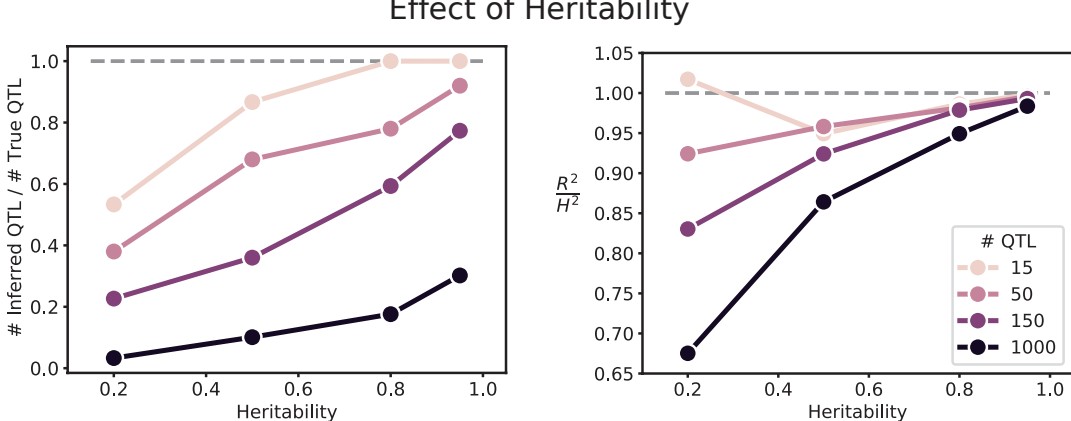

**Appendix 3—figure 5.** Effect of heritability on QTL inference (for 10,000 segregants). Left: Number of inferred QTL, as a fraction of the number of true QTL. Right: Model performance ($R^2$ on a test set of individuals) as a fraction of heritability $H^2$. Dashed line represents $R^2/H^2 = 1$, meaning all of the genetic variance is explained by the model.

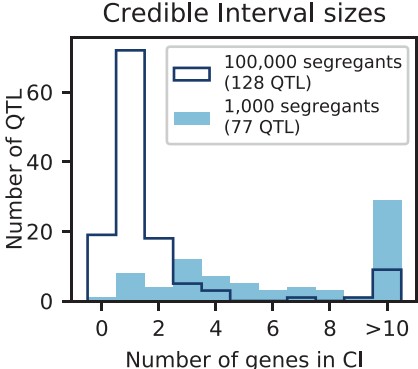

**Appendix 3—figure 6.** Number of genes contained in 95% credible intervals inferred on simulated data (at 150 QTL), for 1000 segregants (light blue shaded) and 100,000 segregants (dark blue outline). Total numbers of inferred QTL are given in the legend.

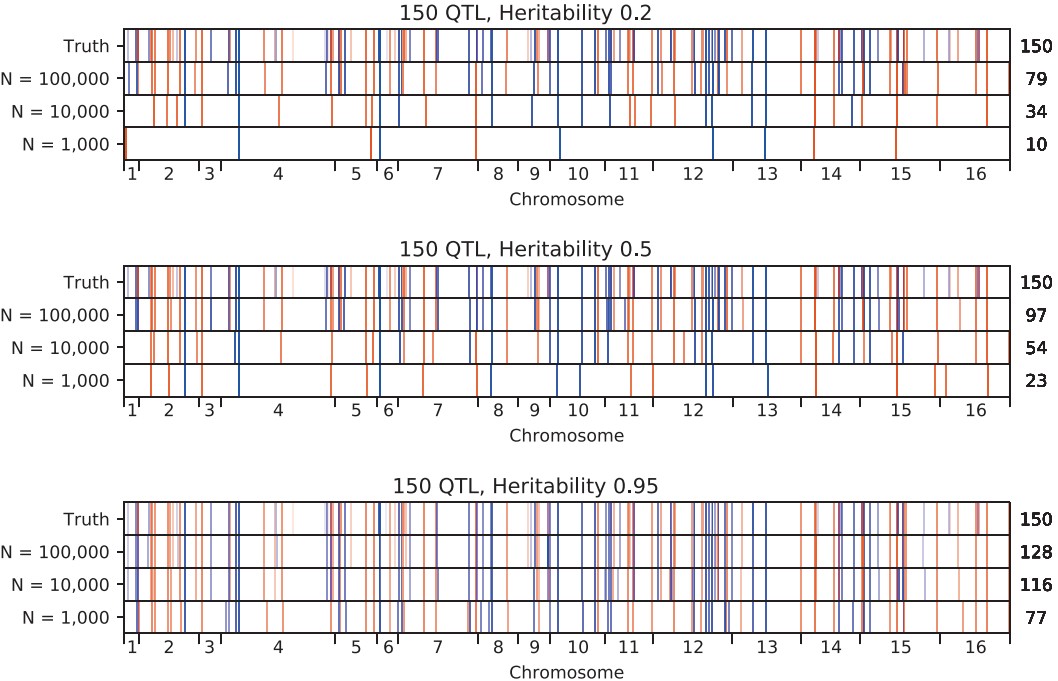

**Appendix 3—figure 7.** Inferred QTL plotted along the genome, for different sample sizes. The true model (same for all subfigures) has 150 QTL and heritability given in the subfigure title. QTL are colored red (blue) if their effect is positive (negative) and opacity is given by effect size (on a log scale). Number of inferred QTL for each model is shown on the right.

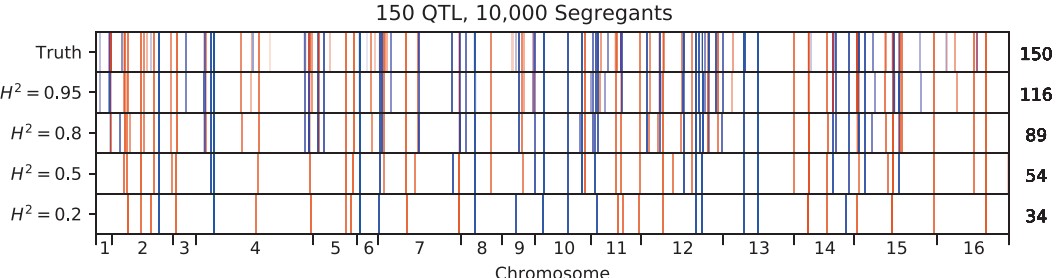

**Appendix 3—figure 8.** Inferred QTL plotted along the genome, for different heritabilities. The true model has 150 QTL, and all inference is performed on a sample size of 10,000 segregants. QTL are colored red (blue) if their effect is positive (negative) and opacity is given by effect size (on a log scale). Number of inferred QTL for each model is shown on the right.

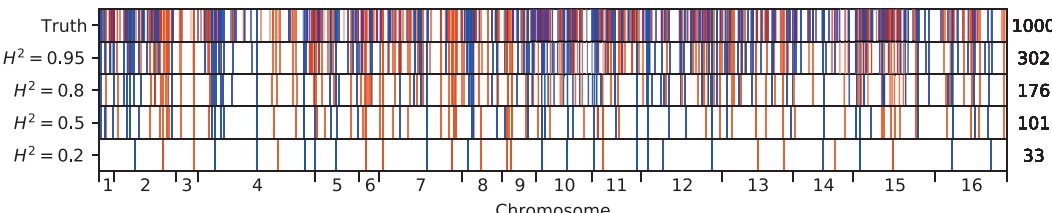

**Appendix 3—figure 9.** Inferred QTL plotted along the genome, for different heritabilities. The true model has 1000 QTL, and all inference is performed on a sample size of 10,000 segregants. QTL are colored red (blue) if their effect is positive (negative) and opacity is given by effect size (on a log scale). Number of inferred QTL for each model is shown on the right.

We can also evaluate the influence of sample size on our ability to fine-map detected QTL to causal genes. In *Appendix 3—figure 6* we show the distribution of credible interval sizes (in terms of number of genes contained) for sample sizes of 1,000 and 100,000 segregants. We observe that at larger sample sizes, we decrease the median CI size from 6 genes to one gene. In fact, for 100,000 segregants, 91 of 128 detected QTL (70%) are mapped to a single causal gene or intergenic region, as opposed to 9 of 77 detected QTL (12%) for the smaller sample size.

These results allow us to gain some intuition for how our cross-validated forward search operates. The algorithm greedily adds QTL to the model until their inclusion to the model (and thus, explained phenotypic variance) no longer exceeds the model bias (squared) and variance. Both these increase the expected error but are greatly reduced at large sample size (but are increased at low heritabilities), and the forward search can therefore identify more QTL as sample size increases. However, while our panel of spores is very large, it remains underpowered in several cases: (1) when QTL have very low effect size, therefore not contributing significantly to the phenotypic variance, and (2) when composite QTL are in strong linkage and few spores have recombination between the QTL, then the individual identification of QTL only contributes marginally to the explained variance and the forward search may also miss them.

However, even given this "true" model, it can be difficult to quantitatively compare the accuracy of a list of putative models from different model search procedures. Typically, a variable-inclusion error rate is used, by simply assessing the number of false positives and false negatives. These metrics are adequate in the absence of strong collinearity. In the QTL mapping case, we believe that it may be more beneficial to assess models based on their identification of the correct number of QTL, the correct effect size estimates, and at least an approximately correct location if not the exact nucleotide. We therefore define an alternative model accuracy metric that combines both effect size and proximity while penalizing missed QTL or false positives. We adapt the model similarity metric developed in Section A3-2.3 for comparing QTL models across traits; this metric ranges from 0 to 1 and can be loosely interpreted as the degree at which two models are correlated with each other. Here, instead of using two inferred models, we simply compare the true model to an inferred model; the resulting similarity $\mathbb{S}^2$ can then be interpreted as the accuracy of the inferred model. Results from the simulations (see *Appendix 3—figure 4*, right panel) show again that increasing the number of segregants used in the inference greatly increases the accuracy of the model, especially at low heritabilities. We highlight the difference between metrics of predictive performance such as $R^2$ (*Appendix 3—figure 4*, center panel) and this metric of model accuracy: $\mathbb{S}^2$ penalizes models explicitly for false negatives and false positives, while as we have seen, $R^2$ values can remain high for models with many false negatives (ie models with low heritability) or many false positives (as in LASSO inference, see above). Thus, our stringent model comparison framework suggests that increasing the heritability or the number of segregants lead to a more accurate view of the genetic architecture of the trait.

## LOD scores
To compare with our forward search procedure, we also calculate LOD scores and estimated effect sizes for each SNP in the classical manner, by independent t-tests (*Broman et al., 2003*; *Lander*

*and Botstein, 1989*). These values are shown in *Figure 2*. We obtain a measure of genome-wide significance by permutation test (95th percentile of genome-wide maximum LOD scores observed over 1,000 permutations) (*Broman et al., 2003*). This value is very similar across all phenotypes (mean 3.25, standard deviation 0.06) and so we use the mean value as the threshold of significance.

## Directional selection

There is often interest in understanding if phenotypic differences between strains are due to selection on new mutations for particular trait values (i.e. directional selection). Several tests have been proposed to address this question. The first one (the 'sign test') is a simple binomial test that counts whether one parental strain is enriched for beneficial or deleterious alleles in an environment (see *Figure 3—figure supplement 2* for the distribution of QTL effect sizes broken down by sign). However, this test suffers from ascertainment biases, since parents are often chosen for a cross or for phenotypic assays only if they differed sufficiently in phenotype. A similar version of this test (the 'constrained sign test'), where the ratio of beneficial to deleterious alleles is constrained by the observed phenotypic difference, can correct for this (*Orr, 1998*). However, this correction for ascertainment bias has been criticized to lack power (*Rice and Townsend, 2012*). Finally, another test (the 'variance test') compares the phenotypic variance in the segregants to the phenotypic variance of the two parents (*Fraser, 2020*). When parents have extreme values with many segregating polymorphisms, this test rejects the null hypothesis of no directional selection. This test, however, also suffers from the ascertainment bias indicated above.

When applying all three tests to our data, we find only weak evidence of directional selection (Table A3-1). This could indicate a mild concordance of our assayed environments with selection in natural environments; however it could also result from a fundamental effect of polygenicity and pleiotropy.

**Appendix 3—table 1.** Values for tests of directional selection.

| Condition | Sign test | Variance test | Constrained sign test |
|-----------|-----------|---------------|-----------------------|
| 23 C | 0.2191 | 0.0248 | 0.9035 |
| 25 C | 0.0193 | 0.0147 | 0.4001 |
| 27 C | 0.0489 | 0.0163 | 0.4786 |
| 30 C | 0.0389 | 0.011 | 0.5603 |
| 33 C | 0.0081 | 0.0053 | 0.2919 |
| 35 C | 0.19 | 0.01 | 0.953 |
| 37 C | 0.1846 | 0.0082 | 0.9536 |
| 4NQO | 0.052 | 0.2871 | 0.1275 |
| li | 0.8264 | 0.4085 | 0.6755 |
| mann | 0.356 | 0.2994 | 0.577 |
| mol | 0.4478 | 0.2428 | 0.4812 |
| sds | 0.0153 | 0.025 | 0.1887 |
| gu | 0.0032 | 0.2353 | 0.0119 |
| suloc | 0.6484 | 0.4292 | 0.6779 |
| ynb | 0.0125 | 0.1264 | 0.0798 |
| raff | 0.9999 | 0.1107 | 0.9865 |
| eth | 0.7433 | 0.4944 | 0.7562 |
| cu | 0.1807 | 0.0621 | 0.6308 |

## Pleiotropy

We infer QTL independently for each of our eighteen phenotypes. However, often we would like to ask such questions as: How do genetic architectures for two different traits relate to one another? How similar or different are the QTL (genes or causal SNPs) and their effect sizes between the two

phenotypes? In this section we describe two methods of quantifying this similarity of QTL across traits (pleiotropy).

## Gene pleiotropy

First, we use a semi-quantitative method to define pleiotropy at the level of genes. If QTL across multiple different phenotypes have lead SNPs in the same gene, they are grouped as a pleiotropic locus. The "lead gene" for this locus is the gene containing the lead SNPs; the consensus gene list for this locus is the list of genes whose coding region overlaps with the credible intervals of *all* QTL in the locus (i.e. the intersection of all credible intervals). The full list of pleiotropic QTL, with associated lead SNP positions and effect sizes for each single-trait QTL, can be found in *Supplementary file 2*. Tests for GO enrichment and intrinsically-disordered region enrichment were performed on the list of 449 lead genes with QTL from at least two environments.

## GO Enrichment

We performed Gene Ontology (GO) enrichment analysis using GO::TermFinder from the *Saccharomyces* Genome Database (*Boyle et al., 2004*). We obtained a list of all genes containing at least one SNP in our cross to use as a reference list (4,636 genes). As shown in *Table 2*, we observe 13 function terms that are significantly enriched (p-value < 0.01) in our pleiotropic QTL genes.

## Model similarity score

Next, we develop a quantitative model comparison method to directly calculate the similarity of inferred genetic architectures across traits. For two phenotypes, we have two models: model one has $n_1$ QTL at positions $X_l$ with effect sizes $\beta_l$ for $l = 1, \ldots, n_1$, while model two has $n_2$ QTL at positions $X_m$ with effect sizes $\beta_m$ for $m = 1, \ldots, n_2$. We imagine that some fraction of these QTL are 'shared', meaning the same causal locus has been detected in both models, although perhaps at somewhat different locations and somewhat different effect sizes; the remaining fraction of QTL are 'unique', or detected only in one model. 'Shared' QTL pairs should count against the model similarity to the extent that their locations and/or effect sizes are different, while 'unique' QTL should count against the model similarity by their effect size. A key question is how to assign 'shared' QTL; we address this below. For now, assume we have such an assignment, which divides all QTL into three sets: a set $S_{\text{match}}$ with matched pairs $\{l, m\}$, a set $S_1$ with the QTL unique to trait 1, and a set $S_2$ with the QTL unique to trait 2. We can then define the following cost functions:

$$C_1 = \sum_{l \in S_1} \left|\left| (X_l - p_l)\, \beta_l \right|\right|_2^2, \tag{A3-22}$$

$$C_2 = \sum_{m \in S_2} \left|\left| (X_m - p_m)\, \beta_m \right|\right|_2^2, \tag{A3-23}$$

$$C_{\text{match}} = \sum_{(l,m) \in S_{\text{match}}} \left|\left| (X_l - p_l)\, \beta_l - (X_m - p_m)\, \beta_m \right|\right|_2^2, \tag{A3-24}$$

where $p_l$ represents the allele frequency of locus $l$ in our genotype data. We account for allele frequencies because we have less power to infer QTL at loci with skewed allele frequencies (different from 50%), so QTL placed there should contribute less to the cost function. For example, if all genotypes have values 0 or 1, $C_1$ at a locus $l$ reduces to

$$\left|\left| (X_l - p_l)\, \beta_l \right|\right|_2^2 = \beta_l^2 \sum_i^N (X_l^i - p_l)^2 = \beta_l^2 p_l(1 - p_l), \tag{A3-25}$$

which is maximized at $p = 0.5$ and goes to zero at $p = 0$ or $p = 1$, as desired (see *Figure 1—figure supplement 3* for measured allele frequencies across all loci). The cost function for matched QTL incorporates the difference in effect sizes as well as the recombination-weighted distance between lead SNPs: the difference $X_l - X_m$ is determined by the recombination probabilities between those two locations, a natural way to parametrize how 'close' the model's prediction was to the true location, with the same allele frequency weighting as above. We also find that the match cost is maximized when the two genotype positions are completely uncorrelated, as desired. Finally, we note that $C_1$ can be interpreted as $C_{\text{match}}$ where $\beta_m = 0$ (and vice-versa for $C_2$), which also represents an intuitive way of describing a QTL that has not been detected in a model. In addition, we define the following cost functions for normalization purposes, representing the total cost for each model:

$$C_{\text{tot},1} = \sum_l \left|\left| (X_l - p_l)\, \beta_l \right|\right|_2^2 , \tag{A3-26}$$

$$C_{\text{tot},2} = \sum_m \left|\left| (X_m - p_m)\, \beta_m \right|\right|_2^2 , \tag{A3-27}$$

where the sum is taken over all QTL in each model, regardless of matching status. With these cost functions, we can define the model similarity metric:

$$\mathbb{S}^2 = 1 - \frac{C_{\text{match}} + C_1 + C_2}{C_{\text{tot},1} + C_{\text{tot},2}}. \tag{A3-28}$$

This metric has some superficial resemblance to Pearson correlation coefficients, with summed squares of residuals that, in our case, incorporate information about differences in lead SNP location as well as differences in effect size. QTL that are detected in only one model also contribute in proportion to their effect size (alternatively, one can imagine them as being 'matched' with a QTL in the other model with 0 effect size).

We note that this metric is not necessarily representative of correlations between expected phenotypes. Briefly, this metric is agnostic to QTL in linkage, and is meant to penalize comparisons between compound QTL and single QTL even though in principle they can produce similar phenotype distributions. Thus, this metric is better seen as a way to compare QTL *models* than as a way to compare *phenotypic predictions*. We find this metric to be especially useful for comparing architectures between different traits (to resolve whether the same loci are influencing multiple traits) as well as for testing the consistency and stability of the modeling algorithm (by comparing models inferred on simulated data to the ground truth, see Section A3-1.5 and Section A3-3.3).

To implement this metric, we must specify a method for 'matching' QTL across two models. This is nontrivial, as multiple QTL can occur near each other, fine-mapping precision can differ across traits, and a true shared QTL may have quite different effect sizes in two different traits, among other effects. These ambiguities could lead to multiple assignments, producing potentially different values for the model similarity metric. This can be viewed as a sequence alignment problem: given two sequences of QTL in the two models, we wish to find the most likely (lowest cost) alignment, where some QTL are paired (matched) and others are not (indels). Importantly, the sequence alignment framework can enforce the fact that the *order* of QTL along the chromosome cannot be permuted between the two models, as well as the fact that 'matched' QTL must be present on the same chromosome.

We use a variant of the Needleman-Wunsch algorithm (*Needleman and Wunsch, 1970*), a dynamic programming algorithm for sequence alignment, to obtain the lowest-cost assignment of QTL in the two models. The cost function matrix for the alignment is drawn from the cost functions given above, where substitutions are matches and insertions (deletions) are unique QTL in model 1 (model 2).

In this way, we obtain a single number $\mathbb{S}^2$ (ranging from 0 to 1) quantifying the overall degree of similarity between any pair of traits. These numbers are used in *Figure 3G and H*. In addition, we obtain an effect-size difference between each pair of matched QTL (and, for unmatched QTL, we consider the corresponding undetected QTL to have an effect size of 0). This gives a distribution of effect-size differences, which we show as a CDF in *Figure 3I*. However, we expect some small differences in effect size to occur by chance, even when measuring the same trait twice. To construct this null expectation, we evaluate the model comparison metric between models inferred from the 10 cross-validation partitions for a single trait. Averaging these 10 CDFs, we obtain the null CDF for each environment shown in *Figure 3I*. For all curves, we standardize the CDF by the standard deviation of effect size, to control for differences in average effect magnitude between traits.

## Epistasis
### QTL mapping with epistasis

We define epistasis between two QTL as deviation from the expectation of additive effects in the double-mutant. This definition (or basis) is different from standard orthogonal decomposition of genetic architectures but is standard in molecular genetics. This choice strongly influences the inferred coefficients and the results of variance partitioning of the epistatic components (*Huang*

*et al., 2016*), and thus we refrain from interpreting these quantities obtained from our data in an absolute sense (although see Appendix 4-1.4 for more discussion of variance partitioning). However, the detection of epistatic interactions is expected to be robust to the choice of basis.

Given the large number of polymorphisms in the pool, we would be underpowered to scan all pairs of SNPs and identify these components, even with our large pool of segregants. Scanning $L^2/2 \sim 1$ trillion SNP pairs or even $L \cdot k \sim 6$ million SNP pairs (each additive QTL against all other SNPs) is also computationally infeasible. Thus, we restrict our scan for epistatic terms to QTL that are identified in the additive-only model. One might expect that we would then miss interactions that involve SNPs with no individual effect. However, in almost all cases, interacting SNPs will have predicted individual nonzero effects $\hat{\beta} \neq 0$ in the additive model, even if their true additive effect relative to the BY genotype is zero. To see this explicitly, imagine two loci $x_1$ and $x_2$ with true individual effects $\beta_1$ and $\beta_2$ and interaction effect $\beta_{12}$. For simplicity, assume the two loci are unlinked and have allele frequencies of 0.5 (which is approximately true for most pairs of detected QTL). Then, under an additive-only model, we would estimate the effect of locus $x_1$ as

$$\hat{\beta}_1 = y_{x_1=1} - y_{x_1=0} \tag{A3-29}$$

$$= (\tfrac{1}{2} y_{x_1=1,x_2=0} + \tfrac{1}{2} y_{x_1=1,x_2=1}) - (\tfrac{1}{2} y_{x_1=0,x_2=0} + \tfrac{1}{2} y_{x_1=0,x_2=1}) \tag{A3-30}$$

$$= \left(\tfrac{1}{2}\beta_1 + \tfrac{1}{2}(\beta_1 + \beta_2 + \beta_{12})\right) - \left(\tfrac{1}{2} \cdot 0 + \tfrac{1}{2}\beta_2\right) \tag{A3-31}$$

$$= \beta_1 + \tfrac{1}{2}\beta_{12}, \tag{A3-32}$$

and, similarly, the effect of locus $x_2$ as

$$\hat{\beta}_2 = \beta_2 + \tfrac{1}{2}\beta_{12}. \tag{A3-33}$$

Therefore, we would miss one of the loci under an additive-only model only if the epistatic effect satisfies $\beta_{12} \approx -2\beta_1$ or $\beta_{12} \approx -2\beta_2$, in which case the estimated additive effect would be too small to detect in the additive-model only forward search. This condition is satisfied when a locus has no individual effect relative to the *average background*, or in other words, a zero effect size in the orthogonal basis commonly used in quantitative genetics.

In any other case, including where true individual effects relative to the BY background $\beta_1$ and/or $\beta_2$ are equal to zero but the interaction term $\beta_{12}$ is nonzero and detectable, we will identify the loci in an additive model (albeit with biased effect sizes). Of course, differences in allele frequencies will affect the value required and differences in linkage will affect our statistical power to resolve small effects, and thus some fraction of interactions will satisfy the above conditions and evade detection. However, a previous study that performed a full scan of all possible SNP pairs (using an alternative approach) found that the majority of epistatic interactions involved two loci that individually had significant additive effects (*Bloom et al., 2015*). Thus this approach enables detection of most epistatic interactions while retaining statistical power and computational feasibility.

Unfortunately, in the forward-search paradigm, there are many alternative choices for how to add single and pairwise effect parameters. As a simple exploration, we proceed by first identifying an exhaustive additive-effect non-interacting model. We then augment the model with interaction terms using the same forward-search and cross-validation strategy as previously discussed. Simulations show that this procedure can accurately find additive coefficients and generally performs well at finding specific epistatic interactions (see Section A3-3.3). The new cost function to be minimized is the following:

$$\min_{\beta,\xi} \left\{ \|Y - X\beta - (X \otimes X)\xi\|_2^2 \right\}, \tag{A3-34}$$

where $\xi$ represents a vector of epistatic interaction coefficients and we define $(X \otimes X)$ to be the matrix of interaction between columns present in $X$, calculated as pointwise multiplication of elements. For pairwise interactions, given that $X$ has $k$ columns, there are $k(k-1)/2$ columns in this matrix of interaction. Considering higher order interactions quickly becomes intractable for even $\sim 100$ QTL, and so we only consider pairwise components. With regularization, we obtain:

$$\min_{\beta,\xi} \left\{ \|Y - X\beta + \lambda_1 \|\beta\|_0 - (X \otimes X)\xi + \lambda_2 \|\xi\|_0\|_2^2 \right\}, \tag{A3-35}$$

where $\lambda_1$ is obtained by cross-validation in the initial search for additive effects, and $\lambda_2$ is obtained by cross-validation in the search for epistatic effects. By minimizing the cost function, we obtain the maximum-likelihood estimates of the epistatic components $\hat{\xi}$ by the same linear regression methods described above. We do not perform any positional refinements for pairwise epistatic components.

Similarly to strongly collinear additive effects, strongly collinear epistatic effects can have unstable coefficients and need regularization. We perform the same merging procedure as described previously, testing all-by-all correlation of pairwise epistatic positions and merging pairs of epistatic pairs beyond a correlation threshold determined by cross-validation. Once merged, we define a single epistatic pair within the merged pairs as the 'representative' epistatic interaction by choosing the pair that maximizes the likelihood.

In addition, we find that loci very close to our selection markers (MAT, HO, and CAN1) are often detected to have strong epistatic interactions, with each other and with other strong QTL across the genome (e.g. *Figure 4—figure supplement 4*). We expect this to be a result of a small number of 'leaker' individuals with the wrong alleles at the marker loci, which result in discrepant phenotypes (diploid or opposite mating type) and thus discrepant fitnesses (see discussion in Appendix 1-1.4). In *Supplementary files 3 and 4*, we have indicated these QTL and interactions involving these loci (at selection markers or immediately neighboring genes) in grey.

## Epistatic model similarity score

In order to quantitatively compare epistasis across traits, we wish to extend the model comparison framework described in Section A3-2.3 to include epistatic coefficients. However, in contrast with the additive terms, linear ordering of locus pairs is not expected to be preserved. Thus, instead of examining the problem from a sequence alignment approach, we use a linear assignment problem (LAP) approach. This approach optimizes the assignment of items from two groups (here epistatic QTL from two models) according to a defined cost matrix. A suitably defined cost matrix can incorporate costs both for matching items (i.e. assigning pairs of interactions) as well as unmatched items (i.e. interactions unique to one model or the other).

Our cost functions are defined as follows, analogously to those in the sequencing-alignment algorithm for additive model similarity, with single-locus genome values and allele frequencies replaced with interaction matrix values (i.e. products from two loci). We imagine Model one with some number $n$ of epistatic interactions at locus pairs $(l, l')$ with effect sizes $\xi_{ll'}$, and Model two with some number $m$ of epistatic interactions at locus pairs $(m, m')$ with effect sizes $\xi_{mm'}$. We divide epistatic QTL into three sets: a set $S_{\text{match}}^{ep}$ with matched pairs $\{l, l'; m, m'\}$, a set $S_1^{ep}$ with the interactions unique to trait 1, and a set $S_2^{ep}$ with the interactions unique to trait 2. We can then define the following cost functions:

$$C_1^{ep} = \sum_{(l,l') \in S_1^{ep}} \left| \left| \left( X_l X_{l'} - p_l p_{l'} \right) \xi_{ll'} \right| \right|_2^2, \tag{A3-36}$$

$$C_2^{ep} = \sum_{(m,m') \in S_2^{ep}} \left| \left| \left( X_m X_{m'} - p_m p_{m'} \right) \xi_{mm'} \right| \right|_2^2, \tag{A3-37}$$

$$C_{\text{match}}^{ep} = \sum_{(l,l';m,m') \in S_{\text{match}}^{ep}} \left| \left| \left( X_l X_{l'} - p_l p_{l'} \right) \xi_{ll'} - \left( X_m X_{m'} - p_m p_{m'} \right) \xi_{mm'} \right| \right|_2^2. \tag{A3-38}$$

Importantly, matched pairs $\{l, l'; m, m'\}$ must have $l$ and $m$ on the same chromosome (and similarly for $l'$ and $m'$).

We also define the total costs for normalization in a similar manner:

$$C_{\text{tot},1}^{ep} = \sum_{(l,l')} \left| \left| \left( X_l X_{l'} - p_l p_{l'} \right) \xi_{ll'} \right| \right|_2^2, \tag{A3-39}$$

$$C_{\text{tot},2}^{ep} = \sum_{(m,m')} \left| \left| \left( X_m X_{m'} - p_m p_{m'} \right) \xi_{mm'} \right| \right|_2^2, \tag{A3-40}$$

where the sum is taken over all interactions in each model, regardless of matching status.

We use the package lap (*Kazmar, 2020*) to optimize the assignments given these cost functions, and use the cost functions for the optimal assignment to calculate the final model similarity score. For the epistatic terms only, our score is

$$\mathbb{S}^2_{ep} = 1 - \frac{C^{ep}_{\text{match}} + C^{ep}_1 + C^{ep}_2}{C^{ep}_{\text{tot},1} + C^{ep}_{\text{tot},2}}. \tag{A3-41}$$

For the full model, including costs for the additive assignments (obtained as explained above in Section A3-2.3) as well as epistatic assignments, our score is

$$\mathbb{S}^2 = 1 - \frac{C_{\text{match}} + C_1 + C_2 + C^{ep}_{\text{match}} + C^{ep}_1 + C^{ep}_2}{C_{\text{tot},1} + C_{\text{tot},2} + C^{ep}_{\text{tot},1} + C^{ep}_{\text{tot},2}}. \tag{A3-42}$$

Thus, for any pair of epistatic models, we can compute separately the similarity of their additive or epistatic terms, in addition to the total similarity.

## Simulations of epistatic architectures

To validate the inference of epistatic interactions, we extended our QTL simulation framework (see Section A3-1.5) to incorporate the effect of epistasis. After generating some number of QTL at random locations, we generate interactions by randomly choosing pairs from these QTL. We primarily consider a model with 150 additive QTL and 150 epistatic interactions, with effect sizes for both drawn from an exponential distribution with mean effect 1% (i.e. additive and epistatic QTL are equally numerous and equal in average effect). The signs of additive and epistatic QTL were chosen randomly. As before, we calculate phenotypes for samples of 1000, 10,000, or 99,950 individuals (drawn from our real genome data). We add Gaussian noise to the phenotypes to capture different desired broad-sense heritabilities, and then infer single QTL and epistatic interactions according to the pipeline described above in Section A3-3.1.

First, we consider how many QTL (additive and epistatic) are inferred under different inference parameters, and the resulting model performance (*Appendix 3—figure 10*). We see that, as with linear models, increasing sample size increases the number of additive QTL that are detected. The effect of sample size is even more pronounced for epistatic interactions: even at heritability $H^2 = 0.95$, only one epistatic interaction is detected with a panel of 1000 segregants, while over 130 are detected with a panel of 100,000 segregants. Notably, the inference of epistasis is conservative, as it is for additive effects: the number of epistatic interactions detected is always less than the number of true interactions.

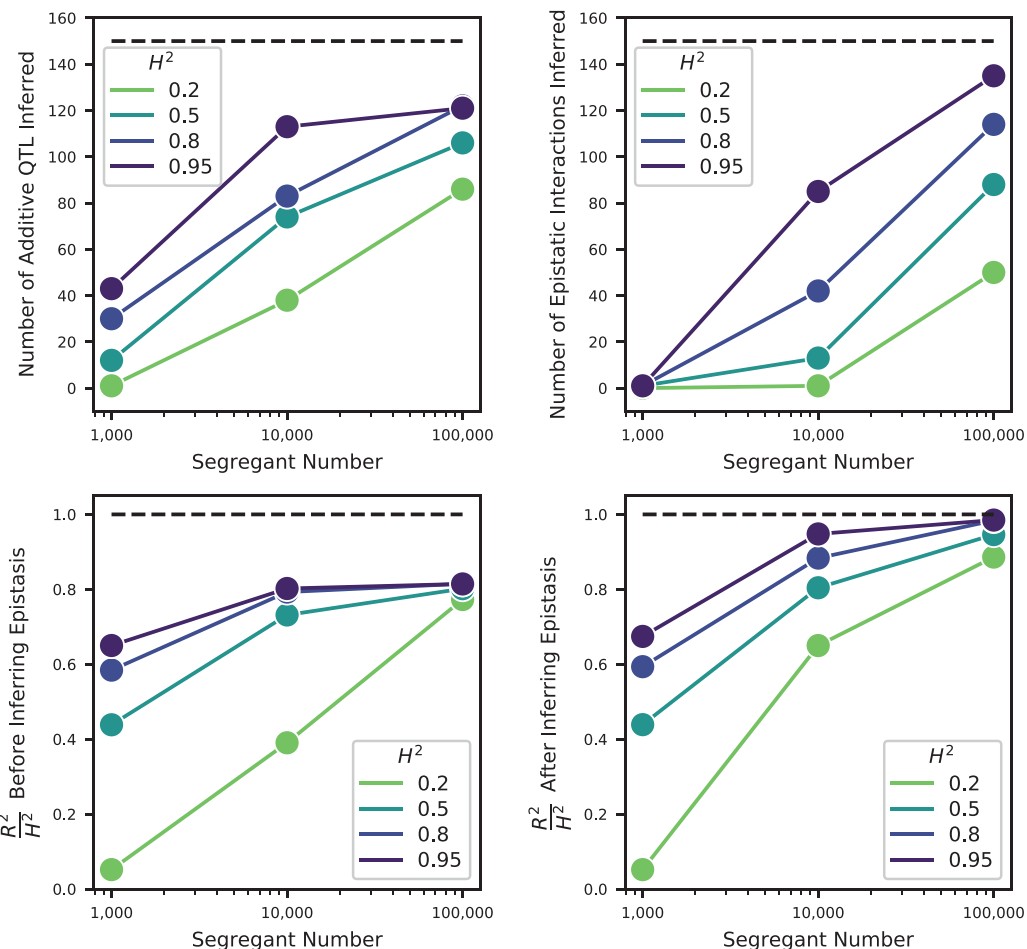

**Appendix 3—figure 10.** Effect of segregant number on QTL inference with epistasis. Top left: Number of inferred additive QTL. Dashed line represents the number of QTL in the true model (150). Top right: Number of inferred epistatic interactions. Dashed line represents the number of epistatic interactions in the true model (150). Bottom left: Model performance ($R^2$ on a test set of individuals) as a fraction of heritability $H^2$, where performance is evaluated on the optimized model with only additive terms (before epistatic interactions are inferred). Dashed line represents $R^2/H^2 = 1$, meaning all of the genetic variance is explained by the model. Bottom right: Model performance ($R^2$ on a test set of individuals) as a fraction of heritability, $H^2$ where performance is evaluated on the fully optimized model with additive and epistatic terms. Dashed line represents $R^2/H^2 = 1$, meaning all of the genetic variance is explained by the model.

We evaluate the performance of the model twice, once before inferring epistatic interactions (i.e. an optimized linear model) and once after inferring epistatic interactions (i.e. an optimized model with linear and pairwise terms). One interesting result is that despite the true architecture having equal number of additive and epistatic coefficients, and them having equal magnitude, a linear-only model can frequently explain the majority of the variance without any epistasis. In particular, the explained variance as a fraction of possible explained variance ($R^2/H^2$) saturates at $\sim 80\%$ for the best additive-only inferred models, while including epistatic interactions can increase this value to $\sim 90\%$ or above (**Appendix 3—figure 10**, bottom panels). This is not unexpected when epistasis is simulated or defined as affecting only the double mutant. However, this does cast doubt on whether the genetic architecture of the trait (and the extent of epistasis) can be inferred from variance partitioning techniques. Note that this view does not change if the model search is performed in an orthogonal epistatic basis, but is a property of the data

generation process. This phenomenon has been thoroughly explained in previous literature (**Huang et al., 2016**).

Moving beyond variance partitioning, we aim to show that our epistasis inference pipeline produces accurate estimates of effect sizes and positions for specific interaction terms, while retaining accurate estimates of effect sizes and positions for the additive terms. In **Appendix 3— figure 11** and **Appendix 3—figure 12** we plot the true epistatic interactions overlaid with epistatic interactions inferred under various parameter values. We see that in all cases, the rate of false positive detections (blue dots) is very small. Increasing the number of segregants or the heritability results in more correct identifications (green dots) and fewer false negatives (yellow dots), and the remaining false negatives tend to be of small effect size. In addition, the positions and effect sizes of the epistatic interactions are largely correct (overlap of dot position and size), although not perfect (of course, small differences are difficult to observe on the whole-genome scale).

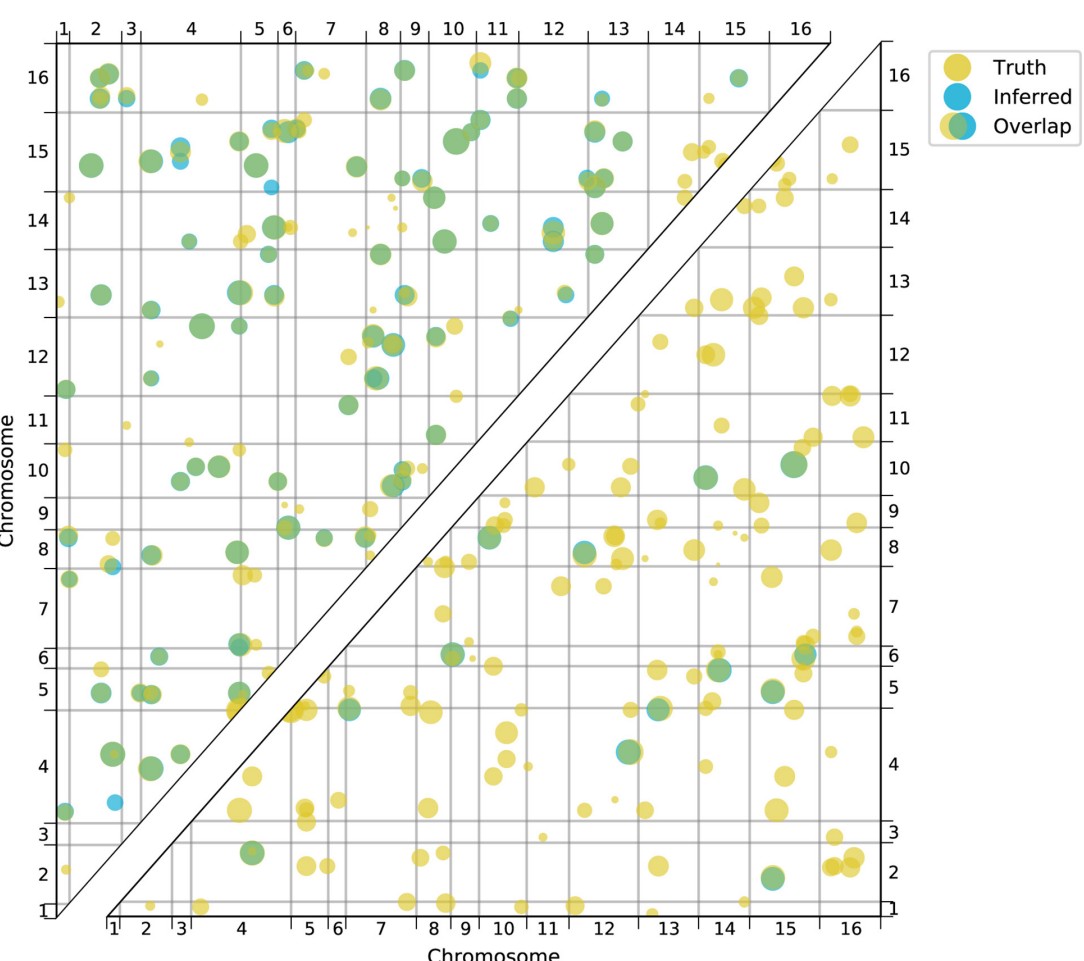

**Appendix 3—figure 11.** Comparison of true and inferred epistatic effects for different segregant numbers. Upper diagonal: True model versus inferred model with 100,000 segregants and heritability $H^2 = 0.5$; lower diagonal: true model versus inferred model with 10,000 segregants and heritability $H^2 = 0.5$. In both plots, epistatic interactions are represented by dots with position given by the genome location of the two SNPs involved and size scaled by the magnitude of effect size of the interaction (on a log scale). Epistatic interactions in the true model are colored yellow and those in the inferred model are colored blue. Dots appear green where the true and inferred interactions overlap; thus yellow dots alone represent false negatives and blue dots alone represent false positives.

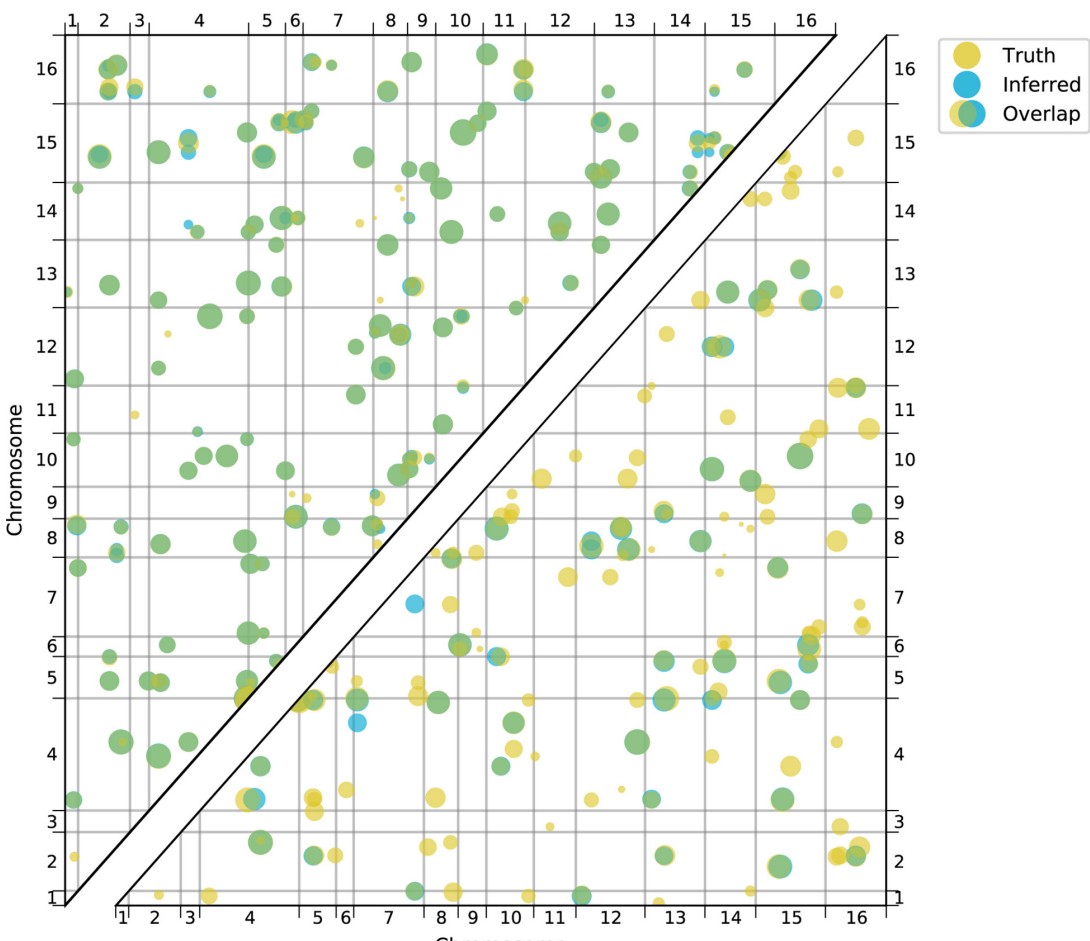

**Appendix 3—figure 12.** Comparison of true and inferred epistatic effects for different heritabilities. Upper diagonal: True model versus inferred model with 100,000 segregants and heritability $H^2 = 0.95$; lower diagonal: true model versus inferred model with 100,000 segregants and heritability.$H^2 = 0.2$ Scaling and coloring as in *Appendix 3—figure 11*.

We can also use our model comparison framework to quantify how similar our inferred additive and interaction effects are to the ground truth. In particular, we can quantify the similarity of additive coefficients (in both location and effect size) separately or in conjunction with the similarity of epistatic coefficients (in both location and effect size). We can see in *Appendix 3—figure 13* that additive effects inferred in the additive-plus-epistatic model are more accurate than those inferred in the additive-only model. As with the additive effects, the accuracy of the epistatic effects improves with increasing sample size.

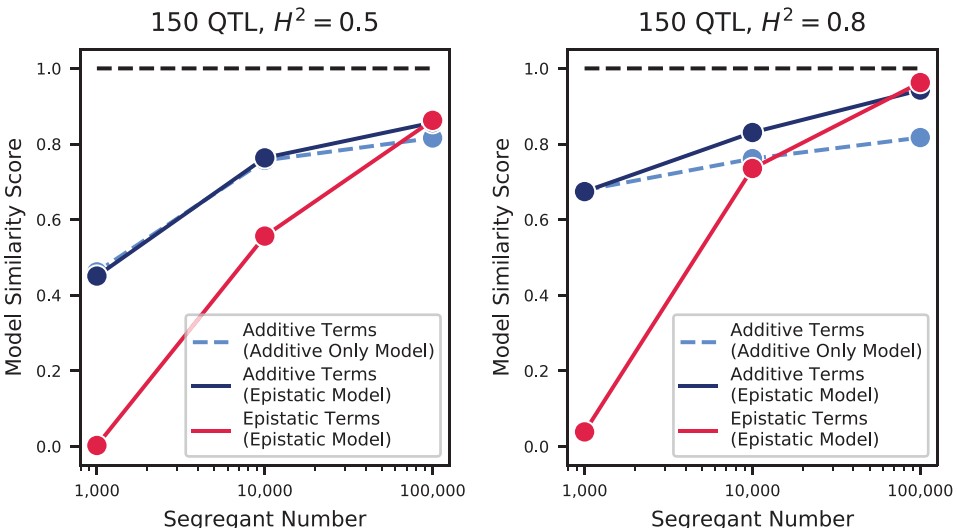

**Appendix 3—figure 13.** Model similarities for simulated epistatic architectures, as a function of segregant number. In each panel, we show the model similarity scores between the true model and the additive-only model (dashed light blue), the additive terms in the additive-plus-epistatic model (dark blue), and the epistatic terms in the additive-plus-epistatic model (red). Left: heritability of 0.5; Right: heritability of 0.8. The true model (same in all cases) has 150 additive QTL and 150 epistatic interactions.

Thus, our forward-search approach retains all of its desired features when extended to epistatic interactions: the approach is conservative, tending to underestimate the number of interactions and to find very few false positives; it accurately estimates locations and effect sizes of specific interactions; and our model similarity framework can quantify the accuracy of both additive and epistatic terms, indicating that additive terms become more accurate when epistasis is included. Even more pronounced than in the additive case, the ability to detect epistatic interactions is strongly dependent on sample size. Finally, even in models where epistatic and additive effects are equally 'important', additive-only models can infer a large majority of the variance (see further discussion in Appendix 4).

## Network analysis of epistasis

We can view the genetic architecture for each of our phenotypes as a graph, where nodes represent genes with detected QTL lead SNPs, and edges represent detected epistatic interactions between gene pairs. (In the case of intergenic lead SNPs, we assign the node or interaction to the closest gene.) This framework allows us to make quantitative comparisons of the architectures across traits and with other studies of cellular networks in yeast.

We generate graphs for each trait separately. To examine properties of these graphs, such as degree distributions or clustering coefficients, we must construct an appropriate null model. The simplest null model is an Erdos-Renyi (ER) random network (***Erdos and Renyi, 1959***), where every edge is equally likely to occur. Specifically, for each phenotype graph, we randomly permute the edges; this preserves the total number of nodes and edges but distributes the edges randomly. We generate 50 such random networks for each trait, and we refer to these below as 'random' networks.

However, we observe in our data that the additive effect size at a QTL is weakly correlated with the number of epistatic interactions detected at that QTL (Pearson correlation coefficient $r = 0.48$). This could be due to a feature of the underlying biological network; alternatively, it could be an artifact of our forward search procedure. For a QTL that has many epistatic interactions, if the apparent additive effect size in a linear model is small, the forward search may not be able to resolve all of its interactions or even resolve that specific locus; in contrast, if the apparent additive effect size is large, then the forward search will be more likely to identify more interactions at that locus. Regardless of the origin, we would like to generate a null network that recapitulates this correlation. To do so, for each phenotype network, we assign a weight to each node, given by the additive effect size of that node plus Gaussian noise (with standard deviation equal to 0.7 times the standard

deviation of additive effects in that graph). Edges are then re-assigned by sampling pairs of nodes according to these weights; this results in a correlation of $r = 0.49$ between degree and additive effect size, over 50 simulations. These networks are referred to below as 'null' networks.

We can then calculate degree distributions, average clustering coefficients, and average shortest path lengths for the data, random networks, and null networks, as shown in *Figure 4*. Average clustering coefficients (where the average is taken over all nodes in a particular phenotype network) and average shortest path lengths (where the average is taken over the largest connected component of a particular phenotype network) are calculated using the Python package NetworkX (*Hagberg et al., 2008*).

Next, we examine the network structure of epistasis across traits. For traits that are relatively similar, observing the same epistatic interactions indicates that our inference is robust. For traits that are uncorrelated, observing the same epistatic interactions indicates that pleiotropy is pervasive at the level of interactions as well as for genes. First, we aim to quantify how often the same interaction (i.e. an interaction between the same two genes) appears across our set of 18 traits. For every edge that is detected in at least one environment, we count its multiplicity (the number of trait networks in which that specific edge is present). We observe that in our data, of the 4469 edges that are detected, 3710 are distinct; 418 distinct interactions are observed twice or more, with a maximum of 10 observations. Averaging across all distinct edges, the average multiplicity is 1.21 (this corresponds to the expected number of occurrences for an edge, given that it has been observed once). We also perform this comparison with the subset of 7 environments that are uncorrelated in phenotype overall (30 °C, 4-nitroquinoline, YNB, guanidinium, lithium, molasses, and suloctidil): of the 1812 total edges, 1755 are distinct; 48 edges are observed twice or more, with a maximum of 4 observations. In this case, the average multiplicity is 1.03.

To determine if this overlap between traits is significant, we perform the same analysis for the null and random networks. For each simulation we generate null and random networks for each of the 18 traits and calculate the average multiplicity. We perform 5000 simulations to generate the empirical null distribution of average multiplicity under the two null hypotheses (random and correlation-preserving null). *Appendix 3—figure 14* shows these two null distributions along with the values from the data. Due to the sparsity of the graphs ($E \ll N(N-1)/2$ where $E$ is the number of edges and $N$ the number of nodes) and the incomplete overlap of nodes between the graphs, it is very rare for a random edge to be observed in two different traits under the ER random model, and so the distribution lies very close to 1. Even in the null networks, where large-effect nodes tend to have more interactions and thus are slightly more likely to have shared interactions across traits, average multiplicities greater than 1.02 are never observed in 5000 trials. Thus the average multiplicities from data (both over all traits and over uncorrelated traits) are significantly larger than expected (empirical p-value $< 2 \cdot 10^{-4}$).

*Appendix 3—figure 14 continued on next page*

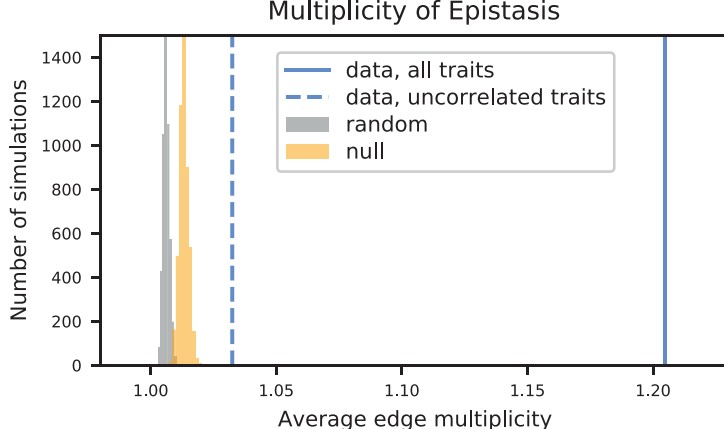

**Appendix 3—figure 14.** Empirical null distributions for average edge multiplicity (the expected number of traits in which an edge will be observed, given that it was observed in at least one trait). Histograms, data from 5000 simulations of random (grey) and null (orange) networks. Values from data are shown as vertical lines (all 18 traits, solid line; group of 7 uncorrelated traits, dashed line).

To better visualize the underlying biological network that is reflected in each trait, we use the edge multiplicity data to obtain a consensus network. We only retain edges that appear at least twice across a set of traits, and the nodes they connect. Edges are then weighted by their multiplicity (the number of times they are observed in that set of traits). We define three sets of traits: first, the total set of 18 traits; second, the group of correlated traits, that is, 23 °C, 25 °C, 27 °C, 30 °C, 33 °C, 35 °C, 37 °C, copper, ethanol, mannose, raffinose, and SDS; and third, the group of uncorrelated traits, that is, 30 °C, 4-nitroquinoline, YNB, guanidinium, lithium, molasses, and suloctidil. The consensus network for the full set is shown in *Figure 4C*; the two smaller networks are shown in *Figure 4—figure supplement 6*.

## Comparison with previously identified epistatic interactions

To assess the capability of our epistatic inference to unveil real functional interactions within the cell, we evaluate the concordance of our results with those of previous studies that employ genetic perturbation methods (*Costanzo et al., 2016*). For comparison reasons, we carry out this analysis at the gene level. Just as for the network analysis, we take the two genes in which the lead SNPs of the detected interaction are located (or, in the case of an intergenic lead SNP, the closest neighboring gene). We define a pair of genes to have been 'screened' for an interaction if both genes have been detected as additive QTL in any one of our 18 traits. (This is because our epistatic model search scans for interactions between pairs of lead SNPs already detected in an additive model, see Section A3-3.1). We further define a pair of genes to have been 'identified' if that epistatic interaction is also detected by the epistatic model search in any of the 18 traits.

*Costanzo et al., 2016* constructed an extensive library of *S. cerevisiae* single and double mutants, which they phenotype (colony size on agar plate) to identify statistical interactions between pairs of genes. For each pair of mutations screened, they calculate a genetic interaction score and an associated p-value for which they suggest cutoffs of different levels of stringency to be used when calling interactions: a Lenient, an Intermediate and a Stringent cutoff (see *Costanzo et al., 2016* for details). For each of these cutoffs, we count the number of epistatic interactions identified by either/both our epistatic model search or/and Costanzo and colleagues among the total 111,175 interactions screened by both studies.

We identify a weak but statistically significant overlap between the two studies, although only when considering the Stringent cutoff (Table A3-2). With these parameters, we observe 96 interactions that are common to both studies. Overall, (*Costanzo et al., 2016*) identifies many more interactions than our analysis, which is likely due to their genetic perturbations (mostly deletions) affecting cellular function more strongly than the SNPs in our panel. Additionally, a number of interactions are identified by only one of the studies. While this pattern could be due to false calls, it can be also due to real differences between the two studies regarding the specific genetic variants screened, phenotyping environments, and statistical power.

**Appendix 3—table 2.** Results from $\chi^2$-test of independence (with Yates' continuity correction) between A (an epistatic interaction screened by our search is identified in our search) and B (an epistatic interaction screened by our search is identified by **Costanzo et al., 2016**) at three levels of stringency for B, as described in the text.

*Overlap* is the number of gene interactions identified by both studies.

| Stringency level | P(A\|B') | P(A\|B) | Overlap | $\chi^2$ | $p_{\chi^2,\mathrm{df}=1}$ |
|---|---|---|---|---|---|
| Lenient | 2.32% | 2.25% | 494 | 0.37 | 0.54 |
| Intermediate | 2.29% | 2.44% | 187 | 0.62 | 0.43 |
| Stringent | 2.29% | 2.86% | 96 | 4.46 | 0.03 |

## Neural network

To investigate if higher-order epistatic effects existed in our dataset, we designed a densely connected neural network. The limited existing literature on predicting complex traits from genetic markers using neural network models has so far found no consistent strong improvement in performance compared to linear methods (**Zingaretti et al., 2020**). To the best of our knowledge, there is just a single study that has predicted complex traits from genetic markers with neural networks and that uses a number of genotypes comparable to ours (n > 50k) (**Bellot et al., 2018**). This study used data from the UK Biobank to predict five complex human traits using multilayer perceptron (MLP) and convolutional neural network (CNN) models alongside traditional linear models. The authors found that MLP, CNN and linear modeling methods performed similarly at predicting height, which had the highest estimated genomic heritability among the investigated traits. Overall the authors found that CNN methods were competitive to linear models but found no case where the neural network-based methods outperformed linear methods by a sizeable margin.

Several studies on smaller datasets have yielded similar results. One recent study investigated CNN models for complex trait prediction based on collections of 1000–2000 polyploid plant genomes (**Zingaretti et al., 2020**). The authors found that predictive accuracies (measured by Pearson correlation between true and predicted values) were similar to those achieved by linear models (respectively 0.01 higher and 0.02 lower average accuracies across traits for the two plant species considered). However, there was one trait out of the five considered for the first species which had a high epistatic component and where a CNN model strongly out-performed the linear model (c. 20% better than any linear model method). Another recent study that used a CNN model to predict trait values based on genetic data from c. 5 k soybean lines found that predictive accuracy was on average 0.01 higher across traits compared to a standard linear model (**Liu et al., 2019b**). Finally, a further very recent study investigating the performance of MLP and CNN models on wheat datasets with < 1000 samples found that MLP models performed better than CNN models and lead to a trial-averaged accuracy increase of 0.03–0.06 across traits compared to a linear modeling approach (**Sandhu et al., 2021**).

For our analysis, we focused on MLP architectures since we sought to investigate the capacity of a flexible class of neural network architectures, without making assumptions about the relationship between epistasis and genomic distance between SNPs. The data were initially randomly split into a training set, a validation set, and a test set (81,776; 9087; and 9087 segregants respectively). We used the Bayesian optimization function `gp_minimize` from the `scikit-optimize` 0.8.1 python library (**Head et al., 2018**) to search over possible MLP architectures, training on the training set and assessing performance on the validation set. We performed `gp_minimize` for 27 iterations searching over architectures with between 2 and 12 hidden layers, between 20 and 2400 neurons per hidden layer, and dropout between 0 and 0.5. Relu activations and batch normalization were applied to all hidden layers, and all networks had an output layer composed of 18 neurons to jointly predict values for the 18 traits. Optimization was performed using the Adam optimizer, with an adaptive learning rate schedule (with patience of 8 epochs), and early stopping (with patience of 24 epochs), training up to maximum of 500 epochs. These settings were chosen to allow sufficient time for the models to plateau in terms of validation loss. Training was performed with a batch size of 35, and mean squared error was used as the loss function.

The top three performing architectures from this search had very similar validation loss ( <0.1% difference), and considerably out-performed the next best architectures ( >5% difference), and all

three were broad (>2000 neurons per layer) shallow (two hidden layers) architectures with high dropout (>0.45). We note that, although some of the studies mentioned above considered MLP architectures, none of these studies considered architectures with >150 neurons per layer, or dropout >0.3, and none of these studies applied batch normalization. Of the three very similarly-performing models we took forward the model which was the most parsimonious in terms of the number of neurons per hidden layer as the final architecture, after rounding the dropout for this model to two decimal places. We performed a final round of tuning to select the final learning rate schedule for this architecture.

The final model was composed of three layers plus the input and output layers: two dense layers (with 2023 neurons and relu activation, followed by batch normalization), and a final dense layer (with 18 neurons and no activation function). Dropout of 0.46 was applied after the first and second hidden layers. Optimization was performed using the Adam optimizer, with a decreasing learning rate schedule – the learning rate was initially set at $1.56 \times 10^{-5}$ and was decreased by a factor of 2 after epoch number 61 and epoch number 79. The model was trained for 80 epochs, with a batch size of 35. Keras 2.2.4 (*Chollet, 2015*) with tensorflow-gpu 1.12.0 (*Abadi et al., 2015*) was used.

While our analysis provides informative results, there are some limitations. The final neural network model explains on average 4% more variance than the additive QTL model, and on average 0.5% more variance than the additive-plus-pairwise QTL model. These results compare favorably with the existing literature (described above) on genetic prediction using MLP (and CNN) models, considering the sample size of the dataset. In this context the results provide evidence to suggest the presence of higher-order epistasis in the data, and suggest that an MLP model is capable of capturing these higher-order relationships to a certain extent. However, given the large parameter space of MLP architectures, we cannot rule out the possibility that there are other MLP architecture parameter regimes that we did not explore which would outperform the architecture we identified. Furthermore, we cannot draw any conclusions about the capacities of other classes of models, such as CNN's, to make predictions on this data.

## Appendix 4

## Variance partitioning

In a typical QTL experiment, the overall phenotypic variability in the dataset arises not only due to genetic components ($H^2$, the broad-sense heritability), but also from measurement error, environmental influence or interactions, epigenetic effects, and other factors. Within the genetic component, we can further distinguish between additive SNP heritability ($h^2$, or the narrow-sense heritability), epistatic effects of various orders, and dominance effects (in non-haploid organisms). Partitioning the sum of squared deviations into these components is often used to quantify the relative importance of each component.

In our dataset, we expect several of these sources of variation to be negligible. In particular, our laboratory experiments allow for great control over the environment (as in large, well-mixed liquid culture) such that all individuals experience a constant environment. Our set of observed SNPs tags the large majority of the genetic variation present in these strains, but not all (see Section A4-1.5 for details). Epistatic effects are seen to play a major role, as shown in Appendix 3-3, but the variance partitioning of epistatic effects does not produce robust quantitative estimates of its relative importance (see discussion below in Section A4-1.4 and in Appendix 3-3). Thus, we focus our efforts here on the three major components of variation that we can estimate: phenotype measurement errors, genotype measurement errors, and tagged additive variants.

However, there are several complications in our dataset that must be considered to obtain a correct variance partitioning. First, we have uncertainty in the independent variable (genotype uncertainty) that limits the variance explained by the model (the well-known phenomenon of regression attenuation). We discuss in Section A4-1.3 how this factor can be corrected. Second, the use of the Gaussian prior in the inference of phenotypes from sequencing data (see Appendix 2) introduces some subtleties in the calculation of phenotype variance and the partitioning of phenotype error, which we address in Section A4-1.1.

### Phenotype measurement error

Our goal is to estimate the realized error component $\sigma^2_{realizederr}$ as a fraction of the total phenotype variability in the population $\sigma^2_{tot}$. We assume a compound normal distribution, where phenotypes are drawn from $y \sim \mathcal{N}(0, \sigma^2_{gen})$, where $\sigma^2_{gen}$ is the variance due to genetic factors, and measurements are drawn from $\mathcal{N}(y, \sigma^2_{err})$. If we only had a single measurement, the total variability in the population follows: $\sigma^2_{tot} = \sigma^2_{gen} + \sigma^2_{err}$. However, taking several replicate measurements will make our observed phenotypes more accurate. The realized error is given by $\sigma^2_{err}/r$, where $r$ is the number of replicate measurements, and the proportion of variance explained by the measurement error is therefore $(\sigma^2_{err}/r)/\sigma^2_{tot}$. Clearly, as the number of replicate measurements grows to infinity, the variance explained by measurement error goes to zero. However, when a limited number of replicate measurements are available, calculating this quantity requires knowledge of the error process for each measurement ($\sigma^2_{err}$). Although it is not possible to know this value a priori, it can be estimated from the Pearson correlation between replicate measurements. Specifically, because we have taken the mean of $r$ replicates as our final measurement, the fraction of the observed variance due to the error would be given by

$$\frac{\sigma^2_{err}/r}{\sigma^2_{gen} + \sigma^2_{err}/r} = \frac{1 - \langle \rho_{r_i, r_j} \rangle}{1 + (r-1)\langle \rho_{r_i, r_j} \rangle}, \tag{A4-1}$$

where the expectation values are averages over all pairwise Pearson correlations among replicates. Thus, observing no correlation between replicate measurements would imply that all the observed variance is due to measurement error. For our bulk fitness assays, the average correlation between two measurements is typically on the order of 0.9, which would indicate that about 5% of the observed phenotypic variance is due to measurement error, or that $\sigma^2_{err}$ is approximately 10% of $\sigma^2_{gen}$.

Importantly, this value is an underestimate of the variance explained by measurement error in our case. As explained in Appendix 2, we use a Gaussian prior in our phenotype inference to constrain the values of low-evidence (low-read-count) lineages. This introduces a systematic bias in our phenotype estimates for low-read-count lineages that is consistent across replicates: a lineage with sparse, noisy read count data in two replicate assays will be largely constrained by the prior, resulting in artificially similar maximum-likelihood estimates of its phenotype. These values would be precise

but not accurate, and so the reliability measures described above will be underestimates of the true phenotyping error. Indeed, even the measurement of total phenotypic variance (as calculated from the maximum-likelihood estimates) will be underestimated.

However, we also obtain measurements of standard error for each individual from the maximum-likelihood procedure (estimated from the Fisher information) that capture the effect of differing coverage, and this allows us to obtain a more accurate estimate for the realized phenotyping error and total phenotypic variance. To see this, we note that for an individual i with true phenotype $y_i$, if we have an average maximum-likelihood estimate $\hat{y}_i$ from $r$ replicate assays, we have:

$$\sqrt{r}(\hat{y}_i - y_i) \xrightarrow{d} \mathcal{N}(0, \psi\sigma_i^2), \tag{A4-2}$$

meaning that maximum likelihood estimator $\hat{y}_i$ converges in distribution to a normal distribution, with variance given by the inverse of the Fisher information $\sigma_i^2$ scaled by the overdispersion factor $\psi$. We explain the estimation of $\psi\sigma_i^2$ in Appendix 2; it scales inversely with sequencing coverage (so low-read-count individuals have high standard errors, and vice versa), and thus accurately captures the heteroskedasticity of phenotyping errors.

By averaging this variance over all $N$ individuals, we obtain an estimate of the realized error:

$$\sigma_{realizederr}^2 = \frac{\sigma_{err}^2}{r} = \frac{1}{N}\sum_i^N \psi\sigma_i^2. \tag{A4-3}$$

This value is consistently larger than the estimate obtained from *Equation A1-1*, as expected, but more accurately characterizes the scale of phenotyping measurement error, to the extent that the maximum likelihood estimates are reliable. It no longer represents a valid partition of the observed phenotypic variance, but instead a partition of the (larger) phenotypic variance we would observe if we re-sampled individual phenotypes from the distribution given in *Equation A4-2*. Because we believe the re-sampled total variance to be more accurate than the observed variance, and the realized error from *Equation A4-3* to be more accurate than that from *Equation A4-1*, we choose to use re-sampled phenotypic variance for all of the variance partitioning calculations. For completeness, *Supplementary file 6* also enumerates the underestimated total variance and realized error variance.

## Additive effects

Next, we turn our attention to estimation of the variance for the additive genetic component ($\sigma_{gen}^2$). One method of estimation is by fitting of all the SNPs as random effects in a mixed linear model (see *Yang et al., 2011* for a complete description of the method). Here the model is

$$\vec{y} = W\vec{u} + \varepsilon, \tag{A4-4}$$

where $\vec{y}$ represents a vector of phenotypes (in our case, re-sampled phenotypes), $\vec{u}$ is a vector of random SNP effects, and the errors $\varepsilon$ are normally distributed with mean zero and variance $\sigma_{err}^2$. $W$ is a standardized genotype matrix for haploid individuals, where the $ij$ th elements are

$$w_{ij} = \frac{x_{ij} - p_j}{\sqrt{p_j(1 - p_j)}}, \tag{A4-5}$$

where $p_j$ is the allele frequency at locus $j$ (allele frequencies in our panel are very close to 0.5 overall but vary at some loci; see *Figure 1—figure supplement 3*). Under this model the total (resampled) phenotypic variance can be written as

$$\sigma_{tot}^2 = WW^T\sigma_u^2 + I\sigma_e^2, \tag{A4-6}$$

with $I$ as the identity matrix. The variance explained by all the SNPs, $\sigma_{gen}^2 = N\sigma_u^2$, can be obtained by defining $A = \left(WW^T\right)/N$, which can be interpreted as a genetic relatedness matrix (GRM) between individuals. The variance components can then be estimated by restricted maximum likelihood (REML). We perform this estimation using the analysis package GCTA (*Yang et al., 2011*) (we use the flag `--make-grm-inbred` to construct a GRM for haploid individuals and perform REML estimation using default parameters). Since GCTA requires binary-valued genotypes, we binarized each locus for each individual by rounding the posterior probability from our HMM. We can then correct for the regression attenuation induced by this binarization; see Section A4-1.3 below.

In contrast to a random effects model, we can also estimate $\sigma^2_{gen}$ from the specific fixed-effect QTL model that we infer, using the PRESS statistic (**Allen, 1974**). This yields the estimates of the "variance explained" by the detected QTL. We provide estimates as fixed effect linear models based on cross-validation as described in Appendix 3-1.3:

$$\frac{\sigma^2_{\text{QTL}}}{\sigma^2_{tot}} = \left\langle 1 - \frac{\text{RSS}}{\text{TSS}} \right\rangle, \tag{A4-7}$$

obtained from the residual sum-of-squares, in the cross-validated sets, between our predicted phenotype under the QTL model and the observed phenotype. As explained in Section A4-1.1, we use phenotypes that have been resampled from the maximum likelihood parameters to obtain more accurate estimates of the phenotypic variation due to measurement error.

## Genotype measurement error

Finally, we address the issue of uncertain genotyping. As explained in Appendix 1, our genotypes are estimated rather than known values. There is an inevitable attenuation bias (or regression dilution) that will occur in the variance explained by the model when genotype values are imperfect. As an extreme example, consider the case where coverage is so low that the genotype values are close to 0.5 for the whole genome for all individuals. In this case, no modeling approaches will provide appreciable heritabilities. For understanding the genetic architecture of traits, we wish to estimate this degree of attenuation.

It is well-known that the attenuation can be corrected given known reliability estimates of the independent variables. The reliability of the genotyping is the $R^2$ between the true genotype and the estimated genotype. We can then obtain the attenuation correction by dividing the estimated variance explained by the reliability estimate (**Spearman, 1904**). In our data, we cannot easily obtain a reliability estimate for the genotype values by taking repeated measurements and calculating Pearson correlations. However, if the posterior probabilities in the genotyping are well-calibrated and accurate (as demonstrated in Appendix 1-1.3), we can take advantage of these probabilities to obtain the expected reliability estimate. Consider a position with a genotype posterior probability value $\pi_{i,k}$. If the true value is 1, which occurs with a probability $\pi_{i,k}$, then it will have a squared-residual value of $(1 - \pi_{i,k})^2$. If the true value is 0, which occurs with a probability $1 - \pi_{i,k}$, then it will have a squared-residual value of $\pi^2_{i,k}$. Thus, the expected squared-residual is

$$\text{RSS}_{i,k} = \pi_{i,k}(1 - \pi_{i,k})^2 + \pi^2_{i,k}(1 - \pi_{i,k}) = \pi_{i,k}(1 - \pi_{i,k}). \tag{A4-8}$$

We can also obtain the expected total sum of squares in a similar manner, given that the average allele frequencies are close to 0.5 in our F1 cross:

$$\text{TSS}_{i,k} = 0.5(1 - 0.5)^2 + 0.5(0 - 0.5)^2 = 0.25. \tag{A4-9}$$

Thus, the $R^2$ between the inferred and true genotype for each individual at each locus is expected to be

$$R^2_{err,gen} = 1 - 4\pi_{i,k}(1 - \pi_{i,k}). \tag{A4-10}$$

Note that the second term is exactly the genotype uncertainty metric proposed in Appendix 1-1.3. Taking the average of this value across individuals gives us our expected reliability estimate of the genotype values for each locus, from which we can correct the attenuation of our heritability estimates. Averaging across all loci, we obtain a value of $\langle 4\pi_{i,k}(1 - \pi_{i,k})\rangle = 0.067$. However, only the loci identified as QTL lead SNPs will be relevant for our regression; at these loci, the average uncertainty value is 0.082. This indicates that the variance explained by our inferred QTL models needs to be increased by a factor of $1/R^2 = 1/(1 - 0.082) \sim 9\%$, to account for regression attenuation. For our GREML inference, because we use the binarized genotypes rather than real values, this correction is slightly different:

$$R^2_{err,gen} = 1 - 4\min(\pi_{i,k}, 1 - \pi_{i,k}), \tag{A4-11}$$

where the binarized uncertainty has an average value of 0.094 across all individuals and loci, resulting in a correction of 1/(1-0.094) ~11%. The variances with and without attenuation correction are given in *Supplementary file 6* and shown in *Appendix 4—figure 1*.

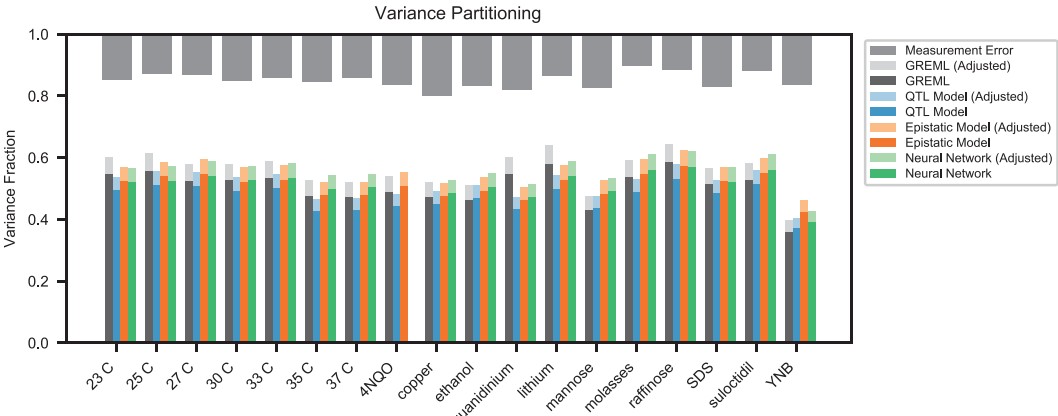

**Appendix 4—figure 1.** Variance partitioning for all traits. Phenotyping measurement error is shown at top (grey). We show the variance explained by a random-effects model (black), our inferred additive QTL model (blue), our inferred additive-plus-pairwise-epistasis QTL model (orange), and a trained deep neural network (green). Light shades indicate correction for genotyping uncertainty.

To see if this is approximately correct, we can obtain a second orthogonal measure of this attenuation correction by obtaining the $R^2$ from the final model on a subset of segregants with the highest coverage. Specifically, we use the top decile of coverage (9,995 segregants) as a held-out test set to evaluate the genetic variance explained by our QTL models. As seen in *Appendix 4—figure 2*, we find that the variance explained on the top-coverage individuals is slightly larger than that obtained by the uncertainty-adjusted QTL model on a random set of individuals, and that our estimate of measurement error is slightly smaller. This may be due to the fact that the top-coverage individuals also tend to have lower phenotyping standard errors, and indicates that our reported partitions for the full dataset are conservative. We report the attenuation values as well as the corrected and uncorrected partitions in *Supplementary file 6*, for both random and top-coverage test individuals.

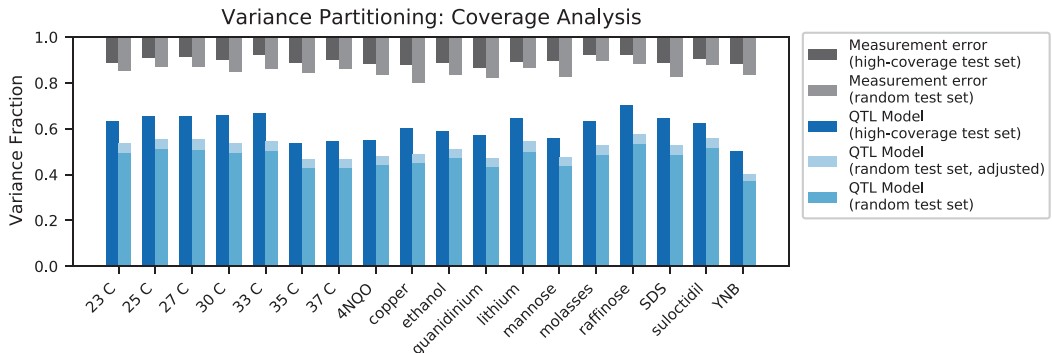

**Appendix 4—figure 2.** Variance partitioning for high-coverage individuals. Measurement error for high-coverage individuals (black) and random individuals (grey) is shown at the top. For each trait, we show the variance explained by our additive QTL model on a high-coverage test set (left) or a random test set (right; correction for genotype uncertainty shown in light blue).

## Epistatic effects

In addition to estimating the variance explained by our additive fixed-effect QTL models, we can perform a similar estimation for our fixed-effect models of additive QTL and epistatic interactions (see Appendix 3-3). These values are only marginally larger than those for additive models alone, as seen in *Appendix 4—figure 1*. This is expected due to our choice of definition (or basis) for epistatic terms, regardless of the number or strength of epistatic coefficients, and thus we cannot conclude from this the relative importance of epistatic versus additive effects (*Huang et al., 2016*). This is also consistent with our simulation results (see Appendix 3-3.3). Although there may be many pairwise or higher-order interactions that our study does not have sufficient power to resolve, from these arguments we expect that they would contribute only marginally more to the total variance explained by our inferred models.

## Other factors

We can see from *Appendix 4—figure 1* that a non-negligible fraction of the phenotypic variance remains unexplained, even after carefully accounting for measurement error and tagged genetic variation. As discussed above, undetected epistatic interactions (either pairwise or higher-order) between tagged variants are not expected to contribute appreciably (even if such interactions are strong, numerous, and meaningful). Here we discuss several experimental complications that may plausibly contribute, at least in part, to the remaining unexplained variance.

Most notably, there are sources of genetic variation that are not captured by our set of tagging SNPs. Copy number of ribosomal DNA (rDNA) is known to vary stochastically and has significant impacts on fitness (*Kwan et al., 2016*). Regions of the genome with large inversions or translocations ( > 1 kb) are observed in the RM strain as compared to BY (see Appendix 1), but they are specifically excluded from our SNP panel, due to mapping complications. If such regions contribute to differences in fitness, their effects would be largely not captured in our models. In addition, heritable variation in the mitochondrial genome sequence and 2 micron plasmid is not captured.

Novel mutations will also be acquired at some rate during the course of strain production (carried out over approximately 70 generations of growth) or during the bulk fitness assays (an additional 55 generations). Mutations that occur in the bulk fitness assays are not expected to rise to high frequency within their barcode lineage during the timescale of the assay, so their effects are likely to be averaged out. However, mutations that appear during strain production (especially barcode transformation, meiosis, and growth before single-cell sorting) can substantially alter the fitness of a barcoded lineage. The effect of such mutations on QTL inference will be small, but they introduce errors in the phenotypes of specific strains that will tend to reduce the estimate of variance explained. As an upper bound, we might imagine that every barcode receives a mutation with effects distributed normally with a standard deviation of 1% in the assay environment (a similar or larger scale to the effects we observe in the cross). In, for example, YPD at 30 °C, we would then attribute ~7.5% of the total phenotypic variance to the effect of new mutations. Thus, if the fitness effects of new mutations in our assay environments are substantial, novel mutations can contribute a non-negligible fraction of variance.

In addition, epigenetic effects may play some role, specifically due to prions as well as copy numbers of mitochondria and the 2 micron plasmid (*Hays et al., 2020*; *MacAlpine et al., 2000*; *True and Lindquist, 2000*). Such effects are outside the scope of the current study.

