## [Editor Report]

This impressive study not only expands the identification of small-effect QTL, but also reveals epistatic interactions at an unprecedented scale. The approach takes advantage of DNA barcodes to increase the scale of genetic mapping studies in yeast by an order of magnitude over previous studies, yielding a more complete and precise view of the QTL landscape and confirming widespread epistatic interactions between the different QTL.

---

## [Decision Letter]

**Decision letter after peer review:**

Thank you for submitting your article "Barcoded Bulk QTL mapping reveals highly polygenic and epistatic architecture of complex traits in yeast" for consideration by *eLife*. Your article has been reviewed by 2 peer reviewers, including Kevin J Verstrepen as Reviewing Editor and Reviewer #1, and the evaluation has been overseen by Patricia Wittkopp as the Senior Editor.

Essential Revisions:

1. Try to further expand the analysis of genetic interactions

2. Simulate a wider range of genetic architectures, including highly polygenic ones

3. Discuss the limitations of the lasso model in the discussion.

4. help readers interpret the model similiarity score.

5. Provide a more detailed rationale for the setup and interpretation of the validation experiments

6. Describe the fine-mapping methodology in more detail.

7. Discuss the impact of the limitations of the HMM model near recombination breakpoints.

8. Cite (PMID: 29487138)

*Reviewer #1:*

Nguyen Ba and coworkers report the development of a clever novel approach for QTL mapping in budding yeast, dubbed "BB-QTL". In brief, they use batches of barcoded yeasts to generated very large barcoded F1 libraries (100,000 cells), followed by a Bar-Seq approach to map the fitness of these individuals and a clever low-coverage whole-genome sequencing coupled to background knowledge of the parental sequences to map their respective genotypes. A custom analysis pipeline then allowed predicting QTLs as well as possible epistatic interactions for a set of 18 phenotypes.

The novel technology expands the precision and power of more traditional approaches. The results mainly confirm previous findings. *S. cerevisiae* phenotypes are typically influenced by many different QTLs of different nature, including coding and noncoding variation; with coding and rare variants often having a larger effect. Moreover, several QTLs located in a set of specific genes like MKT1 and IRA2, were confirmed to influence multiple phenotypes (pleiotropy). Apart from confirming previous findings, the increased power of BB-QTL does offer the advantage of having lower error rates and higher power to detect specific mutations as drivers of a QTL, including some with only small effect sizes. Together, this yields a more complete and precise view of the QTL landscape and, most importantly, confirms widespread epistatic interactions between the different QTLs. Moreover, now that the barcoded pools have been developed, it becomes relatively easy to test these in other conditions. On the other hand, the power to detect many novel (industrially-relevant) QTLs is likely limited by the inclusion of only two parental strains, one being the lab strain BY4741.

Overall, this is an impressive and interesting piece of work that not only expands the identification of small-effect QTL, but also reveals epistatic interactions at an unprecedented scale.

Still, much of the general biological conclusions are perhaps not completely novel, and I wonder whether more can be done here, to further lift the biological insight that we might gain from this unique dataset?

Specifically, I wonder whether it would also make sense to try and detect epistatic interactions in several different ways (eg simply looking at the effect of pairs of variants)? Do you find particularly strong examples of epistasis (eg complete inter-dependency of 2 mutations, or complete suppression)? Can you look for higher-order epistasis? Also, can you investigate in more detail whether epistasis partly explain the discrepancy between a given QTL's predicted effect size, and the real effect size when it is tested experimentally? Lastly, do you find evidence of selection?

One major hurdle of using QTL data to obtain improved industrial yeasts is that a QTL often seems to work in a specific background, or at least has vastly smaller effects. Similarly, in eQTL studies, it has been found that promoters often harbor several variations that together result in a limited effect on expression, likely because some (secondary) mutations were selected as suppressors of an earlier (primary) mutation. On the other hand, if a phenotype is under strong positive selection, one would expect that this compensation is absent. I wonder whether similar observations can be made in this study? For example, if one compares the fitness of the two parental strains in the different conditions, does one see systematically many more "positive" drivers in the strain with the higher fitness? Or are many "positive" QTL linked to the inferior parent? And what about the predicted epistatic interactions – do you seem more "compensatory" (negative) interactions within one genome compared to between genomes? Do you see evidence that such interacting mutations are genetically linked to (ie located in the same region)? You now validated QTL in the BY background – would their effect be different in the RM background?

*Reviewer #2:*

Ngyuyen Ba et al., investigated the genetic architecture of complex traits in yeast using a novel bulk QTL mapping approach. Their approach takes advantage of genetic tools to increase the scale of genetic mapping studies in yeast by an order of magnitude over previous studies. Briefly, their approach works by integrating unique sequenceable barcodes into the progeny of a yeast cross. These progeny were then whole genome sequenced, and bulk liquid phenotyping was carried out using the barcodes as an amplicon-based read-out of relative fitness. The authors used their approach to study the genetic architecture of several traits in ~100,000 progeny from the well-studied cross between the strains RM and BY, revealing in greater detail the polygenic, pleiotropic, and epistatic architecture of complex traits in yeast. The authors developed a new cross-validated stepwise forward search methodology to identify QTL and used simulations to show that if a trait is sufficiently polygenic, a study at the scale they perform is not sufficiently powered to accurately identify all the QTL. In the final section of the paper, the authors engineered 6 individual SNPs and 9 pairs of RM SNPs on the BY background, and measured their effects in 11 of the 18 conditions used for QTL discovery. These results highlighted the difficulty of precisely identifying the causal variants using this study design.

The conclusions in this paper are well supported by the data and analyses presented, but some aspects of the statistical mapping procedure and validation experiments deserve further attention.

In their supplementary section A.3-1.5 the authors perform QTL simulations to assess the performance of their analysis methods. Of particular interest is the performance of their cross-validated stepwise forward search methodology, which was used to identify all the QTL. However, a major limitation of their simulations was their choice of genetic architectures. In their simulations, all variants have a mean effect of 1% and a random sign. They also simulated 15, 50, or 150 QTL, which spans a range of sparse architectures, but not highly polygenic ones. It was unclear how the results would change as a function of different trait heritability. The simulations should explore a wider range of genetic architectures, with effect sizes sampled from normal or exponential distributions, as is more commonly done in the field.

In this simulation section, the authors show that the lasso model overestimates the number of causal variants by a factor of 2-10, and that the model underestimates the number of QTL except in the case of a very sparse genetic architecture of 15 QTL and heritability > 0.8. This indicates that the experimental study is underpowered if there are >50 causal variants, and that the detected QTL do not necessarily correspond to real underlying genetic effects, as revealed by the model similarity scores shown in A3-4. This limitation should be factored into the discussion of the ability of the study to break up “composite” QTL, and more generally, detect QTL of small effect.

In section A3-2.3, the authors develop a model similarity score presented in A3-4 for the simulations. The measure is similar to R^2 in that it ranges from 0 to 1, but beyond that it is not clear how to interpret what constitutes a “good” score. The authors should provide some guidance on interpreting this novel metric. It might also be helpful to see the causal and lead QTLs SNPs compared directly on chromosome plots.

The authors performed validation experiments for 6 individual SNPs and 9 pairs of RM SNPs engineered onto the BY background. It was promising that the experiments showed a positive correlation between the predicted and measured fitness effects; however, the authors did not perform power calculations, which makes it hard to evaluate the success of each individual experiment. The main text also does not make clear why these SNPS were chosen over others-was this done according to their effect sizes, or was other prior information incorporated in the choice to validate these particular variants? The authors chose to focus mostly on epistatic interactions in the validation experiments, but given their limited power to detect such interactions, it would probably be more informative to perform validation for a larger number of individual SNPs in order to test the ability of the study to detect causal variants across a range of effect sizes. The authors should perform some power calculations for their validation experiments, and describe in detail the process they employed to select these particular SNPs for validation.

In section A3-1.4, the authors describe their fine-mapping methodology, but as presented is difficult to understand. Was the fine-mapping performed using a model that includes all the other QTL effects, or was the range of the credible set only constrained to fall between the lead SNPs of the nearest QTL or the ends of the chromosome, whichever is closest to the QTL under investigation? The methodology presented on its face looks similar to the approximate Bayes credible interval described in Manichaikul et al., (PMID: 16783000). The authors should cite the relevant literature, and expand this section so that it is easier to understand exactly what was done.

The text explicitly describes an issue with the HMM employed for genotyping: "we find that the genotyping is accurate, with detectable error only very near recombination breakpoints". The genotypes near recombination breakpoints are precisely what is used to localize and fine-map QTL, and it is therefore important to discuss in the text whether the authors think this source of error impacts their results.

The use of a count-based HMM to infer genotypes has been previously described in the literature (PMID: 29487138), and this should be included in the references.

---

## [Author Response]

Essential revisions:1. Try to further expand the analysis of genetic interactions2. Simulate a wider range of genetic architectures, including highly polygenic ones3. Discuss the limitations of the lasso model in the discussion.4. help readers interpret the model similiarity score.5. Provide a more detailed rationale for the setup and interpretation of the validation experiments6. Describe the fine-mapping methodology in more detail.7. Discuss the impact of the limitations of the HMM model near recombination breakpoints.8. (Cite PMID: 29487138)

We have revised the manuscript to address all eight of these points, as explained in more detail in our responses to the individual reports of the reviewers below.

Reviewer #1:[…]Overall, this is an impressive and interesting piece of work that not only expands the identification of small-effect QTL, but also reveals epistatic interactions at an unprecedented scale.Still, much of the general biological conclusions are perhaps not completely novel, and I wonder whether more can be done here, to further lift the biological insight that we might gain from this unique dataset?Specifically, I wonder whether it would also make sense to try and detect epistatic interactions in several different ways (eg simply looking at the effect of pairs of variants)? Do you find particularly strong examples of epistasis (eg complete inter-dependency of 2 mutations, or complete suppression)? Can you look for higher-order epistasis? Also, can you investigate in more detail whether epistasis partly explain the discrepancy between a given QTL's predicted effect size, and the real effect size when it is tested experimentally? Lastly, do you find evidence of selection?

We appreciate the reviewer’s overall positive reaction, and we have conducted further analyses as suggested, as follows:

(a) To detect epistatic interactions in different ways, we explored the possibility of simply looking at the effects of pairs of variants, as the reviewer suggested. Specifically, we have added a figure supplement (Figure 4, Supp. 4) presenting an example of a two-site regression approach to estimate interaction coefficients. This is analogous to identifying additive effects by looking at LOD scores for individual variants (i.e. to the t-test shown in Figure 2), and we expect it to be much less conservative than our forward search approach (for essentially the same reasons). Consistent with this, we observe that neighboring QTL often show correlated signals of epistatic interaction, presumably due to linkage, as seen in other studies (e.g. Figure 4 in Mackay 2014 Nat Rev Genet is a very similar plot). However, the results are broadly consistent with our forward search (indeed, our forward search procedure often selects a single effect in these high LOD neighborhoods, compare Figure 4, Supp. 4B, C). Overall, the regression coefficients agree in sign and magnitude with those from the forward search, although there are a few examples where the sign of an interaction is flipped and there are disagreements among small-effect sizes (Figure 4, supp. 4D).

(b) We do observe some particularly strong examples of epistasis, including full suppression and sign-flipping. However, these cases are relatively rare, and overall we find that epistatic effects tend to shift additive predictions (as inferred in the epistatic model) by a modest amount. We have added a new figure supplement (Figure 4, Supp. 5A) to present these overall trends and to highlight the strong examples.

(c) Regarding higher-order epistasis, this is a very interesting topic but unfortunately we have not been able to systematically investigate specific higher-order interactions, both because it is not clear how to algorithmically do it within reasonable computational time, and because power is limited even in our very large sample size due to the exponential explosion of potential interactions. We do comment briefly on the potential impact of higher-order effects based on our machine-learning analysis.

(d) We do find that epistasis partly explains the discrepancy between QTL predicted effect sizes and the observed effect sizes when tested experimentally (Figure 5A; note that red points do get closer to the diagonal and the difference between prediction and observed becomes smaller in the epistatic model, with details for individual mutations all included in Figure 5, Source Data 1). We have also broken this down into the predictions for single and double mutations in Figure 5 Supp. 2. These are all imperfect tests (e.g. the “true” observed effect sizes measured experimentally in the BY background could be biased by unaccounted effects of higher-order epistasis), but we believe that this is the best way to assess this question given the inherent limitations of this type of study.

(e) We carried out three statistical tests for directional selection, finding only weak evidence for selection on assayed traits (Appendix 3, section 1.7). Nonetheless, the presence of strong highly pleiotropic QTL favoring either BY or RM does suggest strong selection in their evolutionary history (which is not in itself very surprising). We have added a reference to this analysis in the polygenicity section of the main text. In addition, we have added an analysis of biases in effect sizes and epistatic interactions that could reflect selection, as described in response to the specific suggestions of the reviewer below (see response to next comments).

One major hurdle of using QTL data to obtain improved industrial yeasts is that a QTL often seems to work in a specific background, or at least has vastly smaller effects. Similarly, in eQTL studies, it has been found that promoters often harbor several variations that together result in a limited effect on expression, likely because some (secondary) mutations were selected as suppressors of an earlier (primary) mutation. On the other hand, if a phenotype is under strong positive selection, one would expect that this compensation is absent. I wonder whether similar observations can be made in this study? For example, if one compares the fitness of the two parental strains in the different conditions, does one see systematically many more "positive" drivers in the strain with the higher fitness? Or are many "positive" QTL linked to the inferior parent? And what about the predicted epistatic interactions – do you seem more "compensatory" (negative) interactions within one genome compared to between genomes? Do you see evidence that such interacting mutations are genetically linked to (ie located in the same region)? You now validated QTL in the BY background – would their effect be different in the RM background?

These are interesting questions, and we have added a number of additional analyses along these lines:

(a) We find that there is only a small enrichment in the number of “positive” drivers in the strain with higher fitness (so there are indeed many “positive” QTL linked to the inferior parent). In addition, in most conditions the positive drivers in the strain with higher fitness tend to have slightly larger effect sizes than the positive QTL linked to the inferior parent, but this effect is also modest. We have added a discussion of this point in the main text, and have added a new Figure 3 Supp. 2 presenting these results.

(b) For almost all of our 18 phenotypes, we do find a slight excess of “compensatory” interactions that favor combinations of alleles within one genome compared to between genomes, though this is a small effect. We have added a mention of this point in the main text, and have also added a new Figure 4 Supp. 5B,C presenting these results.

(c) To investigate whether interactions tend to be genetically linked, we compared the fraction of intra-chromosomal interactions among pairs of QTL selected by our epistatic inference as compared to those not selected by it. We find that indeed there is an enrichment of intra-chromosomal interactions (from 7% to 9%; p < 1e-6). This test is confounded by the observation that QTL are not uniformly distributed in the genome, and that large effect QTL tend to have epistatic interactions with each other. Nonetheless, the pattern remains statistically significant even after the removal of any one chromosome. We have added a mention of this point in the Epistasis Results section in the main text.

(d) It is likely that the effects of reconstructed QTL would be different in the RM background (after all, our model only captures ~60% of the variance explained). However, this is impractical to test directly, and we have no reason to expect dramatically reduced predictive performance on the RM background.

Reviewer #2:Ngyuyen Ba et al., investigated the genetic architecture of complex traits in yeast using a novel bulk QTL mapping approach. Their approach takes advantage of genetic tools to increase the scale of genetic mapping studies in yeast by an order of magnitude over previous studies. Briefly, their approach works by integrating unique sequenceable barcodes into the progeny of a yeast cross. These progeny were then whole genome sequenced, and bulk liquid phenotyping was carried out using the barcodes as an amplicon-based read-out of relative fitness. The authors used their approach to study the genetic architecture of several traits in ~100,000 progeny from the well-studied cross between the strains RM and BY, revealing in greater detail the polygenic, pleiotropic, and epistatic architecture of complex traits in yeast. The authors developed a new cross-validated stepwise forward search methodology to identify QTL and used simulations to show that if a trait is sufficiently polygenic, a study at the scale they perform is not sufficiently powered to accurately identify all the QTL. In the final section of the paper, the authors engineered 6 individual SNPs and 9 pairs of RM SNPs on the BY background, and measured their effects in 11 of the 18 conditions used for QTL discovery. These results highlighted the difficulty of precisely identifying the causal variants using this study design.The conclusions in this paper are well supported by the data and analyses presented, but some aspects of the statistical mapping procedure and validation experiments deserve further attention.In their supplementary section A.3-1.5 the authors perform QTL simulations to assess the performance of their analysis methods. Of particular interest is the performance of their cross-validated stepwise forward search methodology, which was used to identify all the QTL. However, a major limitation of their simulations was their choice of genetic architectures. In their simulations, all variants have a mean effect of 1% and a random sign. They also simulated 15, 50, or 150 QTL, which spans a range of sparse architectures, but not highly polygenic ones. It was unclear how the results would change as a function of different trait heritability. The simulations should explore a wider range of genetic architectures, with effect sizes sampled from normal or exponential distributions, as is more commonly done in the field.

As suggested, we have expanded the range of simulations we explore in the revised manuscript. We note that the original simulations discussed in the manuscript involve exponentially distributed effect sizes (with a mean of 1% and random sign) at multiple different heritability values. These are described in Figures A3-4 and A3-5. We also simulated epistatic terms (Figure A3-3.3). In the revision, we have broadened the simulations to add more ‘highly polygenic’ architectures (1000 QTL). We find that the algorithm still performs well, though worse than when 150 QTL are simulated. The forward search behaves in a fairly intuitive way: QTLs get added when the contribution of a true QTL to the explained phenotypic variance overcomes the model bias and variance. QTLs are only missed if their effect size is too low to contribute significantly to phenotypic variance, or if they are in strong linkage and thus their independent discovery barely increases the variance explained (which is all finally controlled by the trait heritability). At much higher polygenicity, composite QTL can be detected as a single QTL when their sum contribute to phenotypic variance, and get broken up if and only if independent sums also contribute significantly to phenotypic variance. Of course, there are many ways to break up composite QTL, but the algorithm proceeds in a greedy fashion focusing on unexplained variance. We have also explored cases with multiple QTL of the same effect, and with different mean effects or different number of epistatic terms, but we found these results were largely redundant. To summarize these conclusions, we have added the following discussion at the end of the Results section:

“The behavior of this approach is simple and intuitive: the algorithm greedily adds QTL if their expected contribution to the total phenotypic variance exceeds the bias and increasing variance of the forward search procedure, which is greatly reduced at large sample size. Thus, it may fail to identify very small effect size variants and may fail to break up composite QTL in extremely strong linkage.”

We have also added additional clarification in the Appendix:

“These results allow us to gain some intuition for how our cross-validated forward search operates. The algorithm greedily adds QTL to the model until their inclusion to the model (and thus, explained phenotypic variance) no longer exceeds the model bias (squared) and variance. Both these increase the expected error but are greatly reduced at large sample size (but are increased at low heritability values), and the forward search can therefore identify more QTL as sample size increases. However, while our panel of spores is very large, it remains underpowered in several cases: (1) when QTL have very low effect size, therefore not contributing significantly to the phenotypic variance, and (2) when composite QTL are in strong linkage and few spores have recombination between the QTL, then the individual identification of QTL only contributes marginally to the explained variance and the forward search may also miss them.”

In this simulation section, the authors show that the lasso model overestimates the number of causal variants by a factor of 2-10, and that the model underestimates the number of QTL except in the case of a very sparse genetic architecture of 15 QTL and heritability > 0.8. This indicates that the experimental study is underpowered if there are >50 causal variants, and that the detected QTL do not necessarily correspond to real underlying genetic effects, as revealed by the model similarity scores shown in A3-4. This limitation should be factored into the discussion of the ability of the study to break up "composite" QTL, and more generally, detect QTL of small effect.

We agree with some aspects of this comment, but the details are a bit subtle. First, we note that the definition of underpowered depends on the specifics of the QTL assumed in the simulation. In addition, many of the simulations were performed at 10,000 segregants, not at 100,000, with no effort to enforce a minimum effect size, or minimum distance between QTL. For example, if 100 QTL are all evenly spaced (in recombination space) and all have the same effect such that they all contribute the same to the phenotypic variance, then the algorithm is in principle maximally powered to detect these. This is why our algorithm is capable of finding >100 QTL per environment. On the other hand, just 2 QTL in complete linkage cannot be distinguished and no panel size will be able to detect these.

However, we do agree with the general need to discuss the limitations in more detail and have clarified these concerns in the ‘Polygenicity’ result section. We have also reiterated the limitations of the LASSO approach within the simulation section. The motivation for an L0 normalization in this data was first discussed in the section A3-1.3:

“Unfortunately, a harsh condition for model consistency is the lack of strong collinearity between true and spurious predictors (Zhao and Yu, 2006). This is always violated in QTL mapping studies if recombination frequencies between nearby SNPs are low. In these cases, the LASSO will almost always choose multiple correlated predictors and distribute the true QTL effect amongst them.”

In section A3-2.3, the authors develop a model similarity score presented in A3-4 for the simulations. The measure is similar to R^2 in that it ranges from 0 to 1, but beyond that it is not clear how to interpret what constitutes a "good" score. The authors should provide some guidance on interpreting this novel metric. It might also be helpful to see the causal and lead QTLs SNPs compared directly on chromosome plots.

We agree that this was unclear, and have added additional discussion in the main text describing how to interpret the model similarity score. Essentially, the score is a Pearson’s correlation coefficient on the model coefficient (as defined in section A3-2.3, after equation A3-28). However, given a single QTL that spans two SNPs in close linkage, a pure Pearson’s correlation coefficient would have high variance, as subtle noise in the data could lead to one SNP being called the lead SNP vs the other, and two models that call the same QTL might have either 100% correlation, or 0% correlation. Instead, our model similarity score ‘aligns’ these predicted QTL before obtaining the correlation coefficient. The degree at which QTL are aligned are based on penalties with respect to collinearity (or linkage) between the SNPs, and the maximum possible score is obtained by dynamic programming. Similar to sequence alignments between two completely unrelated sequences, a score of 0 is unlikely to occur on sufficiently large models as at least a few QTL can usually be paired (erroneously). We have also added a mention in the main text referring to Figures A3-3, A3-7, A3-8, A3-9, which show the causal and lead QTL SNP directly on the chromosome plots.

The authors performed validation experiments for 6 individual SNPs and 9 pairs of RM SNPs engineered onto the BY background. It was promising that the experiments showed a positive correlation between the predicted and measured fitness effects; however, the authors did not perform power calculations, which makes it hard to evaluate the success of each individual experiment. The main text also does not make clear why these SNPS were chosen over others-was this done according to their effect sizes, or was other prior information incorporated in the choice to validate these particular variants? The authors chose to focus mostly on epistatic interactions in the validation experiments, but given their limited power to detect such interactions, it would probably be more informative to perform validation for a larger number of individual SNPs in order to test the ability of the study to detect causal variants across a range of effect sizes. The authors should perform some power calculations for their validation experiments, and describe in detail the process they employed to select these particular SNPs for validation.

We agree with the thrust of the comment, but some of the suggestions are impossible to implement because of practical constraints on the experimental methods (and to a lesser extent on the model inference). First, we chose the SNPs to reconstruct based on three main factors: (a) to ensure that we are validating the right locus, the model must have a confident prediction that specific SNP is causal, (b) the predicted effect must be large enough in at least one environment that we would expect to reliably measure it given the detection limits of our experimental fitness measurements, and (c) the SNP must be in a location that is amenable to CRISPR-Cas9 or Delitto Perfetto reconstruction. In practice, this means that it is impossible to validate SNPs across a wide range of effect sizes, as smaller-effect SNPs have wider confidence intervals around the lead SNP (violating condition a) and have effects that are harder to measure experimentally (violating condition b). In addition, because the cloning constraints mentioned in (c) require experimental testing for each SNP we analyze, it is much easier to construct combinations of a smaller set of SNPs than a larger set of individual SNPs. Together, these considerations motivated our choice of specific SNPs and of the overall structure of the validation experiments (6 individual and 9 pairs, rather than a broader set of individual SNPs).

In the revised manuscript, we have added a more detailed discussion of these motivations for selecting particular SNPs for validation, and mention the inherent limitations imposed by the practical constraints involved. We have also added a description of the power and resolution of the experimental fitness measurements of the reconstructed genotypes (we can detect approximately ~0.5% fitness differences in most conditions). We are unsure if there are any other types of power calculations the reviewer is referring to, but we are only attempting to note an overall positive correlation between predicted and measured effects, not making any claims about the success of any individual validation (these can fail for a variety of reasons including experimental artifacts with reconstructions, model errors in identifying the correct causal SNP, unresolved higher-order epistasis, and noise in our fitness measurements, among others).

In section A3-1.4, the authors describe their fine-mapping methodology, but as presented is difficult to understand. Was the fine-mapping performed using a model that includes all the other QTL effects, or was the range of the credible set only constrained to fall between the lead SNPs of the nearest QTL or the ends of the chromosome, whichever is closest to the QTL under investigation? The methodology presented on its face looks similar to the approximate Bayes credible interval described in Manichaikul et al., (PMID: 16783000). The authors should cite the relevant literature, and expand this section so that it is easier to understand exactly what was done.

We have attempted to clarify section A3-1.4. As the reviewer correctly points out, the fine mapping for a QTL is performed by scanning an interval between neighboring detected QTL (on either side) and using a model that includes all other QTL. For example, if a detected QTL is a SNP found in a closed interval of 12 SNPs produced by its two neighboring QTL, 10 independent likelihoods are obtained (re-optimizing all effect sizes for each), and a posterior probability is obtained for each of the ten possible positions.

We have cited the recommended paper, as our approach is indeed based on an approximate Bayes credible interval similar to the one described in that study (using all SNPs instead of markers). We have added the following sentence to the A3-1.4 section at the end of the second paragraph (similar to the analogous paragraph in Manichaikul et al.,):

“[…] as above by obtaining the maximum likelihood of the data given that a single QTL is found at each possible SNP position between its neighboring QTL and given all detected other QTL (thus obtaining a likelihood profile for the considered positions of the QTL). We then used a uniform prior on the location of the QTL to derive a posterior distribution, from which one can derive an interval that exceeds 0.95.”

Some typos referring to a ‘confidence’ interval were also changed to ‘credible’ interval.

The text explicitly describes an issue with the HMM employed for genotyping: "we find that the genotyping is accurate, with detectable error only very near recombination breakpoints". The genotypes near recombination breakpoints are precisely what is used to localize and fine-map QTL, and it is therefore important to discuss in the text whether the authors think this source of error impacts their results.

This is a good point, we have added a reference in the main text to the Appendix section (A1-1.4) that has an extensive discussion and analysis of the effect of recombination breakpoint uncertainties on fine-mapping.

The use of a count-based HMM to infer genotypes has been previously described in the literature (PMID: 29487138), and this should be included in the references.

We now also add this citation to our text on the count-based HMM.